# Solving Probabilistic Verification Problems of Neural Networks using Branch and Bound

David Boetius [1]   Stefan Leue [1]   Tobias Sutter [1]

## Abstract

Probabilistic verification problems of neural networks are concerned with formally analysing the output distribution of a neural network under a probability distribution of the inputs. Examples of probabilistic verification problems include verifying the demographic parity fairness notion or quantifying the safety of a neural network. We present a new algorithm for solving probabilistic verification problems of neural networks based on an algorithm for computing and iteratively refining lower and upper bounds on probabilities over the outputs of a neural network. By applying state-of-the-art bound propagation and branch and bound techniques from non-probabilistic neural network verification, our algorithm significantly outpaces existing probabilistic verification algorithms, reducing solving times for various benchmarks from the literature from tens of minutes to tens of seconds. Furthermore, our algorithm compares favourably even to dedicated algorithms for restricted probabilistic verification problems. We complement our empirical evaluation with a theoretical analysis, proving that our algorithm is sound and, under mildly restrictive conditions, also complete when using a suitable set of heuristics.

## 1. Introduction

As deep learning spreads through society, it becomes increasingly important to ensure the reliability of artificial neural networks, including aspects of fairness and safety. However, manually introspecting neural networks is infeasible due to their non-transparent nature. Furthermore, empirical assessments of neural networks are challenged by neural networks being fragile with respect to various types of input perturbations (Szegedy et al., 2014; Hosseini et al., 2017; Bibi et al., 2018; Ebrahimi et al., 2018; Hendrycks et al., 2021). In contrast, neural network verification analyses neural networks with mathematical rigour, facilitating the faithful auditing of neural networks.

In this paper, we consider probabilistic verification problems of neural networks, which are concerned with proving statements about the output distribution of a neural network given a distribution of the inputs. We refer to solving probabilistic verification problems as *probabilistic verification*. An example of probabilistic verification is proving that a neural network net making a binary decision affecting a person (for example, hire/do not hire, credit approved/denied) satisfies the demographic parity fairness notion (Barocas et al., 2023) under a probability distribution $\mathbb{P}_{\mathbf{x}}$ of the network inputs $\mathbf{x}$ representing the person

$$\frac{\mathbb{P}_{\mathbf{x}}[\text{net}(\mathbf{x}) = \texttt{yes} \mid \mathbf{x} \text{ is disadvantaged}]}{\mathbb{P}_{\mathbf{x}}[\text{net}(\mathbf{x}) = \texttt{yes} \mid \mathbf{x} \text{ is advantaged}]} \geq \gamma, \quad (1)$$

where '$\mathbf{x}$ is disadvantaged' could, for example, refer to a person not being male and $\gamma \in [0, 1]$ with $\gamma = 0.8$ being a common choice (Feldman et al., 2015). A closely related problem to probabilistic verification is computing bounds on probabilities over a neural network. An example of this is quantifying the safety of a neural network by bounding

$$\mathbb{P}_{\mathbf{x}}[\text{net}(\mathbf{x}) \text{ is unsafe}]. \quad (2)$$

In this paper, we introduce a novel algorithm for computing bounds on probabilities such as Equation (2) using a branch and bound framework (Land & Doig, 2010; Morrison et al., 2016; Bunel et al., 2020). These bounds allow us to verify probabilistic statements like Equation (1) using bound propagation (Moore et al., 2009; Albarghouthi et al., 2017). Our algorithm PROBABILISTICVERIFICATION (PV) is a fast and generally applicable probabilistic verification algorithm for neural networks based on massively parallel branch and bound neural network verification (Xu et al., 2021) using linear relaxations of neural networks (Weng et al., 2018; Zhang et al., 2018; Singh et al., 2019). Our theoretical analysis shows that PV is sound and, under mildly restrictive conditions, complete when using suitable branching and splitting heuristics.

---

[1]Department of Computer and Information Science, University of Konstanz, Konstanz, Baden-Würtemberg, Germany. Correspondence to: David Boetius <david.boetius@uni-konstanz.de>.

*Proceedings of the 42$^{nd}$ International Conference on Machine Learning*, Vancouver, Canada. PMLR 267, 2025. Copyright 2025 by the author(s).

Our experimental evaluation reveals that PV significantly outpaces the probabilistic verification algorithms FAIR-SQUARE (Albarghouthi et al., 2017) and SPACESCAN-NER (Converse et al., 2020). In particular, PV solves benchmark instances that FAIRSQUARE can not solve within 15 minutes in less than one minute and solves the ACAS Xu (Katz et al., 2017b) probabilistic robustness case study of Converse et al. (2020) in a mean runtime of 22 seconds, compared to 33 minutes for SPACESCANNER.

Applying PV to #DNN verification (Marzari et al., 2023a), a subset of probabilistic verification, reveals that PV also compares favourably to specialized algorithms, such as PROVE_SLR (Marzari et al., 2023b). It even compares favourably to $\varepsilon$-PROVE (Marzari et al., 2024) that relaxes #DNN verification to computing a confidence interval on the solution and PREIMGAPPROX (Zhang et al., 2024) that only computes sampling approximations. In contrast to this, PV computes lower and upper bounds on probabilities like Equation (2) that are guaranteed to hold with absolute certainty. Such bounds are preferable to confidence intervals in high-risk machine-learning applications, such as medical applications or autonomous driving and flight.

To test the limits of PV, we introduce a significantly more challenging probabilistic verification benchmark: The Mini-ACSIncome benchmark is based on the ACSIncome dataset (Ding et al., 2021) and is concerned with verifying the demographic parity of neural networks for datasets of increasing input dimensionality. MiniACSIncome provides more complex input distributions of higher dimensionality than earlier probabilistic verification benchmarks. Our main contributions are

- the PV algorithm for the probabilistic verification of neural networks,

- a theoretical analysis proving the soundness and completeness of PV,

- a thorough experimental comparison of PV with existing probabilistic verifiers for neural networks and tools dedicated to restricted subsets of probabilistic verification, and

- MiniACSIncome: a new, challenging probabilistic verification benchmark.

Our code is available at `https://github.com/sen-uni-kn/probspecs`. MiniACSIncome is available as a Python package at `https://pypi.org/project/miniacsincome/`.

## 2. Related Work

Approaches for non-probabilistic neural network verification include Satisfiability Modulo Theories (SMT) solv-

ing (Katz et al., 2017b), Mixed Integer Linear Programming (MILP) (Tjeng et al., 2019; Cheng et al., 2017), and reachability analysis (Bak et al., 2020; Tran et al., 2020a;b). Many of these approaches can be understood as branch and bound algorithms (Bunel et al., 2020). Branch and bound (Land & Doig, 2010; Morrison et al., 2016) also powers the $\alpha,\beta$-CROWN (Zhou et al., 2024) verifier that leads the table in recent international neural network verifier competitions (Brix et al., 2023; 2024). A critical component of a branch and bound verification algorithm is computing bounds on the output of a neural network. Approaches for bounding neural network outputs include interval arithmetic (Pulina & Tacchella, 2010), dual approaches (Wong & Kolter, 2018), and linear bound propagation techniques (Weng et al., 2018; Singh et al., 2019; Xu et al., 2021), such as CROWN (Zhang et al., 2018).

Probabilistic verification algorithms can be divided into *sound* algorithms that provide valid proofs, *probably sound* algorithms that provide valid proofs only with a certain predefined probability, and *unsound* algorithms that do not quantify their probability of providing invalid proofs. Probably sound algorithms provide similar guarantees as *probably approximately correct (PAC)* learning (Bishop, 2007). Fairness verification is a subset of probabilistic verification that studies problems such as Equation (1). Another subset of probabilistic verification is #DNN verification (Marzari et al., 2023a), corresponding to probabilistic verification under uniformly distributed inputs. Table 1 provides an overview of approaches for neural network verification.

By sacrificing soundness, probably sound verification algorithms (Bastani et al., 2019; Baluta et al., 2019; Weng et al., 2019; Marzari et al., 2023a; 2024) obtain efficiency. One example is $\varepsilon$-PROVE (Marzari et al., 2024), a probably sound #DNN verification algorithm. Bastani et al. (2019), Converse et al. (2020), and Marzari et al. (2023b) compare sound and probably sound approaches for probabilistic verification. We study sound algorithms since certainly valid results are preferable in high-risk applications.

Concerning sound approaches, FAIRSQUARE (Albarghouthi et al., 2017) is a sound fairness verification algorithm that partitions the input space into disjoint hyperrectangles and iteratively refines the input space partitioning using SMT solving. PROVE_SLR (Marzari et al., 2023b) is a sound #DNN verifier based on a massively parallel branch and bound algorithm. Converse et al. (2020) (SPACESCANNER) and Borca-Tasciuc et al. (2023) divide the input space into disjoint polytopes using concolic execution and reachable set verification, respectively. Both approaches are unsound for fairness verification due to approximating continuous probability distributions using histograms. SPACES-CANNER is sound for #DNN verification. PREIMGAP-PROX (Zhang et al., 2024) divides the input space into dis-

*Table 1.* Comparison of Related Approaches. Sound approaches provide definite guarantees. Probably sound approaches only provide PAC-type guarantees. Approaches marked 'unsound*' are sound in specific settings but neither sound nor probably sound in general.

| | Verification Problem | | | | |
|---|---|---|---|---|---|
| **Approach** | **Non-Probabilistic** | **#DNN** | **Group Fairness** | **General Probabilistic** | **Verifier Guarantee** |
| Zhou et al. (2024); Tran et al. (2020b) | ✓ | ✗ | ✗ | ✗ | sound |
| Weng et al. (2019) | ✓ | ✗ | ✗ | ✗ | probably sound |
| Borca-Tasciuc et al. (2023) | ✗ | ✓ | ✗ | ✗ | unsound |
| Zhang et al. (2024) | ✗ | ✓ | ✗ | ✗ | unsound* |
| Converse et al. (2020) | ✗ | ✓ | ✓ | ✓ | unsound* |
| Baluta et al. (2019); Marzari et al. (2023a; 2024) | ✗ | ✓ | ✗ | ✗ | probably sound |
| Bastani et al. (2019) | ✗ | ✗ | ✓ | ✗ | probably sound |
| Marzari et al. (2023b) | ✗ | ✓ | ✗ | ✗ | sound |
| Albarghouthi et al. (2017) | ✗ | ✗ | ✓ | ✗ | sound |
| Morettin et al. (2024) | ✗ | ✓ | ✓ | ✓ | sound |
| Ours (PV) | ✗ | ✓ | ✓ | ✓ | sound |

joint polytopes using ReLU branching (Bunel et al., 2020), but is unsound due to using sampling to approximate probabilities. However, it offers a post-verification soundness check for low-dimensional verification problems. Tran et al. (2023) provide a sound verifier based on reachability analysis that is only applicable for truncated Gaussian input distributions. Morettin et al. (2024) use weighted model integration (Belle et al., 2015) to obtain a general sound probabilistic verification algorithm but neither provide code nor report runtimes.

From the above approaches, FAIRSQUARE and PROVE_SLR are most closely related to our algorithm PV. However, PV is more general than FAIRSQUARE and PROVE_SLR, which are restricted to fairness verification and #DNN verification, respectively. Like FAIRSQUARE, PV iteratively refines bounds on probabilities to verify probabilistic statements like Equation (1). However, while FAIRSQUARE uses expensive SMT solving for refining the input space, we use computationally inexpensive input splitting and bound propagation techniques from non-probabilistic neural network verification. PROVE_SLR builds on similar techniques but computes probabilities exactly instead of iteratively refining bounds. These differences allow PV to significantly outpace both FAIRSQUARE and PROVE_SLR, as demonstrated in Section 6.

A related problem to probabilistic verification of neural networks is verifying Bayesian Neural Networks (BNNs) (Cardelli et al., 2019a;b; Wicker et al., 2020; 2024; Berrada et al., 2021; Adams et al., 2023; Batten et al., 2024). In BNNs, the network parameters follow a probability distribution (Neal, 1996). Since neural networks typically have vastly more parameters than inputs, BNN verification is concerned with

much higher dimensional probability distributions than we consider in this paper. Our restriction to deterministic neural networks allows us to provide a sound and complete yet practically scalable probabilistic verification algorithm.

Dependency fairness (Galhotra et al., 2017; Urban et al., 2020) is a non-probabilistic individual fairness notion (Dwork et al., 2012). Approaches for verifying dependency fairness include (Ruoss et al., 2020; Urban et al., 2020; Biswas & Rajan, 2023; Mohammadi et al., 2023; Kim et al., 2024). We are concerned with probabilistic fairness notions.

## 3. Preliminaries and Problem Statement

Throughout this paper, we are concerned with computing (provable) lower and upper bounds.

**Definition 3.1** (Bounds). For $f : \mathbb{R}^n \to \mathbb{R}^m$, we call $\ell, u \in \mathbb{R}^m$ a *lower*, respectively, *upper bound* on $f$ for $\mathcal{X}' \subseteq \mathbb{R}^n$ if $\ell \leq f(\mathbf{x}) \leq u$, for all $\mathbf{x} \in \mathcal{X}'$.

**Neural Networks.** In particular, we are concerned with computing bounds on neural networks $\text{net} : \mathcal{X} \to \mathbb{R}^m$, where $\mathcal{X} \subseteq \mathbb{R}^n$ is the input space of the neural network. A neural network is a composition of linear functions and a predefined set of non-linear functions, such as ReLU and max pooling. We only study Lipschitz continuous neural networks, which includes many common architectures (Szegedy et al., 2014; Ruan et al., 2018). Besides this, we refrain from defining neural networks further, as our approach is not specific to any particular architecture.

**Notation and Terminology.** We use $[\underline{\mathbf{x}}, \overline{\mathbf{x}}] = \{\mathbf{x} \in \mathbb{R}^n \mid \underline{\mathbf{x}} \leq \mathbf{x} \leq \overline{\mathbf{x}}\}$ to denote hyperrectangles and write $[n] = \{1, \ldots, n\}$ for $n \in \mathbb{N}$. The term *bounds* generally refers to a pair of a lower and an upper bound in this paper. We

assume that all random objects are defined on the same abstract probability space $(\Omega, \mathcal{F}, \mathbb{P})$ and that all continuous random variables admit a probability density function.

### 3.1. Probabilistic Verification of Neural Networks

Let $\mathcal{X} \subseteq \mathbb{R}^n$ be a (potentially unbounded) hyperrectangle and let $v \in \mathbb{N}$. We are concerned with proving or disproving whether a neural network $\mathsf{net} : \mathcal{X} \to \mathbb{R}^m$ is feasible for the *probabilistic verification problem*

$$\begin{cases} f_{\mathrm{Sat}}(p_1, \ldots, p_v) \geq 0, \\ p_i = \mathbb{P}_{\mathbf{x}^{(i)}}[g_{\mathrm{Sat}}^{(i)}(\mathbf{x}^{(i)}, \mathsf{net}(\mathbf{x}^{(i)})) \geq 0] \ \ \forall i \in [v], \end{cases} \quad (3)$$

where $\mathbf{x}^{(i)}$, $i \in [v]$, is an $\mathcal{X}$-valued random variable with distribution $\mathbb{P}_{\mathbf{x}^{(i)}}$ and $f_{\mathrm{Sat}} : \mathbb{R}^v \to \mathbb{R}$, $g_{\mathrm{Sat}}^{(i)} : \mathbb{R}^n \times \mathbb{R}^m \to \mathbb{R}$, $i \in [v]$ are *satisfaction functions* that are compositions of linear functions, multiplication, division, and monotone functions. Appendix A expresses Equation (1) in the form of Equation (3) as a concrete example.

Probabilistic verification with a single uniformly distributed random variable corresponds to #DNN verification (Marzari et al., 2023a). As Marzari et al. (2023a) prove, #DNN verification is #P complete, implying that probabilistic verification is #P hard. However, this does not determine which probabilistic verification problems are practically solvable.

### 3.2. Non-Probabilistic Neural Network Verification

Non-probabilistic neural network verification determines whether $\mathsf{net} : \mathcal{X} \to \mathbb{R}^m$ is feasible for

$$g_{\mathrm{Sat}}(\mathbf{x}, \mathsf{net}(\mathbf{x})) \geq 0 \quad \forall \mathbf{x} \in \mathcal{X}', \quad (4)$$

where $\mathcal{X}' \subseteq \mathcal{X}$ is a bounded hyperrectangle, and $g_{\mathrm{Sat}} : \mathbb{R}^n \times \mathbb{R}^m \to \mathbb{R}$ is a *satisfaction function* that indicates whether the output of $\mathsf{net}$ is desirable ($g_{\mathrm{Sat}}(\cdot, \mathsf{net}(\cdot)) \geq 0$) or undesirable ($g_{\mathrm{Sat}}(\cdot, \mathsf{net}(\cdot)) < 0$). In neural network verification, $g_{\mathrm{Sat}}$ can generally be considered a part of $\mathsf{net}$ (Bunel et al., 2020; Xu et al., 2020). *Neural network verifiers* are algorithms for proving or disproving Equation (4). Two desirable properties of neural network verifiers are *soundness* and *completeness*.

**Definition 3.2** (Soundness and Completeness). A verification algorithm is *sound* if it only produces genuine counterexamples and valid proofs for Equation (4). It is *complete* if it produces a counterexample or proof for Equation (4) for any neural network in a finite amount of time.

Analogous notions of soundness and completeness also apply to probabilistic verification. Section 2 discusses *probably sound* verification.

**Interval Arithmetic.** Interval arithmetic (Moore et al., 2009) is a bound propagation technique that derives bounds

on the output of a neural network from bounds on the network input. Assume $\underline{\mathbf{x}} \leq \mathbf{x} \leq \overline{\mathbf{x}}$ are bounds on the network input $\mathbf{x}$. We apply interval arithmetic to compute $\ell \leq g_{\mathrm{Sat}}(\mathbf{x}, \mathsf{net}(\mathbf{x})) \leq u$, $\forall \mathbf{x} \in [\underline{\mathbf{x}}, \overline{\mathbf{x}}]$. If the lower bound is large enough or the upper bound small enough, we can prove or disprove Equation (4) using $\ell \geq 0 \implies g_{\mathrm{Sat}}(\mathbf{x}, \mathsf{net}(\mathbf{x})) \geq 0$, respectively, $u < 0 \implies g_{\mathrm{Sat}}(\mathbf{x}, \mathsf{net}(\mathbf{x})) < 0$. However, if the bounds are *inconclusive*, that is $\ell < 0 \leq u$, we can neither prove nor disprove Equation (4). Therefore, interval arithmetic is incomplete according to Definition 3.2.

Let $f : \mathbb{R}^n \to \mathbb{R}$ be a function that we want to bound for inputs $\mathbf{x} \in [\underline{\mathbf{x}}, \overline{\mathbf{x}}]$. Interval arithmetic and other bound propagation techniques rely on $f = f^{(K)} \circ \cdots \circ f^{(1)}$ being a composition of more fundamental functions $f^{(k)} : \mathbb{R}^{n_k} \to \mathbb{R}^{n_{k+1}}$ for which we can already compute $\ell^{(k)} \leq f^{(k)}(\mathbf{z}) \leq u^{(k)}$ given $\underline{\mathbf{z}} \leq \mathbf{z} \leq \overline{\mathbf{z}}$. Examples of such functions include monotone non-decreasing functions, for which $f^{(k)}(\underline{\mathbf{z}}) \leq f^{(k)}(\mathbf{z}) \leq f^{(k)}(\overline{\mathbf{z}})$. This *bounding rule* already allows us to bound addition, $\min$, $\max$, and ReLU, among others. Appendix D contains further bounding rules. Given a bounding rule $F^{(k)} : \mathbb{R}^{n_k} \times \mathbb{R}^{n_k} \to \mathbb{R}^{n_{k+1}} \times \mathbb{R}^{n_{k+1}}$ for each $f^{(k)}$, interval arithmetic computes bounds on $f$ by computing $(F^{(K)} \circ \cdots \circ F^{(1)})(\underline{\mathbf{x}}, \overline{\mathbf{x}})$.

**Branch and Bound.** Bound propagation approaches, such as interval arithmetic and CROWN (Zhang et al., 2018), are incomplete according to Definition 3.2. To obtain a complete verifier, bound propagation can be combined with *branching* to gain completeness. This algorithmic framework is called *branch and bound* (Land & Doig, 2010; Morrison et al., 2016). In branch and bound, the search space is split (*branching*) when the computed bounds are inconclusive ($\ell < 0 \leq u$). The idea is that splitting improves the precision of the bounds for each part of the split (each *branch*). Bunel et al. (2020) provide an introduction to branch and bound for non-probabilistic neural network verification. The next section introduces our branch and bound algorithm for probabilistic neural network verification.

## 4. Algorithm

This section introduces PROBABILISTICVERIFICATION (PV), our algorithm for probabilistic verification of neural networks as defined in Equation (3). The overall approach of PV is to iteratively refine bounds on each $p_i$ from Equation (3) until a bound propagation approach allows us to prove or disprove $f_{\mathrm{Sat}}(p_1, \ldots, p_v) \geq 0$. For computing the bounds on $p_i$, PV partitions the input space into hyperrectangles since this allows for computing probabilities efficiently (Albarghouthi et al., 2017). To compute tighter bounds on $p_i$, PV uses a branch and bound algorithm that uses computationally inexpensive input splitting and bound propagation techniques from non-probabilistic neural net-

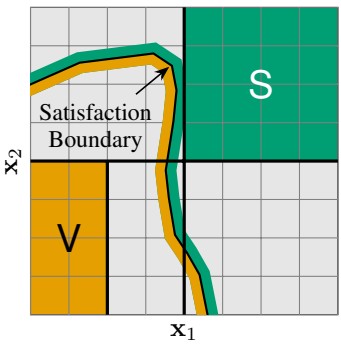

(a) After four Splits.

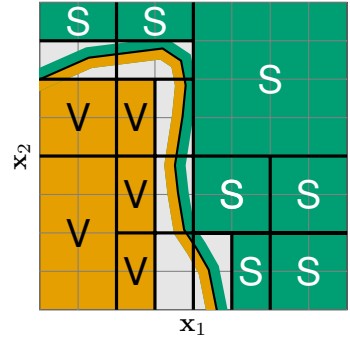

(b) After 17 Splits.

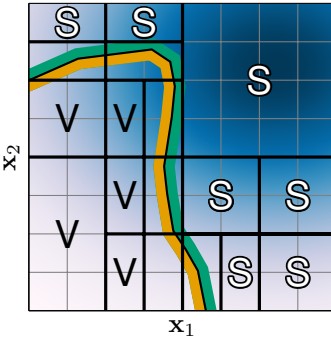

(c) Probability Density Function.

*Figure 1.* Computing bounds on probabilities. This figure illustrates the steps for computing bounds on $p = \mathbb{P}_{\mathbf{x}}[g_{\text{Sat}}(\mathbf{x}, \text{net}(\mathbf{x})) \geq 0]$. Our algorithm successively splits the input space to find regions that do not intersect the satisfaction boundary $g_{\text{Sat}}(\mathbf{x}, \text{net}(\mathbf{x})) \geq 0$ (orange/green line ▬). Green, orange, and grey rectangles (▬/▬/□) denote regions for which we could prove $g_{\text{Sat}}(\mathbf{x}, \text{net}(\mathbf{x})) \geq 0$ (satisfaction) ▬, $g_{\text{Sat}}(\mathbf{x}, \text{net}(\mathbf{x})) < 0$ (violation) ▬, or neither □, respectively. By integrating the probability density $f_{\mathbf{x}}$ in (c) (darker means higher density) over the green rectangles ▬, we obtain a lower bound on $p$. Similarly, we can integrate over the orange rectangles ▬ to construct an upper bound on $p$. Refining the input splitting from (a) to (b) tightens the bounds on $p$.

work verification for refining the input splitting.

Algorithm 1 describes the PV algorithm. The centrepiece of PV is the procedure PROBBOUNDS for computing bounds $\ell_i^{(t)} \leq p_i \leq u_i^{(t)}$ on a probability $p_i$ from Equation (3). Given $\ell_i^{(t)} \leq p_i \leq u_i^{(t)}$, we apply a bound propagation technique to prove or disprove $f_{\text{Sat}}(p_1, \ldots, p_v)$, as described in Section 3.2. If this analysis is inconclusive, PROBBOUNDS refines $\ell_i^{(t)}, u_i^{(t)}$ to obtain $\ell_i^{(t+1)}, u_i^{(t+1)}$ with $\ell_i^{(t)} \leq \ell_i^{(t+1)} \leq p_i \leq u_i^{(t+1)} \leq u_i^{(t)}$. We again apply bound propagation to $f_{\text{Sat}}(p_1, \ldots, p_v)$, this time using $\ell_i^{(t+1)}, u_i^{(t+1)}$. If the result remains inconclusive, we iterate refining the bounds on each $p_i$ until we obtain a conclusive result. PV applies PROBBOUNDS for each $p_i$ in parallel, making use of several CPU cores or several GPUs. Our main contribution is the PROBBOUNDS algorithm for computing a converging sequence of lower and upper bounds on $p_i$.

---

**Algorithm 1** PV

**Require:** Probabilistic Verification Problem as in Equation (3), Batch Size $N$
1: **for** $i \in [v]$ **do** *// Launch $v$ parallel instances*
2:    $\text{PB}_i \leftarrow$ Launch PROBBOUNDS$(p_i, N)$
3: **end for**
4: **for** $t \in \mathbb{N}$ **do**
5:    **for** $i \in [v]$ **do** Gather $b_i^{(t)} = (\ell_i^{(t)}, u_i^{(t)})$ from $\text{PB}_i$
6:    $(\ell^{(t)}, u^{(t)}) \leftarrow$ COMPUTEBOUNDS$(f_{\text{Sat}}, b_1^{(t)}, \ldots, b_v^{(t)})$
7:    **if** $\ell^{(t)} \geq 0$ **then return** Satisfied
8:    **if** $u^{(t)} < 0$ **then return** Violated
9: **end for**

---

**Algorithm 2** PROBBOUNDS

**Require:** Probability $\mathbb{P}_{\mathbf{x}}[g_{\text{Sat}}(\mathbf{x}, \text{net}(\mathbf{x})) \geq 0]$, Batch Size $N$
1: branches $\leftarrow \{\mathcal{X}\}$
2: $\ell^{(0)} \leftarrow 0, u^{(0)} \leftarrow 1$
3: **for** $t \in \mathbb{N}$ **do**
4:    batch $\leftarrow$ SELECT(branches, $N$)
5:    $(\underline{\mathbf{y}}, \overline{\mathbf{y}}) \leftarrow$ COMPUTEBOUNDS$(g_{\text{Sat}}(\cdot, \text{net}(\cdot)), \text{batch})$
6:    $(\text{keep}, \mathcal{X}_{sat}^{(t)}, \mathcal{X}_{viol}^{(t)}) \leftarrow$ PRUNE(batch, $\underline{\mathbf{y}}, \overline{\mathbf{y}}$)
7:    $\ell^{(t)} \leftarrow \ell^{(t-1)} + \mathbb{P}_{\mathbf{x}}[\mathcal{X}_{sat}^{(t)}]$
8:    $u^{(t)} \leftarrow u^{(t-1)} - \mathbb{P}_{\mathbf{x}}[\mathcal{X}_{viol}^{(t)}]$
9:    **yield** $(\ell^{(t)}, u^{(t)})$ *// Report new bounds to PV*
10:   new $\leftarrow$ SPLIT(keep)
11:   branches $\leftarrow$ (branches \ batch) $\cup$ new
12: **end for**

---

### 4.1. Bounding Probabilities

Our PROBBOUNDS algorithm for deriving and refining bounds on a probability is described in detail in Algorithm 2 and illustrated in Figure 1. PROBBOUNDS is a massively parallel input-splitting branch and bound algorithm (Bunel et al., 2020; Wang et al., 2018; Xu et al., 2020) that leverages a bound propagation algorithm for non-probabilistic neural network verification (COMPUTEBOUNDS). Since we only consider a single probability in this section, we denote this probability as $p = \mathbb{P}_{\mathbf{x}}[g_{\text{Sat}}(\mathbf{x}, \text{net}(\mathbf{x})) \geq 0]$.

PROBBOUNDS receives $p$ and a *batch size* $N \in \mathbb{N}$ as input. The algorithm iteratively computes $\ell^{(t)}, u^{(t)} \in [0, 1]$, such that $\ell^{(t)} \leq \ell^{(t')} \leq p \leq u^{(t')} \leq u^{(t)}, \forall t, t' \in \mathbb{N}, t' \geq t$. The following sections describe each step of PROBBOUNDS in detail.

**Initialisation.** Initially, we consider a single branch encompassing net's entire input space $\mathcal{X}$. As in Section 3, we assume $\mathcal{X}$ to be a (potentially unbounded) hyperrectangle. We use the trivial bounds $\ell^{(0)} = 0 \leq p \leq 1 = u^{(0)}$ as initial bounds on $p$.

**Selecting Branches.** First, we select a batch of $N \in \mathbb{N}$ branches. In the spirit of Xu et al. (2021), we leverage the data parallelism of modern CPUs and GPUs to process several branches at once. In iteration $t = 1$, the batch only contains the branch $\mathcal{X}$. Which branches we select determines how fast we obtain tight bounds on $p$. We propose the SELECTPROB heuristic for selecting branches. Inspired by FAIRSQUARE (Albarghouthi et al., 2017), SELECTPROB selects the $N$ branches $\mathcal{B}_i$ with the largest $\mathbb{P}_{\mathbf{x}}[\mathcal{B}_i]$. This heuristic is motivated by the observation that pruning these branches would lead to the largest improvement of $\ell^{(t)}, u^{(t)}$.

**Pruning.** The next step is to prune those branches $\mathcal{B}_j \in$ batch, for which we can determine that $y = g_{\text{Sat}}(\mathbf{x}, \text{net}(\mathbf{x})) \geq 0$ is either certainly satisfied or certainly violated. For this, we first compute $\underline{\mathbf{y}} \leq g_{\text{Sat}}(\cdot, \text{net}(\cdot)) \leq \overline{\mathbf{y}}$ for the entire batch using a bound propagation algorithm for neural networks, such as CROWN (Zhang et al., 2018). If $\underline{\mathbf{y}}_j \geq 0$ ($\mathbf{y}_j \geq 0$ is certainly satisfied) or $\overline{\mathbf{y}}_j < 0$ ($\mathbf{y}_j \geq 0$ is certainly violated), we can prune $\mathcal{B}_j$, meaning that we remove it from branches. We collect the branches with $\underline{\mathbf{y}}_j \geq 0$ in the set $\mathcal{X}_{sat}^{(t)}$ and the branches with $\overline{\mathbf{y}}_j < 0$ in the set $\mathcal{X}_{viol}^{(t)}$, where $t \in \mathbb{N}$ is the current iteration.

**Updating Bounds.** Let $\hat{\mathcal{X}}_{sat}^{(t)} = \bigcup_{t'=1}^{t} \mathcal{X}_{sat}^{(t')}$ and $\hat{\mathcal{X}}_{viol}^{(t)} = \bigcup_{t'=1}^{t} \mathcal{X}_{viol}^{(t')}$, where $t \in \mathbb{N}$ is the current iteration. Then, $\ell^{(t)} = \mathbb{P}_{\mathbf{x}}[\hat{\mathcal{X}}_{sat}^{(t)}] \leq p$. Similarly, $\hbar^{(t)} = \mathbb{P}_{\mathbf{x}}[\hat{\mathcal{X}}_{viol}^{(t)}] \leq \mathbb{P}_{\mathbf{x}}[g_{\text{Sat}}(\mathbf{x}, \text{net}(\mathbf{x})) < 0] = 1 - p$. Therefore, $1 - \hbar^{(t)} = u^{(t)} \geq p$. Practically, we only have to maintain the current bounds $\ell^{(t)}$ and $u^{(t)}$ instead of the sets $\hat{\mathcal{X}}_{sat}^{(t)}$ and $\hat{\mathcal{X}}_{viol}^{(t)}$.

Because $\hat{\mathcal{X}}_{sat}^{(t)}$ and $\hat{\mathcal{X}}_{viol}^{(t)}$ are a union of disjoint hyperrectangles, exactly computing $\mathbb{P}_{\mathbf{x}}[\hat{\mathcal{X}}_{sat}^{(t)}]$ and $\mathbb{P}_{\mathbf{x}}[\hat{\mathcal{X}}_{viol}^{(t)}]$ is feasible for a large class of probability distributions, including most univariate distributions, Mixture Models, and Bayesian Networks. The precise class of supported probability distributions is discussed in Section 4.2. While we do not account for floating point errors in this paper, our approach can readily be extended to this end.

**Splitting.** Splitting refines a branch $\mathcal{B} = [\underline{\mathbf{x}}, \overline{\mathbf{x}}]$ by selecting a dimension $d \in [n]$ to split. We split based on the type of variable that is encoded in $d$. For bounded continuous variables, we split by bisecting $[\underline{\mathbf{x}}, \overline{\mathbf{x}}]$ along $d$. For unbounded variables, we split at zero if $-\underline{\mathbf{x}}_d = \overline{\mathbf{x}}_d = \infty$, at $\max(2\underline{\mathbf{x}}_d, 1)$ if $-\underline{\mathbf{x}}_d < \overline{\mathbf{x}}_d = \infty$, and at $\min(2\overline{\mathbf{x}}_d, -1)$ if $-\infty = \underline{\mathbf{x}}_d < -\overline{\mathbf{x}}_d$. For integer variables, we additionally round the split points to the next smaller, respectively, larger integer. For dimensions containing a binary indicator of a

one-hot encoded categorical variable $A$, we jointly split all indicators of $A$ such that $A$ takes on the value encoded in $d$ in one branch and does not take on this value in the other branch. Appendix B.1 defines these splitting rules formally.

**Split Selection.** We present three heuristics for selecting the dimension $d$ for splitting. We generally select dimensions encoding unbounded variables first in order to obtain bounded branches, since COMPUTEBOUNDS usually computes vacuous bounds for unbounded branches. For bounded variables, the well-known LONGESTEDGE heuristic (Bunel et al., 2020) selects the dimension with the largest *edge length* $\overline{\mathbf{x}}_d - \underline{\mathbf{x}}_d$. Alternatively, we use a variant of the BABSB heuristic (Bunel et al., 2020). BABSB estimates the improvement in bounds that splitting dimension $d$ yields by using a yet less expensive technique than COMPUTEBOUNDS. Our variant of BABSB uses INTERVALARITHMETIC, assuming that we use CROWN for COMPUTEBOUNDS. Appendix B.2 describes our BABSB variant in detail. While LONGESTEDGE is more theoretically accessible, BABSB is practically advantageous, as discussed in Appendix G. Combining the advantages of both approaches, we introduce BABSB-LONGESTEDGE-k. This heuristic alternates using BABSB and LONGESTEDGE, using LONGESTEDGE for every k-th split. If we visualise branches and their descendants from splitting in a branching tree, the splits at level $k, 2k, 3k, \ldots$ use LONGESTEDGE while the splits at all other levels use BABSB.

### 4.2. Input Spaces and Input Distributions

This section discusses the concrete requirements that the input space $\mathcal{X}$ and the input distributions $\mathbb{P}_{\mathbf{x}^{(i)}}$ in Equation (3) need to satisfy for applying PV. Appendix A.5 discusses how some of these requirements can be mitigated, for example, for using polytopes as input spaces.

PV requires $\mathcal{X} \subseteq \mathbb{R}^n$ to be a hyperrectangle, which can be unbounded. The dimensions of $\mathcal{X}$ may encode discrete random variables. For each probability distribution $\mathbb{P}_{\mathbf{x}^{(i)}}$, we require a terminating algorithm that computes the exact probability of a hyperrectangle. This requirement is satisfied by a large class of probability distributions, including discrete distributions with a closed-form probability mass function and univariate continuous distributions with a closed-form cumulative density function, as well as Mixture Models and probabilistic graphical models (Bishop, 2007), such as Bayesian Networks, of such distributions.

## 5. Theoretical Analysis

In this section, we prove that PV is a sound probabilistic verification algorithm when instantiated with a suitable COMPUTEBOUNDS procedure. We also prove that PV is complete under mild assumptions on the probabilistic

verification problem when instantiated with suitable SPLIT, COMPUTEBOUNDS, and SELECT procedures. Soundness and completeness are defined in Definition 3.2. As in Section 4.1, we omit indices and superscripts when considering only a single probability $p = \mathbb{P}_{\mathbf{x}}[g_{\text{Sat}}(\mathbf{x}, \text{net}(\mathbf{x})) \geq 0]$. We defer all proofs to Appendix C. Our first result concerns the soundness of PROBBOUNDS.

**Theorem 5.1** (Sound Bounds). *Let $N \in \mathbb{N}$ be a batch size and assume COMPUTEBOUNDS produces valid bounds. Let $\{(\ell^{(t)}, u^{(t)})\}_{t \in \mathbb{N}}$ be the iterates of $\text{PROBBOUNDS}(\mathbb{P}_{\mathbf{x}}[g_{\text{Sat}}(\mathbf{x}, \text{net}(\mathbf{x})) \geq 0], N)$. It holds that $\ell^{(t)} \leq \mathbb{P}_{\mathbf{x}}[g_{\text{Sat}}(\mathbf{x}, \text{net}(\mathbf{x})) \geq 0] \leq u^{(t)}$ for all $t \in \mathbb{N}$.*

**Corollary 5.2** (Soundness). *PV is sound when using COMPUTEBOUNDS procedures that compute valid bounds.*

Our remaining theoretical results are concerned with the completeness of PV. Concretely, we prove that PV instantiated with SELECTPROB, LONGESTEDGE or BABSB-LONGESTEDGE-k, and INTERVALARITHMETIC or CROWN is complete under a mildly restrictive condition on Equation (3). Appendix C.2 defines a more general class of pruning and splitting heuristics for which PV is complete.

**Assumption 5.3.** *Let $v$, $f_{\text{Sat}}$, $g_{\text{Sat}}^{(i)}$, and $\mathbf{x}^{(i)}$ be as in Equation (3). Assume $f_{\text{Sat}}(p_1, \ldots, p_v) \neq 0$ and $\forall i \in [v]$ : $\mathbb{P}_{\mathbf{x}^{(i)}}[g_{\text{Sat}}^{(i)}(\mathbf{x}^{(i)}, \text{net}(\mathbf{x}^{(i)})) = 0] = 0$.*

Assumption 5.3 is only mildly restrictive, since for every verification problem that does not satisfy Assumption 5.3, there are similar problems that satisfy the assumption. Consider the case that $f_{\text{Sat}}(p_1, \ldots, p_n) = 0$. In this case, Equation (3) does not satisfy Assumption 5.3. However, a slightly stronger verification problem concerned with $f'_{\text{Sat}}(p_1, \ldots, p_n) = f_{\text{Sat}}(p_1, \ldots, p_n) - \varepsilon$ for an arbitrarily small $\varepsilon > 0$ satisfies Assumption 5.3. Appendix C.2 discusses Assumption 5.3 in more detail.

To prove the completeness of PV, we first establish that PROBBOUNDS produces a sequence of lower and upper bounds that converge towards each other.

**Lemma 5.4** (Converging Probability Bounds). *Let $N \in \mathbb{N}$ be a batch size. Let $\{(\ell^{(t)}, u^{(t)})\}_{t \in \mathbb{N}}$ be the iterates of $\text{PROBBOUNDS}(\mathbb{P}_{\mathbf{x}}[g_{\text{Sat}}(\mathbf{x}, \text{net}(\mathbf{x})) \geq 0], N)$ instantiated with SELECTPROB, LONGESTEDGE or BABSB-LONGESTEDGE-k, and INTERVALARITHMETIC or CROWN. Assume $\mathbb{P}_{\mathbf{x}}[g_{\text{Sat}}(\mathbf{x}, \text{net}(\mathbf{x})) = 0] = 0$ as in Assumption 5.3. Then,*

$$\lim_{t \to \infty} \ell^{(t)} = \lim_{t \to \infty} u^{(t)} = \mathbb{P}_{\mathbf{x}}[g_{\text{Sat}}(\mathbf{x}, \text{net}(\mathbf{x})) \geq 0].$$

**Theorem 5.5** (Completeness). *When instantiated with PROBBOUNDS as in Lemma 5.4 and INTERVALARITHMETIC or CROWN for COMPUTEBOUNDS, PV is complete for verification problems satisfying Assumption 5.3.*

Unfortunately, our completeness result does not apply to the BABSB heuristic, which provides the best empirical performance when used in PV. However, our result applies to BABSB-LONGESTEDGE-k, which yields comparable performance as BABSB, as we show in Appendix G.1.

# 6. Experiments

We apply our algorithms to verify the demographic parity fairness notion, count the number of safety violations of neural network controllers in safety-critical systems, and quantify the robustness of a neural network. Table 2 gives an overview of our benchmarks. All verification problems are defined formally in Appendix A. For all benchmarks, PROBBOUNDS use the SELECTPROB and BABSB heuristics and CROWN (Zhang et al., 2018) for COMPUTEBOUNDS, while PV uses INTERVALARITHMETIC.

As our results show, PV (Algorithm 1) outpaces the probabilistic verification algorithms FAIRSQUARE (Albarghouthi et al., 2017) and SPACESCANNER (Converse et al., 2020). Additionally, we show that PROBBOUNDS (Algorithm 2) compares favourably to the PROVE_SLR (Marzari et al., 2023b), $\varepsilon$-PROVE (Marzari et al., 2024), and PREIMGAPPROX (Zhang et al., 2024) algorithms for #DNN verification (Marzari et al., 2023a), which corresponds to probabilistic verification with uniformly distributed inputs.

While no code is publicly available for SPACESCANNER, running PROVE_SLR is very computationally expensive. To enable a faithful comparison, we run our experiments on less powerful hardware (HW1) compared to the hardware used by Converse et al. (2020) and Marzari et al. (2023b) and compare the runtime of our algorithms to the runtimes reported by these authors. All other results reported in this paper were obtained on HW1, including the results for FAIRSQUARE, $\varepsilon$-PROVE, and PREIMGAPPROX.

To test the limits of PV, we introduce a new, challenging benchmark: MiniACSIncome is based on the ACSIncome dataset (Ding et al., 2021). It consists of datasets of varying input dimensionality, probability distributions for these datasets, and neural networks trained on these datasets. Being based on real-world US census data, MiniACSIncome offers more complex input distributions with higher input dimensionality than existing probabilistic verification benchmarks. PV solves seven of eight instances in MiniACSIncome within an hour.

**Hardware and Implementation.** We implement PV in Python, leveraging PyTorch (Paszke et al., 2019) and auto_LiRPA (Xu et al., 2020). We run all experiments on a Ubuntu 22.04 desktop with an Intel i7–4820K CPU, 32 GB of memory, and no GPU (HW1). Appendix F.1 compares our hardware to the hardware used by Converse et al. (2020) and Marzari et al. (2023b).

*Table 2.* Our benchmarks. Network size is the size of the neural network given as #layers×layer size.

| Benchmark | Input Dimension | Input Distributions | Network Size | Source |
|---|---|---|---|---|
| **FairSquare** | 2–3 | independent 2 Bayesian Networks | $1\times1, 1\times2$ | (Albarghouthi et al., 2017) |
| **ACAS Xu** | 5 | uniform | $6\times50$ | (Katz et al., 2017b) |
| **VCAS** | 4 | uniform | $1\times21$ | (Zhang et al., 2024) |
| **MiniACSIncome** | 1–8 | Bayesian Network | $1\times10$–10000 $1\times10 - 10\times10$ | Own |

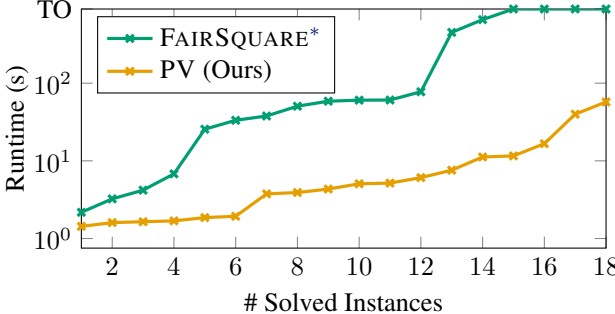

*Figure 2.* FairSquare benchmark results. The timeout (TO) is 15min. *Albarghouthi et al. (2017)

## 6.1. FairSquare Benchmark

Albarghouthi et al. (2017) evaluate their FAIRSQUARE algorithm on an application derived from the Adult dataset (Adult, 1996). In particular, they verify whether three small neural networks satisfy two fairness notions with respect to a person's sex under three different distributions of the network input: a distribution of entirely independent univariate variables and two Bayesian Networks. Appendix F.2 describes the FairSquare benchmark in more detail.

Figure 2 compares the runtimes of PV and FAIRSQUARE on the FairSquare benchmark. PV significantly outperforms FAIRSQUARE. In particular, PV solves four more instances than FAIRSQUARE within the timeout of 15 minutes. For the instances that both tools solve, the median runtime of PV is 4s (mean: 5s, max: 17s) compared to 44s for FAIR-SQUARE (mean: 109s, max: 657s). Appendix F.2 contains the detailed results of this experiment.

## 6.2. Aircraft Collision Avoidance

The ACAS Xu networks (Katz et al., 2017b) are a suite of 45 networks, together forming a collision avoidance system for crewless aircraft. Each ACAS Xu network predicts a horizontal turning direction to avoid collision with another aircraft. VCAS (Julian & Kochenderfer, 2019) is a similar system that predicts vertical steering directions for avoiding collisions. We reproduce the ACAS Xu safety experiments of Marzari et al. (2023b), the ACAS Xu robustness experiments of Converse et al. (2020), and the VCAS correctness experiment of Zhang et al. (2024).

**ACAS Xu Safety.** In this experiment, we seek to *quantify* the number of violations (violation rate) of several ACAS Xu networks (Katz et al., 2017b). This corresponds to computing bounds on Equation (2) under a uniform distribution of **x**. We compare PROBBOUNDS to the PROVE_SLR and $\varepsilon$-PROVE algorithms for #DNN verification. PROVE_SLR computes the violation rate exactly, while $\varepsilon$-PROVE computes an upper bound on the violation rate that is sound with a certain predefined probability. In contrast, PROBBOUNDS provides sound bounds on the violation rate at any time during its execution.

Table 3 compares PROBBOUNDS to PROVE_SLR and $\varepsilon$-PROVE for the ACAS Xu networks investigated by Marzari et al. (2023b). For all three networks, PROBBOUNDS can tighten the bounds to a margin of less than 0.7% within one hour, while PROVE_SLR requires at least four hours to compute the exact violation rate. In comparison to $\varepsilon$-PROVE, PROBBOUNDS produces tighter sound bounds within 10 seconds in two of three cases, while $\varepsilon$-PROVE requires at least 57 seconds to derive a probably sound upper bound for these cases. The extended comparison in Appendix F.3 reveals that in 12 from a total of 36 cases, PROBBOUNDS computes a tighter sound bound faster than $\varepsilon$-PROVE computes a probably sound upper bound.

**ACAS Xu Robustness.** We replicate the experiments of Converse et al. (2020) who apply SPACESCANNER to quantify the robustness of ACAS Xu network $N_{1,1}$ (Katz et al., 2017b) under adversarial perturbations. Overall, the experiment consists of 125 verification problems that concern the probability of obtaining a particular class for uniformly distributed perturbed inputs close to one of 25 reference input points.

The mean runtime of PROBBOUNDS for these 125 instances is 22 seconds (median: 6s, maximum: 213s). In contrast, Converse et al. (2020) report a mean runtime of 33 minutes per instance for SPACESCANNER while running their exper-

*Table 3.* Comparison of PROBBOUNDS, PROVE_SLR, and $\varepsilon$-PROVE. We run PROBBOUNDS with different time budgets (10s, 1m, 1h) and report the lower and upper bounds $(\ell, u)$ computed within this time budget. In contrast, PROVE_SLR computes the exact probabilities (VR), and $\varepsilon$-PROVE computes a 99.9% confidence (confid.) upper bound. The probabilities and probability bounds are given as percentages. The runtimes (Rt) of PROVE_SLR are taken from Marzari et al. (2023b).

| | PROBBOUNDS (Ours) | | | PROVE_SLR[†] | | $\varepsilon$-PROVE[‡] | |
| | **10s** | **1m** | **1h** | **Exact** | | **99.9% confid.** | |
| net | $\ell, u$ | $\ell, u$ | $\ell, u$ | VR | Rt | $u$ | Rt |
|---|---|---|---|---|---|---|---|
| $N_{4,3}$ | $0.17\%, 2.92\%$ | $0.61\%, 2.27\%$ | $1.12\%, 1.75\%$ | $1.43\%$ | 8h 46m | $3.61\%$ | 65s |
| $N_{4,9}$ | $0.00\%, 3.36\%$ | $0.00\%, 1.55\%$ | $0.08\%, 0.29\%$ | $0.15\%$ | 12h 21m | $0.73\%$ | 20s |
| $N_{5,8}$ | $0.89\%, 4.16\%$ | $1.55\%, 3.10\%$ | $1.97\%, 2.57\%$ | $2.20\%$ | 4h 35m | $4.52\%$ | 57s |

[†]Marzari et al. (2023b)    [‡]Marzari et al. (2024)

iments on superior hardware. Appendix F.4 contains more details on this experiment.

**VCAS Correctness.** Zhang et al. (2024) study whether a VCAS network correctly predicts to maintain course in a scenario where there is no risk of collision. Concretely, they verify whether the VCAS network provides the correct output at least 90% of the time. PV is able to prove this within 0.13s. In contrast, PREIMGAPPROX requires 16.42s for computing an unsound empirical lower bound on the probability of obtaining correct outputs.

### 6.3. MiniACSIncome

To test the limits of PV, we introduce the MiniACSIncome benchmark. MiniACSIncome is derived from the ACSIncome dataset (Ding et al., 2021), a replacement of the Adult dataset (Adult, 1996) that is better suited for fair machine learning research. The task is to predict whether a person's yearly income exceeds $50 000 using features such as the person's age, sex, and education. Our benchmark provides probabilistic verification problems of various degrees of difficulty. We apply PV to MiniACSIncome and compare it to a baseline approach for solving MiniACSIncome.

**Benchmark.** To create probabilistic verification problems of increasing difficulty, we consider an increasing number of input variables from ACSIncome. The smallest instance, MiniACSIncome-1, only contains the binary 'SEX' variable. In contrast, the largest instance, MiniACSIncome-8, contains 'SEX' and seven more variables from ACSIncome, including age, education, and working hours per week. Our benchmark's task is to verify the demographic parity of neural networks with varying input dimension under a Bayesian Network as input distribution. These Bayesian Networks provide complex multi-modal input distributions, as they fit the real-world US census data in ACSIncome. Appendix F.5 describes MiniACSIncome in detail.

**Results.** Since all variables in MiniACSIncome are discrete,

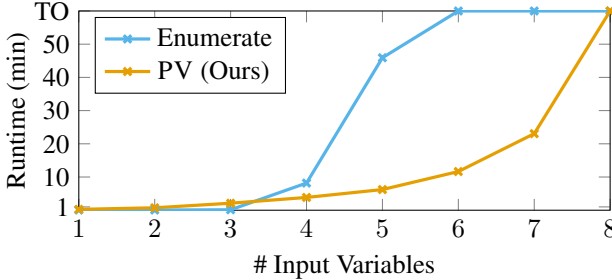

*Figure 3.* MiniACSIncome results. The timeout (TO) is one hour.

a baseline approach for verifying the demographic parity of a MiniACSIncome network is to enumerate all values in the input space. Figure 3 displays the runtime of PV and the baseline enumeration approach for shallow 10-neuron neural networks with increasing input size. While enumeration is faster than PV when the network can be evaluated for all discrete values in one batch, enumeration falls behind PV as soon as this becomes infeasible. PV can solve MiniACSIncome for up to seven input variables in less than 30 minutes, only exceeding the timeout of one hour for eight input variables. While we only consider a small network here, the runtime of PV is largely unaffected by network size on this benchmark. This unexpected result can be attributed to both large and small networks learning similar decision boundaries for MiniACSIncome. Appendix F.5.5 discusses this result in more detail.

## 7. Conclusion

Our PV algorithm for the probabilistic verification of neural networks significantly outpaces existing algorithms for probabilistic verification. We achieve this speedup by applying a massively parallel branch and bound algorithm based on bound propagation algorithms for neural networks. Our MiniACSIncome benchmark provides a challenging testbed for future probabilistic verification algorithms.

## Impact Statement

This work is concerned with providing mathematical guarantees on the output distribution of a neural network given a distribution of the inputs. Since mathematical guarantees enhance the transparency of neural networks and facilitate their faithful auditing, we anticipate that our work will have a predominantly positive societal impact. However, obtaining an input distribution for probabilistic verification requires significant domain expertise and careful design. For example, a poorly designed input distribution may lead to certifying an unfair classifier as fair. Therefore, verification results are only meaningful if the concrete probabilistic verification problem that was solved is reported and made available alongside the verification result, including the input distribution. Ideally, verification should be conducted by a separate certification body for critical applications.

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

# A. Probabilistic Verification Problems

This section contains the formal definitions of all probabilistic verification problems in this paper.

**Example A.1.** We express the demographic parity fairness notion from Equation (1) as a probabilistic verification problem. Let $\mathcal{X} \subseteq \mathbb{R}^n$ be an input space that encodes information about a person, including a categorical protected attribute, such as gender, race, or disability status that is one-hot encoded at the indices $A \subset [n]$. We assume a single historically advantaged category encoded at the index $a \in A$. Consider a neural network $\mathsf{net} : \mathbb{R}^n \to \mathbb{R}^2$ that acts as a binary classifier making a decision affecting a person, such as hiring or credit approval. The neural network produces a score for each class and assigns the class with the higher score to an input. We rewrite Equation (1) as

$$\frac{\mathbb{P}_\mathbf{x}[\mathsf{net}(\mathbf{x}) = \mathtt{yes} \mid \mathbf{x} \text{ is disadvantaged}]}{\mathbb{P}_\mathbf{x}[\mathsf{net}(\mathbf{x}) = \mathtt{yes} \mid \mathbf{x} \text{ is advantaged}]} \geq \gamma$$

$$\iff \frac{\mathbb{P}_\mathbf{x}[\mathsf{net}(\mathbf{x})_1 - \mathsf{net}(\mathbf{x})_2 \geq 0 \mid \mathbf{x}_a \leq 0]}{\mathbb{P}_\mathbf{x}[\mathsf{net}(\mathbf{x})_1 - \mathsf{net}(\mathbf{x})_2 \geq 0 \mid \mathbf{x}_a \geq 1]} \geq \gamma$$

$$\iff \frac{\mathbb{P}_\mathbf{x}[\mathsf{net}(\mathbf{x})_1 - \mathsf{net}(\mathbf{x})_2 \geq 0 \wedge \mathbf{x}_a \leq 0]/\mathbb{P}_\mathbf{x}[\mathbf{x}_a \leq 0]}{\mathbb{P}_\mathbf{x}[\mathsf{net}(\mathbf{x})_1 - \mathsf{net}(\mathbf{x})_2 \geq 0 \wedge \mathbf{x}_a \geq 1]/\mathbb{P}_\mathbf{x}[\mathbf{x}_a \geq 1]} \geq \gamma$$

$$\iff \frac{\mathbb{P}_\mathbf{x}[\min(\mathsf{net}(\mathbf{x})_1 - \mathsf{net}(\mathbf{x})_2, -\mathbf{x}_a) \geq 0]/\mathbb{P}_\mathbf{x}[-\mathbf{x}_a \geq 0]}{\mathbb{P}_\mathbf{x}[\min(\mathsf{net}(\mathbf{x})_1 - \mathsf{net}(\mathbf{x})_2, \mathbf{x}_a - 1) \geq 0]/\mathbb{P}_\mathbf{x}[\mathbf{x}_a - 1 \geq 0]} \geq \gamma$$

$$\iff \frac{\mathbb{P}_\mathbf{x}[g_{\mathrm{Sat}}^{(1)}(\mathbf{x}, \mathsf{net}(\mathbf{x})) \geq 0]/\mathbb{P}_\mathbf{x}[g_{\mathrm{Sat}}^{(2)}(\mathbf{x}, \mathsf{net}(\mathbf{x})) \geq 0]}{\mathbb{P}_\mathbf{x}[g_{\mathrm{Sat}}^{(3)}(\mathbf{x}, \mathsf{net}(\mathbf{x})) \geq 0]/\mathbb{P}_\mathbf{x}[g_{\mathrm{Sat}}^{(4)}(\mathbf{x}, \mathsf{net}(\mathbf{x})) \geq 0]} - \gamma \geq 0$$

$$\iff f_{\mathrm{Sat}}\left(\mathbb{P}_\mathbf{x}\left[g_{\mathrm{Sat}}^{(1)}(\mathbf{x}, \mathsf{net}(\mathbf{x})) \geq 0\right], \ldots, \mathbb{P}_\mathbf{x}\left[g_{\mathrm{Sat}}^{(4)}(\mathbf{x}, \mathsf{net}(\mathbf{x})) \geq 0\right]\right) \geq 0$$

where, $f_{\mathrm{Sat}}(p_1, p_2, p_3, p_4) = (p_1 p_4)/(p_2 p_3) - \gamma$, $g_{\mathrm{Sat}}^{(1)}(\mathbf{x}, \mathsf{net}(\mathbf{x})) = \min(\mathsf{net}(\mathbf{x})_1 - \mathsf{net}(\mathbf{x})_2, -\mathbf{x}_a)$, $g_{\mathrm{Sat}}^{(2)}(\mathbf{x}, \mathsf{net}(\mathbf{x})) = -\mathbf{x}_a$, $g_{\mathrm{Sat}}^{(3)}(\mathbf{x}, \mathsf{net}(\mathbf{x})) = \min(\mathsf{net}(\mathbf{x})_1 - \mathsf{net}(\mathbf{x})_2, \mathbf{x}_a - 1)$, and $g_{\mathrm{Sat}}^{(4)}(\mathbf{x}, \mathsf{net}(\mathbf{x})) = \mathbf{x}_a - 1$.

## A.1. Parity of Qualified Persons

The following probabilistic verification problem concerns verifying the parity of qualified persons, a variant of demographic parity that only considers the subpopulation of persons qualified for, for example, hiring (Albarghouthi et al., 2017). Let $\mathcal{X} \subseteq \mathbb{R}^n$, $A \subset [n]$, $a \in A$, and $\mathsf{net} : \mathbb{R}^n \to \mathbb{R}^2$ be as in Example A.1. Additionally, let $q \in [n] \setminus A$ and $\hat{q} \in \mathbb{R}$, such that persons with $\mathbf{x}_q \geq \hat{q}$ are considered to be qualified. In their extended set of experiments, Albarghouthi et al. (2017) consider a $q$ that encodes age and $\hat{q} = 18$ so that only persons who are at least 18 years old are considered to be qualified. The parity of qualified persons fairness notion is

$$\frac{\mathbb{P}_\mathbf{x}[\mathsf{net}(\mathbf{x}) = \mathtt{yes} \mid \mathbf{x} \text{ is disadvantaged} \wedge \mathbf{x} \text{ is qualified}]}{\mathbb{P}_\mathbf{x}[\mathsf{net}(\mathbf{x}) = \mathtt{yes} \mid \mathbf{x} \text{ is advantaged} \wedge \mathbf{x} \text{ is qualified}]} \geq \gamma$$

$$\iff \frac{\mathbb{P}_\mathbf{x}[\mathsf{net}(\mathbf{x})_1 - \mathsf{net}(\mathbf{x})_2 \geq 0 \mid \mathbf{x}_a \leq 0 \wedge \mathbf{x}_q \geq \hat{q}]}{\mathbb{P}_\mathbf{x}[\mathsf{net}(\mathbf{x})_1 - \mathsf{net}(\mathbf{x})_2 \geq 0 \mid \mathbf{x}_a \geq 1 \wedge \mathbf{x}_q \geq \hat{q}]} \geq \gamma$$

$$\iff \frac{\mathbb{P}_\mathbf{x}[\mathsf{net}(\mathbf{x})_1 - \mathsf{net}(\mathbf{x})_2 \geq 0 \mid \min(-\mathbf{x}_a, \mathbf{x}_q - \hat{q}) \geq 0]}{\mathbb{P}_\mathbf{x}[\mathsf{net}(\mathbf{x})_1 - \mathsf{net}(\mathbf{x})_2 \geq 0 \mid \min(\mathbf{x}_a - 1, \mathbf{x}_q - \hat{q}) \geq 0]} \geq \gamma$$

$$\iff f_{\mathrm{Sat}}\left(\mathbb{P}_\mathbf{x}\left[g_{\mathrm{Sat}}^{(1)}(\mathbf{x}, \mathsf{net}(\mathbf{x})) \geq 0\right], \ldots, \mathbb{P}_\mathbf{x}\left[g_{\mathrm{Sat}}^{(4)}(\mathbf{x}, \mathsf{net}(\mathbf{x})) \geq 0\right]\right) \geq 0$$

where $\gamma \in [0, 1]$, $f_{\mathrm{Sat}}(p_1, p_2, p_3, p_4) = (p_1 p_4)/(p_2 p_3) - \gamma$, $g_{\mathrm{Sat}}^{(1)}(\mathbf{x}, \mathsf{net}(\mathbf{x})) = \min(\mathsf{net}(\mathbf{x})_1 - \mathsf{net}(\mathbf{x})_2, -\mathbf{x}_a, \mathbf{x}_q - \hat{q})$, $g_{\mathrm{Sat}}^{(2)}(\mathbf{x}, \mathsf{net}(\mathbf{x})) = \min(-\mathbf{x}_a, \mathbf{x}_q - \hat{q})$, $g_{\mathrm{Sat}}^{(3)}(\mathbf{x}, \mathsf{net}(\mathbf{x})) = \min(\mathsf{net}(\mathbf{x})_1 - \mathsf{net}(\mathbf{x})_2, \mathbf{x}_a - 1, \mathbf{x}_q - \hat{q})$, and $g_{\mathrm{Sat}}^{(4)}(\mathbf{x}, \mathsf{net}(\mathbf{x})) = \min(\mathbf{x}_a - 1, \mathbf{x}_q - \hat{q})$.

## A.2. ACAS Xu Safety

Next, we consider Equation (2) for an ACAS Xu network, where to be safe means satisfying property $\phi_2$ of Katz et al. (2017b). For quantifying the number of violations, we first define what it means for an ACAS Xu neural network $\mathsf{net} : \mathbb{R}^5 \to \mathbb{R}^5$ to

violate $\phi_2$. Using the satisfaction functions of Bauer-Marquart et al. (2022), violating $\phi_2$ means

$$g_{\text{Sat}}(\mathbf{x}, \text{net}(\mathbf{x})) = \max_{i=2}^{5} \text{net}(\mathbf{x})_i - \text{net}(\mathbf{x})_1 < 0 \quad \forall \mathbf{x} \in \mathcal{X}_{\phi_2} \cap \mathcal{X}, \tag{5}$$

where $\mathcal{X}$ is the bounded hyperrectangular input space of net and

$$\mathcal{X}_{\phi_2} = [55947.961, \infty] \times \mathbb{R}^2 \times [1145, \infty] \times [-\infty, 60].$$

We refer to Katz et al. (2017b) for an interpretation of $\phi_2$ in the application context. Quantifying the number of violations with respect to $\phi_2$ corresponds to computing

$$\ell \leq \mathbb{P}_{\mathbf{x}}[g_{\text{Sat}}(\mathbf{x}, \text{net}(\mathbf{x})) < 0] = \mathbb{P}_{\mathbf{x}}[-g_{\text{Sat}}(\mathbf{x}, \text{net}(\mathbf{x})) \geq 0] \leq u,$$

where $g_{\text{Sat}}$ is as in Equation (5) and $\mathbf{x}$ is uniformly distributed on $\mathcal{X}_{\phi_2} \cap \mathcal{X}$ with all points outside $\mathcal{X}_{\phi_2} \cap \mathcal{X}$ having zero probability.

### A.3. ACAS Xu Robustness

For the ACAS Xu robustness experiment in Section 6.2, we solve five probabilistic verification problems for each reference input $\mathbf{x}$ — one for each of the five classes. Our goal is to bound the probability of net classifying an input $\mathbf{x}'$ as class $i \in [5]$, where $\mathbf{x}'$ is close to the reference input $\mathbf{x}$ in the first two dimensions and identical to $\mathbf{x}$ in the remaining dimensions.

Let net be the ACAS Xu network $N_{1,1}$ of Katz et al. (2017b) with input space $\mathcal{X} = [\underline{\mathbf{x}}, \overline{\mathbf{x}}]$. Let $\mathbf{x}$ be a reference input. Note that the ACAS Xu networks assign the class with the *minimal* score to an input instead of using the maximal score. For bounding the probability of obtaining class $i \in [5]$ for inputs close to $\mathbf{x}$, we compute bounds on

$$\mathbb{P}_{\mathbf{x}'}[g_{\text{Sat}}(\mathbf{x}', \text{net}(\mathbf{x}')) \geq 0],$$

$$g_{\text{Sat}}(\mathbf{x}', \text{net}(\mathbf{x}')) = \min_{\substack{j=1 \\ j \neq i}}^{5} \text{net}(\mathbf{x}')_j - \text{net}(\mathbf{x}')_i$$

where $\mathbf{x}'$ is uniformly distributed on the set $\mathcal{X} \cap \left([\mathbf{x}_{1:2} - 0.05 \cdot \mathbf{w}_{1:2}, \mathbf{x}_{1:2} + 0.05 \cdot \mathbf{w}_{1:2}] \times \{\mathbf{x}_{3:5}\}\right)$, where $\mathbf{w} = \overline{\mathbf{x}} - \underline{\mathbf{x}}$ and $\mathbf{z}_{i:j}$ is the vector containing the elements $i, \ldots, j$ of a vector $\mathbf{z}$.

### A.4. VCAS Correctness

In the VCAS correctness experiment in Section 6.2, the goal is to prove whether the VCAS network net : $\mathbb{R}^4 \rightarrow \mathbb{R}^9$ of Zhang et al. (2024) satisfies

$$\mathbb{P}_{\mathbf{x}}[g_{\text{Sat}}(\mathbf{x}, \text{net}(\mathbf{x})) \geq 0] \geq 0.9,$$

where $g_{\text{Sat}}(\mathbf{x}, \text{net}(\mathbf{x})) = \text{net}(\mathbf{x})_1 - \min_{j=2}^{9} \text{net}(\mathbf{x})_j$, which encodes that the network predicts 'Clear Of Conflict', and $\mathbf{x}$ is uniformly distributed on the set $[-8000, 0] \times [0, 100] \times \{-30\} \times [0, 40]$. We refer to Zhang et al. (2024) for an interpretation of this specification in the application context.

### A.5. Useful Modelling Techniques

As discussed in Section 4.2, PV requires the input space $\mathcal{X}$ to be a hyperectangle. Further, it requires that each input distribution $\mathbb{P}_{\mathbf{x}}$ allows for computing the probability of a hyperrectangle in closed form. This section shows how some of these restrictions can be mitigated. Concretely, we show how to use multivariate normal distributions as input distributions and polytopes as input spaces.

We first show how we can apply PV to multivariate normal distributions by transforming the input distribution and the network to verify. Consider a multivariate normal distribution $\mathbb{P}_{\mathbf{z}}$ with mean $\boldsymbol{\mu}$ and covariance $\boldsymbol{\Sigma} = \mathbf{A}\mathbf{A}^{\top}$. If $\boldsymbol{\Sigma}$ is diagonal, the probability of a hyperrectangle has a closed-form solution, so that we can compute it efficiently. Here, we are interested in the case where $\boldsymbol{\Sigma}$ is not diagonal, so that we can not compute the probability of a hyperrectangle directly. In this case, let $\mathbb{P}_{\mathbf{x}}$ be a standard multivariate normal distribution. Now, $\mathbf{z} = \mathbf{A}\mathbf{x} + \boldsymbol{\mu}$ is distributed according to $\mathbb{P}_{\mathbf{z}}$. Therefore, by prepending the linear transformation $\mathbf{A}\mathbf{x} + \boldsymbol{\mu}$ to net, we can apply PV to general multivariate normal distributions, since the probability of a hyperrectangle under a standard multivariate normal distribution has a closed-form solution.

Second, we show how to apply PV to polytopal input spaces, even though they can not be used as input space directly. Let $\mathcal{P} = \{\mathbf{x} \in \mathbb{R}^n \mid \mathbf{A}\mathbf{x} \leq \mathbf{b}\}$ be a polytope and let $\mathcal{X} \supseteq \mathcal{P}$ be a hyperrectangle enclosing $\mathcal{P}$. By using $\mathcal{X}$ as the input space and using

$$\mathbb{P}_{\mathbf{x}}[g_{\text{Sat}}(\mathbf{x}, \mathsf{net}(\mathbf{x})) \geq 0 \mid \mathbf{A}\mathbf{x} \leq \mathbf{b}] = \frac{\mathbb{P}_{\mathbf{x}}[g_{\text{Sat}}(\mathbf{x}, \mathsf{net}(\mathbf{x})) \geq 0 \wedge \mathbf{A}\mathbf{x} \leq \mathbf{b}]}{\mathbb{P}_{\mathbf{x}}[\mathbf{A}\mathbf{x} \leq \mathbf{b}]}$$

we can apply PV to polytopes as input spaces. However, before applying PV, we should check whether $\mathbb{P}_{\mathbf{x}}[\mathbf{A}\mathbf{x} \leq \mathbf{b}] > 0$, since, otherwise, the verification problem is ill-defined. We can apply PROBBOUNDS for this purpose by computing bounds on $\mathbb{P}_{\mathbf{x}}[\mathbf{A}\mathbf{x} \leq \mathbf{b}]$.

## B. Additional Details on PROBBOUNDS

This section contains additional details on PROBBOUNDS (Algorithm 2). It includes a detailed description of our procedure for splitting dimensions and a motivation and additional details on our BABSB SPLIT heuristic.

### B.1. Splitting

Section 4.1 describes how to split a dimension $d \in [n]$ to refine a branch. This section formally defines the splitting procedure that PROBBOUNDS applies. A dimension can encode several types of variables. We consider continuous variables, such as normalised pixel values, integer variables, such as age, and dimensions containing one indicator of a one-hot encoded categorical variable like gender. The type of variable encoded in $d$ determines how we split $d$.

- For continuous variables, we further differentiate whether $\mathcal{B}$ is bounded, unbounded in one direction, or unbounded in both directions in dimension $d$.
  - If $\mathcal{B}$ is bounded in dimension $d$, we bisect $\mathcal{B}$ along $d$ resulting in two new branches $[\underline{\mathbf{x}}', \overline{\mathbf{x}}']$ and $[\underline{\mathbf{x}}'', \overline{\mathbf{x}}'']$. Concretely, $\underline{\mathbf{x}}'_{d'} = \underline{\mathbf{x}}''_{d'} = \underline{\mathbf{x}}_{d'}$ and $\overline{\mathbf{x}}'_{d'} = \overline{\mathbf{x}}''_{d'} = \overline{\mathbf{x}}_{d'}$ for all $d' \in [n] \backslash \{d\}$ while $\overline{\mathbf{x}}'_d = \underline{\mathbf{x}}''_d = (\underline{\mathbf{x}}_d + \overline{\mathbf{x}}_d)/2$, $\underline{\mathbf{x}}'_d = \underline{\mathbf{x}}_d$, and $\overline{\mathbf{x}}''_d = \overline{\mathbf{x}}_d$.
  - If $\mathcal{B}$ is unbounded in both directions in $d$, we split $d$ at zero, so that $\overline{\mathbf{x}}'_d = \underline{\mathbf{x}}''_d = 0$. The remaining bounds of the new branches $[\underline{\mathbf{x}}', \overline{\mathbf{x}}']$ and $[\underline{\mathbf{x}}'', \overline{\mathbf{x}}'']$ are as in the bounded case.
  - If $\mathcal{B}$ is bounded from below but unbounded from above in $d$, that is $-\infty < \underline{\mathbf{x}}_d < \overline{\mathbf{x}}_d = \infty$, we split at $\overline{\mathbf{x}}'_d = \underline{\mathbf{x}}''_d = \max(2\underline{\mathbf{x}}_d, 1)$, all else being as above. Effectively, this split rule performs an exponential search over unbounded dimensions until the remaining unbounded branches are no longer selected by SELECT, for example, because they have diminishing probability in the case of SELECTPROB. We handle the case where $d$ is bounded from above but unbounded from below analogously.

- For integer variables, we split $d$ as for a continuous variable to obtain $[\underline{\mathbf{x}}', \overline{\mathbf{x}}']$, $[\underline{\mathbf{x}}'', \overline{\mathbf{x}}'']$ and round $\overline{\mathbf{x}}'_d$ to the next smaller integer while rounding $\underline{\mathbf{x}}''_d$ to the next larger integer.

- For a one-hot encoded categorical variable $V$ encoded in the dimensions $A \subseteq [n]$ with $d \in A$, we create one split where $V$ is equal to the category represented by $d$ and one where $V$ is different from this category. Formally, $\underline{\mathbf{x}}'_d = \overline{\mathbf{x}}'_d = 1$ and $\underline{\mathbf{x}}'_{d'} = \overline{\mathbf{x}}'_{d'} = 0$ for $d' \in A \backslash \{d\}$ defines $[\underline{\mathbf{x}}', \overline{\mathbf{x}}']$. For $[\underline{\mathbf{x}}'', \overline{\mathbf{x}}'']$, we set $\underline{\mathbf{x}}''_d = \overline{\mathbf{x}}''_d = 0$ and leave the remaining values are they are in $\underline{\mathbf{x}}$ and $\overline{\mathbf{x}}$. This splitting procedure eventually creates a new branch where all dimensions are set to zero. This branch has zero probability and can be discarded immediately.

In any case, we need to ensure not to select $d$ if $\underline{\mathbf{x}}_d = \overline{\mathbf{x}}_d$.

### B.2. BABSB

Our BABSB split selection heuristic is a variation of the BABSB heuristic for non-probabilistic neural network verification of Bunel et al. (2020). One difference is that Bunel et al. (2020) use the method of Wong & Kolter (2018) for estimating the improvement in bounds, while we use INTERVALARITHMETIC. Another difference is that while Bunel et al. (2020) are mainly interested in lower bounds, we are equally interested in lower and upper bounds. Let $[\underline{\mathbf{x}}^{(d,1)}, \overline{\mathbf{x}}^{(d,1)}]$ and $[\underline{\mathbf{x}}^{(d,2)}, \overline{\mathbf{x}}^{(d,2)}]$ be the two new branches originating from splitting dimension $d \in [n]$ and let $\underline{y}^{(d,1)}, \overline{y}^{(d,1)}, \underline{y}^{(d,2)}, \overline{y}^{(d,2)}$ be the bounds that INTERVALARITHMETIC computes on $g_{\text{Sat}}(\cdot, \mathsf{net}(\cdot))$ for these branches. Our BABSB heuristic selects $d = \arg\max_{d \in [n]} \tilde{y}^{(d)}$, where $\tilde{y}^{(d)} = \max(\max(\underline{y}^{(d,1)}, \underline{y}^{(d,2)}), -\min(\overline{y}^{(d,1)}, \overline{y}^{(d,2)}))$. In other words, we select the

dimension $d$ that yields the largest lower bound or smallest upper bound in any of the new branches, while Bunel et al. (2020) select the dimension $d$ with the largest lower bound among the smaller lower bound for the two branches originating from splitting $d$. We found this variant to be the most successful for our application. Bunel et al. (2020) discuss further variants.

**Implementation.** We round all bounds to four decimal places to mitigate floating point issues. If several dimensions yield equal improvements in bounds, we randomly select one of these dimensions. Without this random tie-breaking, we might split a single dimension repeatedly if the INTERVALARITHMETIC bounds are very loose. We use a separate pseudo-random number generator with a fixed seed for this tie-breaking so that BABSB remains entirely deterministic.

## C. Extended Theoretical Analysis

This section contains the proofs of the theorems in Section 5. We also give a more general completeness analysis of PV than presented in Section 5.

### C.1. Soundness

This section contains the proofs for our soundness results from Section 5.

*Proof of Theorem 5.1.* Let $t \in \mathbb{N}$ and let $\mathcal{X}_{sat}^{(t)}$ and $\mathcal{X}_{viol}^{(t)}$ be as in Algorithm 2. PROBBOUNDS computes $\ell^{(t)}$ as the total probability of all previously pruned satisfied branches $\hat{\mathcal{X}}_{sat}^{(t)} = \bigcup_{t'=1}^{t} \mathcal{X}_{sat}^{(t')}$. Similarly, $u^{(t)} = 1 - \hbar^{(t)}$ where $\hbar^{(t)}$ is the total probability of all previously pruned violated branches $\hat{\mathcal{X}}_{viol}^{(t)} = \bigcup_{t'=1}^{t} \mathcal{X}_{viol}^{(t')}$. Since we assumed that COMPUTEBOUNDS produces valid bounds, PRUNE only prunes branches that are actually satisfied or violated. Therefore, $\hat{\mathcal{X}}_{sat}^{(t)} \subseteq \{\mathbf{x} \in \mathcal{X} \mid g_{\text{Sat}}(\mathbf{x}, \text{net}(\mathbf{x})) \geq 0\}$ and $\hat{\mathcal{X}}_{viol}^{(t)} \subseteq \{\mathbf{x} \in \mathcal{X} \mid g_{\text{Sat}}(\mathbf{x}, \text{net}(\mathbf{x})) < 0\}$. From this, it follows directly that

$$\ell^{(t)} = \mathbb{P}_{\mathbf{x}}\left[\hat{\mathcal{X}}_{sat}^{(t)}\right] \leq \mathbb{P}_{\mathbf{x}}[g_{\text{Sat}}(\mathbf{x}, \text{net}(\mathbf{x})) \geq 0]$$

$$\hbar^{(t)} = \mathbb{P}_{\mathbf{x}}\left[\hat{\mathcal{X}}_{viol}^{(t)}\right] \leq \mathbb{P}_{\mathbf{x}}[g_{\text{Sat}}(\mathbf{x}, \text{net}(\mathbf{x})) < 0],$$

which implies $u^{(t)} = 1 - \hbar^{(t)} \geq 1 - \mathbb{P}_{\mathbf{x}}[g_{\text{Sat}}(\mathbf{x}, \text{net}(\mathbf{x})) < 0] = \mathbb{P}_{\mathbf{x}}[g_{\text{Sat}}(\mathbf{x}, \text{net}(\mathbf{x})) \geq 0]$. This shows that PROBBOUNDS is sound. □

*Proof of Corollary 5.2.* Corollary 5.2 follows from Theorem 5.1 and the soundness of the COMPUTEBOUNDS procedure applied by PV. □

### C.2. Completeness

This section is concerned with proving our completeness result from Section 5. We first discuss in more detail why Assumption 5.3 is only mildly restrictive. Next, we define conditions on the SPLIT, SELECT, and COMPUTEBOUNDS procedures that ensure the completeness of PV. We then prove that the SELECTPROB, LONGESTEDGE, BABSB-LONGESTEDGE-k heuristics and INTERVALARITHMETIC, as well as CROWN satisfy these conditions. Finally, we prove the completeness of PV.

**Discussion of Assumption 5.3.** The proof of Theorem 5.5 is based on Lemma 5.4 that states that PROBBOUNDS produces a sequence of lower and upper bounds that converge towards each other. Intuitively, we require Assumption 5.3 since converging bounds on $f_{\text{Sat}}(p_1, \ldots, p_n)$ are insufficient for proving $f_{\text{Sat}}(p_1, \ldots, p_n) \geq 0$ if $f_{\text{Sat}}(p_1, \ldots, p_n) = 0$ (Albarghouthi et al., 2017). Note that $p_1, \ldots, p_v$ and $f_{\text{Sat}}(p_1, \ldots, p_v)$ are unknown but fixed values in Equation (3).

For illustration, assume we want to show $y \geq 0$, where $y \in \mathbb{R}$ is an unknown constant. We are provided with converging sequences of bounds $(\ell_t)_{t \in \mathbb{N}}$ and $(u_t)_{t \in \mathbb{N}}$ with $\ell_t \leq y \leq u_t$ for each $t \in \mathbb{N}$ and $\lim_{t \to \infty} \ell_t = \lim_{t \to \infty} u_t = y$. If $y = 0$, the sequences of bounds that only converge in the limit do not suffice to prove $y \geq 0$, since there may not be a $T \in \mathbb{N}$ with $\ell_T = 0$. However, if the $y \neq 0$, obtaining a finite number of iterates of $(\ell_t)_{t \in \mathbb{N}}$ and $(u_t)_{t \in \mathbb{N}}$ always suffices for proving or disproving $y \geq 0$. Concretely, there will be a $T \in \mathbb{N}$, such that either $\ell_T > 0$ or $u_T < 0$, that proves, respectively, disproves $y \geq 0$. The assumption that $f_{\text{Sat}}(p_1, \ldots, p_v) \neq 0$ corresponds to assuming $y \neq 0$ in this example.

In Section 5, we describe studying $f'_{\text{Sat}}(p_1, \ldots, p_n) = f_{\text{Sat}}(p_1, \ldots, p_n) - \varepsilon$ for some $\varepsilon > 0$, if we suspect that $f_{\text{Sat}}(p_1, \ldots, p_n) = 0$. If $f_{\text{Sat}}(p_1, \ldots, p_n) = 0$, the probabilistic verification problem with $f'_{\text{Sat}}$ in place of $f_{\text{Sat}}$ satisfies Assumption 5.3 and is only marginally stronger than the original verification problem.

The motivation for requiring $\mathbb{P}_{\mathbf{x}^{(i)}}[g^{(i)}_{\text{Sat}}(\mathbf{x}^{(i)}, \text{net}(\mathbf{x}^{(i)})) = 0] = 0, \forall i \in [v]$ is similar as for requiring $f_{\text{Sat}}(p_1, \ldots, p_n) \neq 0$. If $\mathbb{P}_{\mathbf{x}^{(i)}}[g^{(i)}_{\text{Sat}}(\mathbf{x}^{(i)}, \text{net}(\mathbf{x}^{(i)})) = 0] \neq 0$, there can be a region of the input space with positive probability that we can never prune, since the bounds computed by interval arithmetic or CROWN may only converge in the limit for this region. However, if this is the case, we can tighten $g^{(i)}_{\text{Sat}}(\mathbf{x}^{(i)}, \text{net}(\mathbf{x}^{(i)})) \geq 0$ to $g^{(i)}_{\text{Sat}}(\mathbf{x}^{(i)}, \text{net}(\mathbf{x}^{(i)})) \geq \varepsilon$ for some $\varepsilon > 0$ such that $\mathbb{P}_{\mathbf{x}^{(i)}}[g^{(i)}_{\text{Sat}}(\mathbf{x}^{(i)}, \text{net}(\mathbf{x}^{(i)})) = \varepsilon] = 0$. Such an $\varepsilon > 0$ exists because $\mathbb{P}_{\mathbf{x}}[g_{\text{Sat}}(\mathbf{x}, \text{net}(\mathbf{x})) = 0] = 0$ means that the satisfaction boundary has positive volume but any neural network has only finitely many flat regions that can produce a satisfaction boundary of positive volume. We now define conditions on the COMPUTEBOUNDS, SPLIT, and SELECT procedures that ensure the completeness of PV.

**Definition C.1** (Convergent Bounds). Let $f : \mathbb{R}^n \to \mathbb{R}^m$. We call a COMPUTEBOUNDS procedure that computes $\underline{\mathbf{y}} \leq f(\mathbf{x}) \leq \overline{\mathbf{y}}$ for $\mathbf{x} \in [\underline{\mathbf{x}}, \overline{\mathbf{x}}]$ *convergent* if $\|\overline{\mathbf{y}} - \underline{\mathbf{y}}\| \to 0$ as $\|\overline{\mathbf{x}} - \underline{\mathbf{x}}\| \to 0$ and $\|\overline{\mathbf{y}} - \underline{\mathbf{y}}\| = 0$ if $\|\overline{\mathbf{x}} - \underline{\mathbf{x}}\| = 0$.

**Definition C.2** (Dimension Alternation). Let $[\underline{\mathbf{x}}, \overline{\mathbf{x}}] \subseteq \mathbb{R}^n$. A splitting procedure SPLIT is *dimension-alternating* if for every $d \in [n]$ with $\underline{\mathbf{x}}_d \neq \overline{\mathbf{x}}_d$

$$\exists t \in \mathbb{N} : \exists [\underline{\mathbf{x}}', \overline{\mathbf{x}}'] \in \text{branches}^{(t)} : \overline{\mathbf{x}}'_d - \underline{\mathbf{x}}'_d < \overline{\mathbf{x}}_d - \underline{\mathbf{x}}_d,$$

where $\text{branches}^{(t)} = \text{SPLIT}(\text{branches}^{(t-1)})$ for $t \in \mathbb{N}$ and $\text{branches}^{(0)} = [\underline{\mathbf{x}}, \overline{\mathbf{x}}]$.

**Definition C.3** (Branch Alternation). A branch selection procedure SELECT is *branch-alternating* if

$$\forall t \in \mathbb{N} : \forall \mathcal{B} \in \text{branches}^{(t)} : \mathbb{P}_{\mathbf{x}}[\mathcal{B}] > 0 \implies \exists t' \geq t : \mathcal{B} \in \text{SELECT}(\text{branches}^{(t')}, N),$$

where $N \in \mathbb{N}$ and $\text{branches}^{(t)}$ is the value of the branches variable of PROBBOUNDS in iteration $t$ where PROBBOUNDS is instantiated with SELECT and a COMPUTEBOUNDS procedure satisfying Definition C.1.

In the following, we prove that SELECTPROB, LONGESTEDGE, and BABSB-LONGESTEDGE-k as introduced in Section 4.1 are branch alternating and dimensional alternating, respectively. It is well-known that INTERVALARITHMETIC satisfies Definition C.1 (Moore et al., 2009). We provide a proof in Appendix D.2. We show that CROWN satisfies Definition C.1 in Appendix E.

**Proposition C.4.** LONGESTEDGE *satisfies Definition C.2*.

*Proof.* Let $[\underline{\mathbf{x}}, \overline{\mathbf{x}}] \subseteq \mathbb{R}^n, d \in [n]$ with $\underline{\mathbf{x}}_d \neq \overline{\mathbf{x}}_d$, and let $\text{branches}^{(t)}$ for $t \in \mathbb{N}_0$ be as in Definition C.2. We call $\overline{\mathbf{x}}_d - \underline{\mathbf{x}}_d$ the *edge length* of $d$ in $[\underline{\mathbf{x}}, \overline{\mathbf{x}}]$.

If $\overline{\mathbf{x}}_d - \underline{\mathbf{x}}_d > \max_{d' \neq d} \overline{\mathbf{x}}_{d'} - \underline{\mathbf{x}}_{d'}$, the dimension $d$ is selected for splitting immediately. In the following, we not only show that $\overline{\mathbf{x}}_d - \underline{\mathbf{x}}_d$ decreases when split but also that $\overline{\mathbf{x}}_d - \underline{\mathbf{x}}_d \to 0$ in at least one branch when we split $d$ repeatedly. This result is required for the second part of this proof. We differentiate several cases based on the variable encoded in $d$.

- *Bounded Continuous Variable.* Let $d$ encode a continuous variable with $\overline{\mathbf{x}}_d - \underline{\mathbf{x}}_d < \infty$. As described in Section 4.1 we split such dimensions by bisecting $[\underline{\mathbf{x}}, \overline{\mathbf{x}}]$ along $d$. Bisecting decreases the edge length of $d$ in the resulting branches so that we have $\overline{\mathbf{x}}'_d - \underline{\mathbf{x}}'_d < \overline{\mathbf{x}}_d - \underline{\mathbf{x}}_d$ for all $[\underline{\mathbf{x}}', \overline{\mathbf{x}}'] \in \text{branches}^{(1)} = \text{SPLIT}([\underline{\mathbf{x}}, \overline{\mathbf{x}}])$. Furthermore, the edge length of $d$ converges towards zero if we bisect along $d$ repeatedly.

- *Continuous Variable Bounded from Below but Unbounded from Above.* Let $d$ encode a continuous variable with $-\infty < \underline{\mathbf{x}}_d < \overline{\mathbf{x}}_d = \infty$. Since splitting such a dimension creates one branch where $d$ is bounded, we have $\overline{\mathbf{x}}'_d - \underline{\mathbf{x}}'_d < \overline{\mathbf{x}}_d - \underline{\mathbf{x}}_d$ for the bounded branch $[\underline{\mathbf{x}}', \overline{\mathbf{x}}'] \in \text{branches}^{(1)}$. Furthermore, repeatedly splitting the bounded branch along $d$ lets the edge length of $d$ converge towards zero, as discussed above.

- *Continuous Variable Bounded from Above but Unbounded from Below.* This case proceeds analogously to the previous case.

- *Continuous Variable Unbounded from Both Sides.* Splitting along such variables creates two branches that are bounded from one side. Therefore, after two splits, we obtain two bounded branches, such that $\overline{\mathbf{x}}'_d - \underline{\mathbf{x}}'_d < \overline{\mathbf{x}}_d - \underline{\mathbf{x}}_d$ for two $[\underline{\mathbf{x}}', \overline{\mathbf{x}}'] \in \text{branches}^{(2)} = \text{SPLIT}(\text{branches}^{(1)})$. Similarly, repeatedly splitting the bounded branches along $d$ lets the edge length of $d$ converge towards zero.

- *Integer Variable.* Let $d$ encode an integer variable. Splitting $d$ proceeds as for a continuous variable, except for excluding non-integer values from the new branches. This decreases the edge length of $d$ at least as much as if we were splitting a continuous variable. Therefore, we have that $\overline{\mathbf{x}}'_d - \underline{\mathbf{x}}'_d < \overline{\mathbf{x}}_d - \underline{\mathbf{x}}_d$ for at least one branch $[\underline{\mathbf{x}}', \overline{\mathbf{x}}'] \in \text{branches}^{(1)}$. Furthermore, the edge length of $d$ reaches zero after finitely many splits in all bounded branches since we exclude non-integer values.

- *One-Hot Encoded Categorical Variable.* Let $d$ contain an indicator of a one-hot encoded categorical variable. Splitting $d$ decreases the edge length of $d$ to zero, so that we have $\overline{\mathbf{x}}'_d - \underline{\mathbf{x}}'_d = 0 < \overline{\mathbf{x}}_d - \underline{\mathbf{x}}_d$ for all $[\underline{\mathbf{x}}', \overline{\mathbf{x}}'] \in \text{branches}^{(1)}$.

Overall, Definition C.2 is satisfied if dimension $d$ is selected for splitting immediately.

Now consider the case that $d$ is not selected for splitting immediately. In this case, a different dimension $d' \in [n], d' \neq d$ with $\overline{\mathbf{x}}_{d'} - \underline{\mathbf{x}}_{d'} \geq \overline{\mathbf{x}}_d - \underline{\mathbf{x}}_d$, is selected for splitting by LONGESTEDGE. As we have argued above, repeatedly splitting $d'$ lets the edge length of $d'$ decrease towards zero in at least one branch. Therefore, we eventually obtain $[\underline{\mathbf{x}}', \overline{\mathbf{x}}'] \in \text{branches}^{(t)}$ with $\overline{\mathbf{x}}'_{d'} - \underline{\mathbf{x}}'_{d'} < \overline{\mathbf{x}}_d - \underline{\mathbf{x}}_d$. Since this holds for all $d'' \in [n], d'' \neq d$ with $\overline{\mathbf{x}}_{d''} - \underline{\mathbf{x}}_{d''} \geq \overline{\mathbf{x}}_d - \underline{\mathbf{x}}_d$, we eventually obtain a branch where LONGESTEDGE splits $d$. Therefore, Definition C.2 is also satisfied if dimension $d$ is not selected for splitting immediately. Overall, LONGESTEDGE satisfies Definition C.2. ☐

**Corollary C.5.** BABSB-LONGESTEDGE-$k$ *satisfies Definition C.2.*

**Proposition C.6.** SELECTPROB *satisfies Definition C.3.*

*Proof.* Let $\text{branches}^{(t)}$ be the value of the branches variable of PROBBOUNDS (Algorithm 2) in iteration $t \in \mathbb{N}$, where PROBBOUNDS is instantiated with SELECTPROB and a COMPUTEBOUNDS procedure satisfying Definition C.1. Let $N, t \in \mathbb{N}$ and $\mathcal{B} \in \text{branches}^{(t)}$ with $\mathbb{P}_\mathbf{x}[\mathcal{B}] > 0$. Our goal is to show

$$\exists t' \geq t : \mathcal{B} \in \text{SELECTPROB}(\text{branches}^{(t')}, N). \tag{6}$$

If $\mathcal{B} \in \text{SELECTPROB}(\text{branches}^{(t)}, N)$, Equation (6) holds immediately. Otherwise, there are at least $N$ branches $\mathcal{B}'_t$ in iteration $t$ with $\mathbb{P}_\mathbf{x}[\mathcal{B}'_t] \geq \mathbb{P}_\mathbf{x}[\mathcal{B}]$. We show

$$\exists t' > t : \forall \mathcal{B}'_t, \mathbb{P}_\mathbf{x}[\mathcal{B}'_t] \geq \mathbb{P}_\mathbf{x}[\mathcal{B}] : \underbrace{\forall \mathcal{B}'_{t'}, \mathcal{B}'_t \rightsquigarrow \mathcal{B}'_{t'} : \mathbb{P}_\mathbf{x}[\mathcal{B}'_{t'}] < \mathbb{P}_\mathbf{x}[\mathcal{B}]}_{(*)}, \tag{7}$$

where $\mathcal{B}'_t \rightsquigarrow \mathcal{B}'_{t'}$ if $\mathcal{B}'_{t'}$ is a branch in iteration $t' \in \mathbb{N}$ that originates from splitting $\mathcal{B}'_t$, meaning that $\mathcal{B}'_{t'} \subset \mathcal{B}'_t$.

Let $\mathcal{B}'_t$ be a branch in iteration $t$ with $\mathbb{P}_\mathbf{x}[\mathcal{B}'_t] \geq \mathbb{P}_\mathbf{x}[\mathcal{B}_t]$. First of all, if $\mathcal{B}'_t$ is pruned by PROBBOUNDS in iteration $t$, then there are no new branches originating from $\mathcal{B}'_t$, so that $(*)$ holds vacuously. Otherwise, PROBBOUNDS splits $\mathcal{B}'_t$.

We first consider the special case where the input space only contains categorical and bounded integer variables. Dimensions encoding such variables can only be split finitely often. Therefore, splitting $\mathcal{B}'_t$ eventually produces a finite set of branches $[\underline{\mathbf{x}}, \overline{\mathbf{x}}]$ with $\underline{\mathbf{x}} = \overline{\mathbf{x}}$. Since we assumed COMPUTEBOUNDS to satisfy Definition C.1, COMPUTEBOUNDS computes the bounds $\underline{\mathbf{y}} = \overline{\mathbf{y}}$ for branches with $\underline{\mathbf{x}} = \overline{\mathbf{x}}$. Branches with $\underline{\mathbf{y}} = \overline{\mathbf{y}}$ are certainly pruned by PROBBOUNDS. Therefore, if we choose $t' > t$ large enough, Equation (7) holds with $(*)$ holding vacuously, since all branches originating from $\mathcal{B}'_t$ have been pruned.

Otherwise, let the input space contain at least one continuous or unbounded integer variable. We show that there is an iteration $t' > t$ such that $(*)$ holds for $\mathcal{B}'_t$. Without loss of generality, assume that the dimension selected for splitting encodes a continuous variable or an unbounded integer variable. This does not harm generality since categorical variables and bounded integer variables can only be split finitely often and, therefore, will eventually become unavailable for splitting.

First, consider splitting along a dimension $d$ encoding a bounded continuous variable. Since we split continuous variables by bisection, the volume of all branches $\mathcal{B}'_{t'}$ originating from $\mathcal{B}'_t$ decreases towards zero as we split $d$ repeatedly. As stated

in Section 3, we assume that all continuous random variables admit a probability density function. This implies that the probability in all branches $\mathcal{B}'_{t'}$ originating from splitting $\mathcal{B}'_t$ decreases towards zero as the volume decreases towards zero.

Now, consider splitting along a dimension $d$ encoding an unbounded variable. Without loss of generality, assume that $\mathcal{B}'_t$ is bounded in $d$ in at least one direction. This does not harm generality since dimensions unbounded in both directions are split into two parts, each bounded in one direction. Given this, splitting along $d$ creates a bounded and an unbounded part. If $d$ encodes an integer variable, the bounded part contains only finitely many discrete values. Therefore, as argued above, all branches originating from this bounded part are eventually pruned. If $d$ encodes a continuous variable, the bounded part behaves as described above with the probability of all branches $\mathcal{B}'_{t'}$ originating from $\mathcal{B}'_t$ decreasing towards zero. Therefore, we only have to show that the probability remaining in the unbounded part decreases towards zero as we continue splitting to show $(*)$. In fact, this follows from the properties of a probability measure.

Above, we have shown that splitting repeatedly either leads to pruning the resulting branches or the probability of all resulting branches decreases towards zero. Therefore, there is a $t'$, such that $(*)$ is satisfied for $\mathcal{B}'_t$. Since there are only finitely many branches in any iteration of PROBBOUNDS, the above implies that Equation (7) is satisfied. In turn, this directly implies that $\mathcal{B}$ is eventually selected by PROBBOUNDS, proving Proposition C.6. $\qquad\square$

Next, we prove a generalised version of Lemma 5.4.

**Lemma C.7** (Converging Probability Bounds). *Let $N \in \mathbb{N}$ be a batch size. Let $\left\{(\ell^{(t)}, u^{(t)})\right\}_{t\in\mathbb{N}}$ be the iterates of* PROBBOUNDS$(\mathbb{P}_{\mathbf{x}}[g_{\mathrm{Sat}}(\mathbf{x}, \mathsf{net}(\mathbf{x})) \geq 0], N)$ *instantiated with a* SELECT *procedure satisfying Definition C.2, a sound* COMPUTEBOUNDS *procedure satisfying Definition C.1 and a* SPLIT *procedure satisfying Definition C.3. Assume* $\mathbb{P}_{\mathbf{x}}[g_{\mathrm{Sat}}(\mathbf{x}, \mathsf{net}(\mathbf{x})) = 0] = 0$ *as in Assumption 5.3. Then,*

$$\lim_{t\to\infty} \ell^{(t)} = \lim_{t\to\infty} u^{(t)} = \mathbb{P}_{\mathbf{x}}[g_{\mathrm{Sat}}(\mathbf{x}, \mathsf{net}(\mathbf{x})) \geq 0].$$

*Proof.* We first prove that $\lim_{t\to\infty} \ell^{(t)} = \mathbb{P}_{\mathbf{x}}[g_{\mathrm{Sat}}(\mathbf{x}, \mathsf{net}(\mathbf{x})) \geq 0]$. The convergence of the upper bound, $\lim_{t\to\infty} u^{(t)} = \mathbb{P}_{\mathbf{x}}[g_{\mathrm{Sat}}(\mathbf{x}, \mathsf{net}(\mathbf{x})) \geq 0]$ follows analogously.

Let $\mathcal{X}^*_{sat} = \{\mathbf{x} \in \mathcal{X} \mid g_{\mathrm{Sat}}(\mathbf{x}, \mathsf{net}(\mathbf{x})) > 0\}$, where $\mathcal{X}$ is the input space of net. Note that $\mathbb{P}_{\mathbf{x}}[\mathcal{X}^*_{sat}] = \mathbb{P}_{\mathbf{x}}[\{\mathbf{x} \in \mathcal{X} \mid g_{\mathrm{Sat}}(\mathbf{x}, \mathsf{net}(\mathbf{x})) \geq 0\}]$ due to Assumption 5.3. Further, let $\hat{\mathcal{X}}^{(t)}_{sat}$ be as in the proof of Theorem 5.1 and recall $\ell^{(t)} = \mathbb{P}_{\mathbf{x}}[\hat{\mathcal{X}}^{(t)}_{sat}]$.

First, we give an argument why the limit $\lim_{t\to\infty} \ell^{(t)}$ exists. Due to Theorem 5.1, the sequence $\{\ell^{(t)}\}_{t\in\mathbb{N}}$ is bounded from above. Furthermore, $\{\ell^{(t)}\}_{t\in\mathbb{N}}$ is non-decreasing in $t$ since PROBBOUNDS only adds elements to $\hat{\mathcal{X}}^{(t)}_{sat}$. Therefore, $\lim_{t\to\infty} \ell^{(t)}$ exists. Now, we can equivalently rewrite

$$\ell^{(t)} \xrightarrow[t\to\infty]{} \mathbb{P}_{\mathbf{x}}[g_{\mathrm{Sat}}(\mathbf{x}, \mathsf{net}(\mathbf{x})) \geq 0]$$

$$\Longleftrightarrow \qquad \mathbb{P}_{\mathbf{x}}\left[\hat{\mathcal{X}}^{(t)}_{sat}\right] \xrightarrow[t\to\infty]{} \mathbb{P}_{\mathbf{x}}[\mathcal{X}^*_{sat}]$$

$$\Longleftrightarrow \quad \mathbb{P}_{\mathbf{x}}\left[\hat{\mathcal{X}}^{(t)}_{sat}\right] - \mathbb{P}_{\mathbf{x}}[\mathcal{X}^*_{sat}] \xrightarrow[t\to\infty]{} 0$$

$$\Longleftrightarrow \qquad \mathbb{P}_{\mathbf{x}}\left[\mathcal{X}^*_{sat} \setminus \hat{\mathcal{X}}^{(t)}_{sat}\right] \xrightarrow[t\to\infty]{} 0. \tag{8}$$

We now argue why (8) holds. Consider the case that the input space does not contain a continuous variable. Let $\mathcal{B}_t$ be a bounded branch in iteration $t \in \mathbb{N}$ for an input space containing only discrete variables. As discussed in the proof of Proposition C.6, there is an iteration $t' > t$ so that all branches $\mathcal{B}_{t'}$ that originate from splitting $\mathcal{B}_t$ are pruned. Due to the properties of a probability measure, the probability of the unbounded branches $\mathcal{B}_t$ that remain decreases towards zero as $t \to \infty$. Therefore, (8) holds if the input space does not contain a continuous variable.

Now, assume the input space contains at least one continuous variable. Let $\tilde{\mathcal{X}}^{(t)}_{sat} = \mathcal{X}^*_{sat} \setminus \hat{\mathcal{X}}^{(t)}_{sat}$. With the goal of obtaining a contradiction, assume $\lim_{t\to\infty} \mathbb{P}_{\mathbf{x}}[\tilde{\mathcal{X}}^{(t)}_{sat}] > 0$. Since we assumed in Section 3 that every continuous probability distribution admits a density function, $\mathrm{vol}(\tilde{\mathcal{X}}^{(t)}_{sat}) > 0$, where vol denotes the volume. Let $t \in \mathbb{N}$. Since the branches maintained by PROBBOUNDS form a partition of the input space, there is a branch $\mathcal{B}_t$ in iteration $t$ of PROBBOUNDS such that $\mathrm{vol}(\tilde{\mathcal{X}}^{(t)}_{sat} \cap \mathcal{B}_t) > 0$.

Due to SPLIT satisfying Definition C.2 and SELECT satisfying Definition C.3, we eventually obtain $\mathcal{B}_{t'}$ in iteration $t' > t$ with $\mathcal{B}_{t'} \subseteq \tilde{\mathcal{X}}_{sat} \cap \mathcal{B}_t$. Additionally, since SPLIT satisfies Definition C.2, we have that $\|\overline{\mathbf{x}} - \underline{\mathbf{x}}\| \to 0$ for all not yet pruned branches $[\underline{\mathbf{x}}, \overline{\mathbf{x}}]$ as $t \to \infty$. Since COMPUTEBOUNDS satisfies Definition C.1, we have that the bounds $\underline{y} \leq g_{\mathrm{Sat}}(\mathbf{x}, \mathsf{net}(\mathbf{x})) \leq \overline{y}$ produced by COMPUTEBOUNDS converge towards $g_{\mathrm{Sat}}(\mathbf{x}, \mathsf{net}(\mathbf{x}))$. Note that $g_{\mathrm{Sat}}(\mathbf{x}, \mathsf{net}(\mathbf{x})) > 0$ for all $\mathbf{x} \in [\underline{\mathbf{x}}, \overline{\mathbf{x}}]$ since $[\underline{\mathbf{x}}, \overline{\mathbf{x}}] \subseteq \mathcal{B}_{t'} \subseteq \mathcal{X}_{sat}^*$. This implies that there is an iteration $t'' > t'$ in which PROBBOUNDS considers a branch $\mathcal{B}_{t''} \subset \tilde{\mathcal{X}}_{sat}$ for which $\underline{y} > 0$, which means that PROBBOUNDS prunes $\mathcal{B}_{t''}$. This contradicts the construction of $\tilde{\mathcal{X}}_{sat}^{(t)}$. With this contradiction, we have shown $\lim_{t \to \infty} \ell^{(t)} = \mathbb{P}_{\mathbf{x}}[g_{\mathrm{Sat}}(\mathbf{x}, \mathsf{net}(\mathbf{x})) \geq 0]$ when the input space contains a continuous variable.

Overall, we have shown $\lim_{t \to \infty} \ell^{(t)} = \mathbb{P}_{\mathbf{x}}[g_{\mathrm{Sat}}(\mathbf{x}, \mathsf{net}(\mathbf{x})) \geq 0]$ for all possible compositions of the input space. The convergence of the upper bound $u^{(t)}$ follows from an analogous argument on $\hbar^{(t)}$ as in the proof of Theorem 5.1 where $u^{(t)} = 1 - \hbar^{(t)}$. This establishes Lemma C.7. $\qquad\square$

Lemma 5.4 follows from Lemma C.7 by inserting SELECTPROB for SELECT, LONGESTEDGE or BABSB-LONGESTEDGE-k for SPLIT, and INTERVALARITHMETIC or CROWN for COMPUTEBOUNDS. SELECTPROB, LONGESTEDGE, and BABSB-LONGESTEDGE-k satisfy the requirements of Lemma C.7 due to Proposition C.6, Proposition C.4 and Corollary C.5, respectively. We show that INTERVALARITHMETIC and CROWN satisfy the requirements of Lemma C.7 in Appendix D and Appendix E, respectively. We now prove a generalised version of Theorem 5.5.

**Theorem C.8** (Completeness). *When instantiated with* PROBBOUNDS *as in Lemma C.7 and a* COMPUTEBOUNDS *procedure that satisfies Definition C.1,* PV *is complete for verification problems satisfying Assumption 5.3.*

*Proof.* Let net, $f_{\mathrm{Sat}}, g_{\mathrm{Sat}}^{(1)}, \ldots, g_{\mathrm{Sat}}^{(v)}$ be as in Equation (3). First, consider $f_{\mathrm{Sat}}(p_1, \ldots, p_v) > 0$. As a consequence of Lemma C.7, the bounds $\ell \leq f_{\mathrm{Sat}}(p_1, \ldots, p_v) \leq u$ produced by COMPUTEBOUNDS converge towards $f_{\mathrm{Sat}}(p_1, \ldots, p_v)$, since COMPUTEBOUNDS satisfies Definition C.1. This implies that eventually $\ell > 0$, meaning that PV eventually proves $f_{\mathrm{Sat}}(p_1, \ldots, p_v) \geq 0$.

If $f_{\mathrm{Sat}}(p_1, \ldots, p_v) < 0$ we eventually obtain $u < 0$ with the same argument as above. Since $u < 0$ disproves $f_{\mathrm{Sat}}(p_1, \ldots, p_v) \geq 0$, PV is complete for probabilistic verification problems satisfying Assumption 5.3. $\qquad\square$

Similarly to Lemma 5.4 and Lemma C.7, Theorem 5.5 follows from Theorem C.8 by inserting INTERVALARITHMETIC or CROWN for COMPUTEBOUNDS.

# D. Interval Arithmetic

This section introduces additional interval arithmetic bounding rules for linear functions, multiplication, and division, complementing the interval arithmetic bounding rules for monotone functions in Section 3.2. Furthermore, we provide Theorems 5.1 and 6.1 Moore et al. (2009) for reference and provide a proof that INTERVALARITHMETIC satisfies Definition C.1. These results provide the foundation for the theoretical analysis of CROWN in Appendix E.

## D.1. Bounding Rules

Let $f^{(k)}$ be as in Section 3.2. First, consider the multiplication of two scalars, that is, $f^{(k)}(z, w) = zw$ where $\underline{z} \leq z \leq \overline{z}$ and $\underline{w} \leq w \leq \overline{w}$. We have

$$\min(\underline{z}\underline{w}, \underline{z}\overline{w}, \overline{z}\underline{w}, \overline{z}\overline{w}) \leq z_1 z_2 \leq \max(\underline{z}\underline{w}, \underline{z}\overline{w}, \overline{z}\underline{w}, \overline{z}\overline{w}).$$

For the element-wise multiplication of vectors, we apply the above rule to each element separately. Multiplication of several arguments can be rewritten as several multiplications of two arguments.

Now, consider computing bounds of the reciprocal $f^{(k)}(z) = \frac{1}{z}$ with $\underline{z} \leq z \leq \overline{z}$. We differentiate the following cases

$$\begin{array}{llll} \dfrac{1}{\overline{z}} \leq \dfrac{1}{z} & \text{if } 0 \notin (\underline{z}, \overline{z}] & \dfrac{1}{z} \leq \dfrac{1}{\underline{z}} & \text{if } 0 \notin [\underline{z}, \overline{z}) \\[2mm] -\infty \leq \dfrac{1}{z} & \text{if } 0 \in (\underline{z}, \overline{z}] & \dfrac{1}{z} \leq \infty & \text{if } 0 \in [\underline{z}, \overline{z}). \end{array}$$

Using bounds on the reciprocal, we can compute bounds on a division by rewriting division as multiplication by the reciprocal. Lastly, for an affine function $f^{(k)}(\mathbf{z}) = \mathbf{W}\mathbf{z} + \mathbf{b}$ where $\underline{\mathbf{z}} \leq \mathbf{z} \leq \overline{\mathbf{z}}$, we have

$$[\mathbf{W}]^{+}\underline{\mathbf{z}} + [\mathbf{W}]^{-}\overline{\mathbf{z}} + \mathbf{b} \leq \mathbf{W}\mathbf{z} + \mathbf{b} \leq [\mathbf{W}]^{+}\overline{\mathbf{z}} + [\mathbf{W}]^{-}\underline{\mathbf{z}} + \mathbf{b}, \tag{9}$$

where $[\mathbf{W}]_{i,j}^{+} = \max(0, \mathbf{W}_{i,j})$ and $[\mathbf{W}]_{i,j}^{-} = \min(0, \mathbf{W}_{i,j})$.

### D.2. Theoretical Properties

We include Theorems 5.1 and 6.1 of Moore et al. (2009) and relevant definitions for reference. Let $\mathbb{H}^n = \{[\underline{\mathbf{x}}, \overline{\mathbf{x}}] \mid \underline{\mathbf{x}}, \overline{\mathbf{x}} \in \mathbb{R}^n, \underline{\mathbf{x}} \leq \overline{\mathbf{x}}\}$ be the set of hyperrectangles in $\mathbb{R}^n$. Let $w : 2^{\mathbb{R}^n} \to \mathbb{R}_{\geq 0}$ with

$$w(\mathcal{X}) = \max_{i \in [n]} \left( \max_{\mathbf{x} \in \mathcal{X}} \mathbf{x}_i - \min_{\mathbf{x} \in \mathcal{X}} \mathbf{x}_i \right) = \max_{\mathbf{x}, \mathbf{x}' \in \mathcal{X}} \|\mathbf{x} - \mathbf{x}'\|_{\infty}$$

be the *width* of the set $\mathcal{X} \subseteq \mathbb{R}^n$. We denote the *image* of a hyperrectangle $[\underline{\mathbf{x}}, \overline{\mathbf{x}}]$ under $f : \mathbb{R}^n \to \mathbb{R}^m$ as $f([\underline{\mathbf{x}}, \overline{\mathbf{x}}]) = \{f(\mathbf{x}) \mid \mathbf{x} \in [\underline{\mathbf{x}}, \overline{\mathbf{x}}]\}$.

#### D.2.1. DEFINITIONS

Theorem 5.1 of Moore et al. (2009) applies to inclusion isotonic interval extensions as defined below.

**Definition D.1.** Let $F : \mathbb{H}^n \to \mathbb{H}^m$ and $f : \mathbb{R}^n \to \mathbb{R}^m$.

- $F$ is an *interval extensions* of $f$ if $\forall \mathbf{x} \in \mathbb{R}^n : F([\mathbf{x}, \mathbf{x}]) = [f(\mathbf{x}), f(\mathbf{x})]$.

- $F$ is *inclusion isotonic* if $\forall [\underline{\mathbf{x}}, \overline{\mathbf{x}}], [\underline{\mathbf{x}}', \overline{\mathbf{x}}'] \in \mathbb{H}^n, [\underline{\mathbf{x}}', \overline{\mathbf{x}}'] \subseteq [\underline{\mathbf{x}}, \overline{\mathbf{x}}] : F([\underline{\mathbf{x}}', \overline{\mathbf{x}}']) \subseteq F([\underline{\mathbf{x}}, \overline{\mathbf{x}}])$.

Theorem 6.1 of Moore et al. (2009) requires an inclusion isotonic *Lipschitz* interval extension.

**Definition D.2.** Let $F : \mathbb{H}^n \to \mathbb{H}^m$ be an interval extension of $f : \mathbb{R}^n \to \mathbb{R}^m$. $F$ is *Lipschitz* if there exists an $L \in \mathbb{R}$ such that $\forall [\underline{\mathbf{x}}, \overline{\mathbf{x}}] \in \mathbb{H}^n : w(F([\underline{\mathbf{x}}, \overline{\mathbf{x}}])) \leq Lw([\underline{\mathbf{x}}, \overline{\mathbf{x}}])$.

INTERVALARITHMETIC, as introduced in Section 3.2, corresponds to *natural interval extensions* in Moore et al. (2009). As Moore et al. (2009) show, natural interval extensions — and, therefore, INTERVALARITHMETIC — satisfy Definitions D.1 and D.2.

#### D.2.2. THEOREMS AND PROPOSITIONS

Theorem 5.1 of Moore et al. (2009) is known as the fundamental theorem of interval analysis. It proves that INTERVALARITHMETIC is sound. We also provide a proof for the well-known property that INTERVALARITHMETIC satisfies Definition C.1. This result is closely related to Theorem 6.1 of Moore et al. (2009), which we also include for reference.

**Theorem D.3** (Theorem 5.1 of Moore et al. (2009)). *If $F : \mathbb{H}^n \to \mathbb{H}^m$ is an inclusion isotonic interval extension of $f : \mathbb{R}^n \to \mathbb{R}^m$, we have $f([\underline{\mathbf{x}}, \overline{\mathbf{x}}]) \subseteq F([\underline{\mathbf{x}}, \overline{\mathbf{x}}])$ for every $[\underline{\mathbf{x}}, \overline{\mathbf{x}}] \in \mathbb{H}^n$.*

**Theorem D.4** (Theorem 6.1 of Moore et al. (2009)). *Let $F : \mathbb{H}^n \to \mathbb{H}^m$ be an inclusion isotonic Lipschitz interval extension of $f : \mathbb{R}^n \to \mathbb{R}^m$. Let $\mathcal{X} = [\underline{\mathbf{x}}, \overline{\mathbf{x}}] \in \mathbb{H}^n$. We define the $M$-step uniform subdivision of $[\underline{\mathbf{x}}, \overline{\mathbf{x}}]$ with $M \in \mathbb{N}$ as*

$$\mathcal{X}_{i,j} = \left[ \underline{\mathbf{x}}_i + (j-1)\frac{w([\underline{\mathbf{x}}_i, \overline{\mathbf{x}}_i])}{M}, \underline{\mathbf{x}}_i + j\frac{w([\underline{\mathbf{x}}_i, \overline{\mathbf{x}}_i])}{M} \right], \quad j \in [M].$$

*Further, let*

$$F^{(M)}([\underline{\mathbf{x}}, \overline{\mathbf{x}}]) = \bigcup_{j_i=1}^{M} F(\mathcal{X}_{1,j_1} \times \cdots \times \mathcal{X}_{n,j_n}).$$

*It holds that*

$$w(F^{(M)}([\underline{\mathbf{x}}, \overline{\mathbf{x}}])) - w(f([\underline{\mathbf{x}}, \overline{\mathbf{x}}])) \leq 2L\frac{w(\mathcal{X})}{M},$$

*where $L$ is the Lipschitz constant of $F$.*

**Proposition D.5.** *Every Lipschitz interval extension satisfies Definition C.1.*

*Proof.* Let $F : \mathbb{H}^n \to \mathbb{H}^m$ be a Lipschitz interval extension with Lipschitz constant $L$. We write $[\underline{\mathbf{y}}_{[\underline{\mathbf{x}},\overline{\mathbf{x}}]}, \overline{\mathbf{y}}_{[\underline{\mathbf{x}},\overline{\mathbf{x}}]}] = F([\underline{\mathbf{x}}, \overline{\mathbf{x}}])$ for $[\underline{\mathbf{x}}, \overline{\mathbf{x}}] \in \mathbb{H}^n$. Using that $F$ is Lipschitz, we obtain

$$\lim_{\|\overline{\mathbf{x}} - \underline{\mathbf{x}}\| \to 0} \|\overline{\mathbf{y}}_{[\underline{\mathbf{x}},\overline{\mathbf{x}}]} - \underline{\mathbf{y}}_{[\underline{\mathbf{x}},\overline{\mathbf{x}}]}\| = \lim_{\|\overline{\mathbf{x}} - \underline{\mathbf{x}}\| \to 0} w(F([\underline{\mathbf{x}}, \overline{\mathbf{x}}])) \leq \lim_{\|\overline{\mathbf{x}} - \underline{\mathbf{x}}\| \to 0} L w([\underline{\mathbf{x}}, \overline{\mathbf{x}}]) = 0.$$

Further, since $F$ is an interval extension, we have $\|\overline{\mathbf{y}}_{[\mathbf{x},\mathbf{x}]} - \underline{\mathbf{y}}_{[\mathbf{x},\mathbf{x}]}\| = 0$ for any $\mathbf{x} \in \mathbb{R}^n$. This proves that $F$ satisfies Definition C.1. $\qquad\square$

**Corollary D.6.** INTERVALARITHMETIC *satisfies Definition C.1.*

# E. Theoretical Analysis of CROWN

In this section, we prove that CROWN (Zhang et al., 2018) satisfies Definition C.1. In fact, we provide a slightly stronger result showing that CROWN possesses the properties of INTERVALARITHMETIC that follow from Theorem 5.1 and 6.1 of Moore et al. (2009). Concretely, we prove that the bounds computed by CROWN are always contained in the bounds computed by an inclusion isotonic Lipschitz interval extension, such that Proposition D.5 and Theorems D.3 and D.4 apply. Since the bounds computed by this interval extension converge and they contain the bounds computed by CROWN, the bounds computed by CROWN also need to converge, establishing that CROWN satisfies Definition C.1. We first revisit CROWN, discuss why CROWN itself is not inclusion isotonic, and finally provide the inclusion isotonic Lipschitz interval extension that is guaranteed to compute *looser* bounds than CROWN.

In the following, we restrict our attention to ReLU-activated fully-connected neural networks since these are the most relevant for this paper. However, similar arguments to the arguments we make below can also be made for other architectures and activation functions.

## E.1. CROWN

We revisit CROWN for ReLU-activated fully-connected neural networks. Applying CROWN for other activation functions and architectures is described by Zhang et al. (2018); Xu et al. (2020). Let net : $\mathcal{X} \to \mathbb{R}^m$ be a ReLU-activated fully-connected neural network with $K \in \mathbb{N}$ layers, that is, net $= f^{(K)} \circ \ldots \circ f^{(1)}$, where $f^{(k)} : \mathbb{R}^{n_k} \to \mathbb{R}^{n_{k+1}}$ is either an affine function or ReLU. We use $f^{(:k)} = f^{(k)} \circ \cdots \circ f^{(1)}$ and $f^{(k:)} = f^{(K)} \circ \cdots \circ f^{(k)}$ to denote partial evaluation of net. Further, let $[\underline{\mathbf{x}}, \overline{\mathbf{x}}] \subseteq \mathcal{X}$. CROWN computes a linear lower bound and a linear upper bound on net

$$\underline{\mathbf{A}}\mathbf{x} + \underline{\mathbf{d}} \leq \mathsf{net}(\mathbf{x}) \leq \overline{\mathbf{A}}\mathbf{x} + \overline{\mathbf{d}}, \qquad \forall \mathbf{x} \in [\underline{\mathbf{x}}, \overline{\mathbf{x}}].$$

These linear bounds can be *concretised* into constant bounds as

$$\underline{\mathbf{y}}_{\text{CROWN}} = [\underline{\mathbf{A}}]^+ \underline{\mathbf{x}} + [\underline{\mathbf{A}}]^- \overline{\mathbf{x}} + \underline{\mathbf{d}} \leq \mathsf{net}(\mathbf{x}) \leq [\overline{\mathbf{A}}]^+ \overline{\mathbf{x}} + [\overline{\mathbf{A}}]^- \underline{\mathbf{x}} + \overline{\mathbf{d}} = \overline{\mathbf{y}}_{\text{CROWN}},$$

where $\underline{\mathbf{x}}, \overline{\mathbf{x}}$ are bounds on the input, $\mathbf{x} \in [\underline{\mathbf{x}}, \overline{\mathbf{x}}]$, $[\mathbf{A}]^+_{i,j} = \max(0, \mathbf{A}_{i,j})$, and $[\mathbf{A}]^-_{i,j} = \min(0, \mathbf{A}_{i,j})$.

CROWN computes the linear bounds by a backwards walk over the network net, starting from $f^{(K)}$ and propagating linear bounds backwards until reaching $f^{(1)}$. Algorithm 3 defines CROWN for feed-forward neural networks, such as net. The algorithm iteratively computes linear bounds

$$\underline{\mathbf{A}}^{(k)}\mathbf{z}^{(k)} + \underline{\mathbf{d}}^{(k)} \leq f^{(k+1:)}(\mathbf{z}^{(k)}) \leq \overline{\mathbf{A}}^{(k)}\mathbf{z}^{(k)} + \overline{\mathbf{d}}^{(k)}, \qquad \forall \mathbf{z}^{(k)} \in [\underline{\mathbf{z}}^{(k)}, \overline{\mathbf{z}}^{(k)}],$$

where $[\underline{\mathbf{z}}^{(k)}, \overline{\mathbf{z}}^{(k)}]$ are bounds on $f^{(:k)}(\mathbf{x})$ for $\mathbf{x} \in [\underline{\mathbf{x}}, \overline{\mathbf{x}}]$ that are computed before invoking CROWN, for example, using INTERVALARITHMETIC (Zhang et al., 2020). The bounds $\underline{\mathbf{z}}^{(k)}, \overline{\mathbf{z}}^{(k)}$ are sometimes known as pre-activation bounds. Alternatively to using INTERVALARITHMETIC, we can compute $\underline{\mathbf{z}}^{(k)}, \overline{\mathbf{z}}^{(k)}$ by invoking Algorithm 3 iteratively for $f^{(:1)}, \ldots, f^{(:K)}$ and use the concretised bounds on $f^{(:1)}, \ldots, f^{(:k-1)}$ as the layer bounds for the $k$-th invocation of Algorithm 3 (Zhang et al., 2018). Algorithm 4 describes this approach in more detail. Applying Algorithm 3 with the layer bounds computed by INTERVALARITHMETIC is known as CROWN-IBP (Zhang et al., 2020), while using Algorithm 4 is referred to as just CROWN. To avoid confusion, in this section, we use CROWN to only refer to Algorithm 3.

---

**Algorithm 3** CROWN

---

**Require:** Neural Network net $= f^{(K)} \circ \cdots \circ f^{(1)}$, Input Bounds $[\underline{\mathbf{x}}, \overline{\mathbf{x}}]$, Layer Bounds $[\underline{\mathbf{z}}^{(1)}, \overline{\mathbf{z}}^{(1)}], \ldots, [\underline{\mathbf{z}}^{(K-1)}, \overline{\mathbf{z}}^{(K-1)}]$

1: $\underline{\mathbf{A}}^{(K)} \leftarrow \mathbf{I}, \overline{\mathbf{A}}^{(K)} \leftarrow \mathbf{I}$ // $\mathbf{I} \in \mathbb{R}^{m \times m}$ *is the identity matrix*
2: $\underline{\mathbf{d}}^{(K)} \leftarrow \mathbf{0}, \overline{\mathbf{d}}^{(K)} \leftarrow \mathbf{0}$ // $\mathbf{0} \in \mathbb{R}^m$ *is the all-zero vector*
3: $\underline{\mathbf{z}}^{(0)} \leftarrow \underline{\mathbf{x}}, \overline{\mathbf{z}}^{(0)} \leftarrow \overline{\mathbf{x}}$
4: **for** $k \in \{K, \ldots, 1\}$ **do**
5: $\quad (\underline{\mathbf{A}}^{(k-1)}, \overline{\mathbf{A}}^{(k-1)}, \underline{\boldsymbol{\Delta}}, \overline{\boldsymbol{\Delta}}) \leftarrow \text{CROWNRULE}(f^{(k)}, \underline{\mathbf{A}}^{(k)}, \overline{\mathbf{A}}^{(k)}, \underline{\mathbf{z}}^{(k-1)}, \overline{\mathbf{z}}^{(k-1)})$
6: $\quad \underline{\mathbf{d}}^{(k-1)} \leftarrow \underline{\mathbf{d}}^{(k)} + \underline{\boldsymbol{\Delta}}$
7: $\quad \overline{\mathbf{d}}^{(k-1)} \leftarrow \overline{\mathbf{d}}^{(k)} + \overline{\boldsymbol{\Delta}}$
8: **end for**
9: **return** $(\underline{\mathbf{A}}^{(0)}, \overline{\mathbf{A}}^{(0)}, \underline{\mathbf{d}}^{(0)}, \overline{\mathbf{d}}^{(0)})$

---

**Algorithm 4** CROWN with CROWN Layer Bounds

---

**Require:** Neural Network net $= f^{(K)} \circ \cdots \circ f^{(1)}$, Input Bounds $[\underline{\mathbf{x}}, \overline{\mathbf{x}}]$

1: **for** $k \in [K]$ **do** // *Below,* CROWN *refers to Algorithm 3*
2: $\quad [\underline{\mathbf{z}}^{(k)}, \overline{\mathbf{z}}^{(k)}] \leftarrow \text{CONCRETISE}(\text{CROWN}(f^{(:k)}, [\underline{\mathbf{x}}, \overline{\mathbf{x}}], [\underline{\mathbf{z}}^{(1)}, \overline{\mathbf{z}}^{(1)}], \ldots, [\underline{\mathbf{z}}^{k-1}, \overline{\mathbf{z}}^{k-1}]))$
3: **end for**
4: **return** $\text{CROWN}(\text{net}, [\underline{\mathbf{x}}, \overline{\mathbf{x}}], [\underline{\mathbf{z}}^{(1)}, \overline{\mathbf{z}}^{(1)}], \ldots, [\underline{\mathbf{z}}^{K-1}, \overline{\mathbf{z}}^{K-1}])$

---

Similarly to INTERVALARITHMETIC, Algorithm 3 uses CROWNRULEs to propagate linear bounds through more fundamental functions, such as affine layers or ReLU layers. Specifically, the CROWNRULE for an affine function $f^{(k)}(\mathbf{z}) = \mathbf{W}\mathbf{z} + \mathbf{b}$ computes

$$\underline{\mathbf{A}}^{(k-1)} = \underline{\mathbf{A}}^{(k)}\mathbf{W}, \qquad \overline{\mathbf{A}}^{(k-1)} = \overline{\mathbf{A}}^{(k)}\mathbf{W}, \qquad \underline{\boldsymbol{\Delta}}^{(k-1)} = \underline{\mathbf{A}}^{(k)}\mathbf{b}, \qquad \overline{\boldsymbol{\Delta}}^{(k-1)} = \overline{\mathbf{A}}^{(k)}\mathbf{b}. \tag{10}$$

The CROWNRULE for a ReLU layer $f^{(k)}(\mathbf{z}) = [\mathbf{z}]^+$ is

$$\begin{aligned}
\underline{\mathbf{A}}^{(k-1)} &= \left[\underline{\mathbf{A}}^{(k)}\right]^+ \text{diag}(\underline{\boldsymbol{\alpha}}) + \left[\underline{\mathbf{A}}^{(k)}\right]^- \text{diag}(\overline{\boldsymbol{\alpha}}), \quad \underline{\boldsymbol{\Delta}}^{(k-1)} = \left[\underline{\mathbf{A}}^{(k)}\right]^- \overline{\boldsymbol{\beta}}, \\
\overline{\mathbf{A}}^{(k-1)} &= \left[\overline{\mathbf{A}}^{(k)}\right]^+ \text{diag}(\overline{\boldsymbol{\alpha}}) + \left[\overline{\mathbf{A}}^{(k)}\right]^- \text{diag}(\underline{\boldsymbol{\alpha}}), \quad \overline{\boldsymbol{\Delta}}^{(k-1)} = \left[\overline{\mathbf{A}}^{(k)}\right]^+ \overline{\boldsymbol{\beta}},
\end{aligned} \tag{11}$$

where $\text{diag}(\boldsymbol{\alpha})$ is a diagonal matrix with the vector $\boldsymbol{\alpha}$ on its diagonal and $\underline{\boldsymbol{\alpha}}, \overline{\boldsymbol{\alpha}}, \overline{\boldsymbol{\beta}} \in \mathbb{R}^{n_k}$ have

$$\overline{\boldsymbol{\beta}}_i = \begin{cases} 0 & \overline{\mathbf{z}}_i^{(k-1)} \leq 0 \\ 1 & \underline{\mathbf{z}}_i^{(k-1)} \geq 0 \\ \frac{\overline{\mathbf{z}}_i^{(k-1)}}{\overline{\mathbf{z}}_i^{(k-1)} - \underline{\mathbf{z}}_i^{(k-1)}} & \text{otherwise,} \end{cases} \qquad \overline{\boldsymbol{\beta}}_i = \begin{cases} 0 & 0 \notin (\underline{\mathbf{z}}_i^{(k-1)}, \overline{\mathbf{z}}_i^{(k-1)}) \\ -\underline{\mathbf{z}}_i^{(k-1)} \overline{\boldsymbol{\alpha}}_i & \text{otherwise,} \end{cases}$$

$$\underline{\boldsymbol{\alpha}}_i = \begin{cases} 0 & \overline{\mathbf{z}}_i^{(k-1)} \leq 0 \\ 1 & \underline{\mathbf{z}}_i^{(k-1)} \geq 0 \\ 0 & \underline{\mathbf{z}}_i^{(k-1)} < 0 < \overline{\mathbf{z}}_i^{(k-1)} \text{ and } |\underline{\mathbf{z}}_i^{(k-1)}| \geq |\overline{\mathbf{z}}_i^{(k-1)}| \\ 1 & \text{otherwise,} \end{cases}$$

for every $i \in [n_k]$. Instead of the last two cases for $\underline{\boldsymbol{\alpha}}_i$, we can also optimise each $\underline{\boldsymbol{\alpha}}_i$ using gradient descent to improve the linear bounds on the network (Xu et al., 2021), as long as we maintain $\underline{\boldsymbol{\alpha}}_i \in [0, 1]$.

**CROWN is not Inclusion-Isotonic.** We demonstrate that CROWN is not inclusion-isotonic according to Definition D.1 using an example. Consider a single ReLU neuron $[x]^+$ with input bounds $[\underline{x}, \overline{x}] = [-3, 2]$. The CROWN bounds are $0 \leq [x]^+ \leq \frac{2}{5}x + \frac{6}{5}$ so that the concretised lower bound $\underline{y}_{\text{CROWN}} = 0$. However, the CROWN bounds for $[\underline{x}', \overline{x}'] = [-1, 2] \subset [\underline{x}, \overline{x}]$ are $x \leq [x]^+ \leq \frac{2}{3}x + \frac{2}{3}$ which yields the concretised lower bound $\underline{y}'_{\text{CROWN}} = -1$. Since $\underline{y}'_{\text{CROWN}} < \underline{y}_{\text{CROWN}}$ although $[\underline{x}', \overline{x}'] \subset [\underline{x}, \overline{x}]$, CROWN is not inclusion-isotonic.

## E.2. CROWN Interval Extension

In this section, we introduce the CROWN INTERVAL EXTENSION (CIE). CIE is a theoretical device that we use to prove that the width of the concretised CROWN bounds linearly converges to zero as the width of the input bounds converges to zero, analogously to Theorem D.4. Concretely, CIE is an inclusion-isotonic Lipschitz interval extension according to Definition D.1 and Definition D.2 that is guaranteed to contain the concretised CROWN bounds. Since Theorem D.4 applies to CIE and the concretised CROWN bounds are always contained in the CIE bounds, the convergence properties of Theorem D.4 also apply to CROWN. In the following, we first define CIE, show that it is an inclusion-isotonic Lipschitz interval extension, show that the bounds computed by INTERVALARITHMETIC are contained in the bounds computed by CIE, and finally prove that the concretised CROWN bounds are contained in the bounds computed by CIE.

**CIE Algorithm.** To define CIE, we define a bounding rule for ReLU. We use the same bounding rule for affine functions as INTERVALARITHMETIC, as introduced in Appendix D.1. Similarly to INTERVALARITHMETIC, CIE then computes bounds on a ReLU-activated feed-forward neural network $\mathsf{net} = f^{(K)} \circ \cdots \circ f^{(1)}$ by computing $(F_{\mathrm{CIE}}^{(K)} \circ \cdots \circ F_{\mathrm{CIE}}^{(1)})(\underline{\mathbf{x}}, \overline{\mathbf{x}})$, where $[\underline{\mathbf{x}}, \overline{\mathbf{x}}] \subseteq \mathcal{X}$ and $F_{\mathrm{CIE}}^{(k)} : \mathbb{R}^{n_k} \times \mathbb{R}^{n_k} \to \mathbb{R}^{n_k} \times \mathbb{R}^{n_k}$ is the CIE bounding rule for $f^{(k)}$. The CIE bounding rule for ReLU layers $f^{(k)}(\mathbf{v}) = [\mathbf{v}]^+$ is $F^{(k)}([\underline{\mathbf{v}}, \overline{\mathbf{v}}]) = [\underline{\mathbf{w}}, \overline{\mathbf{w}}]$, where

$$\underline{\mathbf{w}}_i = \begin{cases} 0 & \overline{\mathbf{v}}_i \leq 0 \\ \underline{\mathbf{v}}_i & \text{otherwise,} \end{cases} \qquad \overline{\mathbf{w}}_i = \begin{cases} 0 & \overline{\mathbf{v}}_i \leq 0 \\ \overline{\mathbf{v}}_i & \text{otherwise,} \end{cases} \tag{12}$$

where $i \in [n_k]$. Comparing this bounding rule to the CROWNRULE for ReLU, we can see that the bounds computed by CIE always contain the concretised CROWN bounds for a ReLU neuron. The remainder of this section aims to show that this is also the case for a complete neural network.

**Proposition E.1.** *Let* $\mathsf{net} = f^{(K)} \circ \cdots \circ f^{(1)}$ *be a ReLU-activated fully-connected neural network. The* CIE *interval function* $F_{\mathrm{CIE}}^{(K)} \circ \cdots \circ F_{\mathrm{CIE}}^{(1)}$ *is a inclusion-isotonic Lipschitz interval extension of* $\mathsf{net}$.

*Proof.* We first show that the CIE bounding rule for ReLU is an inclusion-isotonic Lipschitz interval extension of ReLU. Let $F^{(k)}$ be the CIE bounding rule for the ReLU layer $f^{(k)} : \mathbb{R}^{(n_k)} \to \mathbb{R}^{(n_k)}$ according to Equation (12).

- *Interval extension.* Let $\mathbf{v} \in \mathbb{R}^{n_k}$ and $[\underline{\mathbf{w}}, \overline{\mathbf{w}}] = F^{(k)}([\mathbf{v}, \mathbf{v}])$. Let $i \in [n_k]$. If $\mathbf{v}_i \leq 0$, we have $\underline{\mathbf{w}}_i = \overline{\mathbf{w}} = 0 = [\mathbf{v}]_i^+$. Otherwise, if $\mathbf{v}_i > 0$, we have $\underline{\mathbf{w}}_i = \overline{\mathbf{w}}_i = \mathbf{v}_i = [\mathbf{v}]_i^+$.

- *Inclusion-isotonic.* Let $[\underline{\mathbf{v}}', \overline{\mathbf{v}}'] \subseteq [\underline{\mathbf{v}}, \overline{\mathbf{v}}] \subseteq \mathbb{R}^{n_k}$, $[\underline{\mathbf{w}}', \overline{\mathbf{w}}'] = F^{(k)}([\underline{\mathbf{v}}', \overline{\mathbf{v}}'])$ and $[\underline{\mathbf{w}}, \overline{\mathbf{w}}] = F^{(k)}([\underline{\mathbf{v}}, \overline{\mathbf{v}}])$. Let $i \in [n_k]$. If $\overline{\mathbf{v}}_i \leq 0$, we have $[\underline{\mathbf{w}}_i, \overline{\mathbf{w}}_i] = [0, 0] = [\underline{\mathbf{w}}'_i, \overline{\mathbf{w}}'_i]$ since $\overline{\mathbf{v}}'_i \leq \overline{\mathbf{v}}_i$. On the other hand, if $\overline{\mathbf{v}}_i > 0$, we have $[\underline{\mathbf{w}}_i, \overline{\mathbf{w}}_i] = [\underline{\mathbf{v}}_i, \overline{\mathbf{v}}_i]$. If $\overline{\mathbf{v}}'_i \leq 0$, $[\underline{\mathbf{w}}'_i, \overline{\mathbf{w}}'_i] = [0, 0] \subseteq [\underline{\mathbf{w}}_i, \overline{\mathbf{w}}_i]$ since $\underline{\mathbf{v}}_i \leq \overline{\mathbf{v}}'_i < \overline{\mathbf{v}}$. Otherwise, $[\underline{\mathbf{w}}'_i, \overline{\mathbf{w}}'_i] = [\underline{\mathbf{v}}'_i, \overline{\mathbf{v}}'_i] \subseteq [\underline{\mathbf{w}}_i, \overline{\mathbf{w}}_i]$. Therefore, $F^{(k)}$ is inclusion-isotonic.

- *Lipschitz.* Let $[\underline{\mathbf{v}}, \overline{\mathbf{v}}] \subseteq \mathbb{R}^{n_k}$ and $[\underline{\mathbf{w}}, \overline{\mathbf{w}}] = F^{(k)}([\underline{\mathbf{v}}, \overline{\mathbf{v}}])$. For all $i \in [n_k]$ with $\overline{\mathbf{w}}_i \leq 0$ we have $w([\underline{\mathbf{w}}_i, \overline{\mathbf{w}}_i]) = 0$ and, therefore, trivially $w([\underline{\mathbf{w}}_i, \overline{\mathbf{w}}_i]) \leq w([\underline{\mathbf{v}}, \overline{\mathbf{v}}])$. For $i \in [n_k]$ with $\overline{\mathbf{w}}_i > 0$, we have $[\underline{\mathbf{w}}_i, \overline{\mathbf{w}}_i] = [\underline{\mathbf{v}}_i, \overline{\mathbf{v}}_i]$. Overall, $w([\underline{\mathbf{w}}, \overline{\mathbf{w}}]) = \max_{i \in [n_k]} \overline{\mathbf{w}}_i - \underline{\mathbf{w}}_i \leq \max_{i \in [n_k]} \overline{\mathbf{v}}_i - \underline{\mathbf{v}}_i = w([\underline{\mathbf{v}}, \overline{\mathbf{v}}])$. Therefore, $F^{(k)}$ is Lipschitz with Lipschitz constant 1.

For affine functions, CIE uses the same bounding rule as INTERVALARITHMETIC that is an inclusion-isotonic Lipschitz interval extension (Moore et al., 2009). Lemma 6.3 of Moore et al. (2009) now gives us that the composition $F_{\mathrm{CIE}}^{(K)} \circ \cdots \circ F_{\mathrm{CIE}}^{(1)}$ of inclusion-isotonic Lipschitz interval extensions is also inclusion inclusion-isotonic and Lipschitz. Clearly, a composition of interval extensions is also an interval extension. $\square$

**Proposition E.2.** *Let* $\mathsf{net} = f^{(K)} \circ \cdots \circ f^{(1)}$ *be a ReLU-activated fully-connected neural network. Let* $[\underline{\mathbf{y}}, \overline{\mathbf{y}}]$ *and* $[\underline{\mathbf{w}}, \overline{\mathbf{w}}]$ *be the bounds on* $\mathsf{net}(\mathbf{x})$ *for* $\mathbf{x} \in [\underline{\mathbf{x}}, \overline{\mathbf{x}}]$ *computed by* INTERVALARITHMETIC *and* CIE*, respectively. It holds that* $[\underline{\mathbf{y}}, \overline{\mathbf{y}}] \subseteq [\underline{\mathbf{w}}, \overline{\mathbf{w}}]$.

*Proof.* Let $\mathsf{net} = f^{(K)} \circ \cdots \circ f^{(1)}$ and $[\underline{\mathbf{x}}, \overline{\mathbf{x}}]$ be as in Proposition E.2. We prove Proposition E.2 by finite induction over $k \in \{0, \ldots, K\}$. Let $[\underline{\mathbf{y}}^{(k)}, \overline{\mathbf{y}}^{(k)}]$ and $[\underline{\mathbf{v}}^{(k)}, \overline{\mathbf{v}}^{(k)}]$ be the bounds that INTERVALARITHMETIC and CIE respectively compute on $f^{(:k)}(\mathbf{x})$ for $\mathbf{x} \in [\underline{\mathbf{x}}, \overline{\mathbf{x}}]$.

*Induction start.* For $k = 0$, we consider the identity function $\mathrm{net}(\mathbf{x}) = \mathbf{x}$, for which both CIE and INTERVALARITHMETIC compute $[\underline{\mathbf{y}}^{(0)}, \overline{\mathbf{y}}^{(0)}] = [\underline{\mathbf{v}}^{(0)}, \overline{\mathbf{v}}^{(0)}] = [\underline{\mathbf{x}}, \overline{\mathbf{x}}]$.

*Induction step.* Now, let $k \in [K]$ and assume $[\underline{\mathbf{y}}^{(k-1)}, \overline{\mathbf{y}}^{(k-1)}] \subseteq [\underline{\mathbf{v}}^{(k-1)}, \overline{\mathbf{v}}^{(k-1)}]$. We show $[\underline{\mathbf{y}}^{(k)}, \overline{\mathbf{y}}^{(k)}] \subseteq [\underline{\mathbf{v}}^{(k)}, \overline{\mathbf{v}}^{(k)}]$. We differentiate two cases:

- If $f^{(k)}$ is an affine function, both INTERVALARITHMETIC and CIE use the INTERVALARITHMETIC bounding rule for affine functions from Appendix D.1 to compute $[\underline{\mathbf{y}}^{(k)}, \overline{\mathbf{y}}^{(k)}]$ and $[\underline{\mathbf{v}}^{(k)}, \overline{\mathbf{v}}^{(k)}]$. Since this bounding rule is inclusion-isotonic (Moore et al., 2009), we have $[\underline{\mathbf{y}}^{(k)}, \overline{\mathbf{y}}^{(k)}] \subseteq [\underline{\mathbf{v}}^{(k)}, \overline{\mathbf{v}}^{(k)}]$.

- Otherwise, $f^{(k)}$ is ReLU. The INTERVALARITHMETIC bounding rule for ReLU computes $[\underline{\mathbf{y}}^{(k)}, \overline{\mathbf{y}}^{(k)}] = \left[ [\underline{\mathbf{y}}^{(k-1)}]^+, [\overline{\mathbf{y}}^{(k-1)}]^+ \right]$. Since the CIE rule for ReLU is an inclusion-isotonic interval extension (Proposition E.1), it follows that $\left[ [\underline{\mathbf{y}}^{(k-1)}]^+, [\overline{\mathbf{y}}^{(k-1)}]^+ \right] \subseteq [\underline{\mathbf{v}}^{(k)}, \overline{\mathbf{v}}^{(k)}]$.

Overall, this shows that $[\underline{\mathbf{y}}, \overline{\mathbf{y}}] = [\underline{\mathbf{y}}^{(K)}, \overline{\mathbf{y}}^{(K)}] \subseteq [\underline{\mathbf{v}}^{(K)}, \overline{\mathbf{v}}^{(K)}] = [\underline{\mathbf{v}}, \overline{\mathbf{v}}]$. $\qquad\square$

**Proposition E.3.** *Let* $\mathrm{net} = f^{(K)} \circ \cdots \circ f^{(1)}$ *be a ReLU-activated fully-connected neural network, let* $[\underline{\mathbf{x}}, \overline{\mathbf{x}}] \subseteq \mathbb{R}^n$ *and let* $[\underline{\mathbf{v}}^{(k)}, \overline{\mathbf{v}}^{(k)}]$ *be the* CIE *bounds on* $f^{(:k)}(\mathbf{x})$ *for* $\mathbf{x} \in [\underline{\mathbf{x}}, \overline{\mathbf{x}}]$, $\forall k \in [K]$. *Further, let* $[\underline{\mathbf{z}}^{(k)}, \overline{\mathbf{z}}^{(k)}]$ *also be bounds on* $f^{(:k)}(\mathbf{x})$ *for* $\mathbf{x} \in [\underline{\mathbf{x}}, \overline{\mathbf{x}}]$ *with* $[\underline{\mathbf{z}}^{(k)}, \overline{\mathbf{z}}^{(k)}] \subseteq [\underline{\mathbf{v}}^{(k)}, \overline{\mathbf{v}}^{(k)}]$, $\forall k \in [K]$. *It holds that*

$$\underline{\mathbf{v}}^{(K)} \le \underline{\mathbf{A}}\mathbf{x} + \underline{\mathbf{d}} \quad and \quad \overline{\mathbf{A}}\mathbf{x} + \overline{\mathbf{d}} \le \overline{\mathbf{v}}^{(K)}, \qquad \forall \mathbf{x} \in [\underline{\mathbf{x}}, \overline{\mathbf{x}}],$$

*where* $\underline{\mathbf{A}}\mathbf{x} + \underline{\mathbf{d}}$ *and* $\overline{\mathbf{A}}\mathbf{x} + \overline{\mathbf{d}}$ *are the bounds on* $\mathrm{net}(\mathbf{x})$ *for* $\mathbf{x} \in [\underline{\mathbf{x}}, \overline{\mathbf{x}}]$ *computed by* CROWN *(Algorithm 3) using the layer bounds* $[\underline{\mathbf{z}}^{(1)}, \overline{\mathbf{z}}^{(1)}], \ldots, [\underline{\mathbf{z}}^{(K-1)}, \overline{\mathbf{z}}^{(K-1)}]$.

*Proof.* Let $\mathrm{net} = f^{(K)} \circ \cdots \circ f^{(1)}$, $[\underline{\mathbf{x}}, \overline{\mathbf{x}}]$, $[\underline{\mathbf{z}}^{(k)}, \overline{\mathbf{z}}^{(k)}]$, and $[\underline{\mathbf{v}}^{(k)}, \overline{\mathbf{v}}^{(k)}]$ for all $k \in [K]$ be as in Proposition E.3. Let $\underline{\mathbf{A}}^{(k)}, \overline{\mathbf{A}}^{(k)}, \underline{\mathbf{d}}^{(k)}, \overline{\mathbf{d}}^{(k)}$ for all $k \in \{0, \ldots, K\}$ be as in Algorithm 3 for the inputs $\mathrm{net}$, $[\underline{\mathbf{x}}, \overline{\mathbf{x}}]$ and $[\underline{\mathbf{z}}^{(1)}, \overline{\mathbf{z}}^{(1)}], \ldots, [\underline{\mathbf{z}}^{(K)}, \overline{\mathbf{z}}^{(K)}]$. Further, let $[\underline{\mathbf{z}}^{(0)}, \overline{\mathbf{z}}^{(0)}] = [\underline{\mathbf{v}}^{(0)}, \overline{\mathbf{v}}^{(0)}] = [\underline{\mathbf{x}}, \overline{\mathbf{x}}]$. In the following, we prove

$$\underline{\mathbf{A}}^{(k)}\mathbf{z}^{(k)} + \underline{\mathbf{d}}^{(k)} \ge \underline{\mathbf{v}}^{(K)} \quad and \quad \overline{\mathbf{A}}^{(k)}\mathbf{z}^{(k)} + \overline{\mathbf{d}}^{(k)} \le \overline{\mathbf{v}}^{(K)}, \qquad \forall \mathbf{z}^{(k)} \in [\underline{\mathbf{v}}^{(k)}, \overline{\mathbf{v}}^{(k)}], \tag{13}$$

for every $k \in \{0, \ldots, K\}$. Note that $[\underline{\mathbf{z}}^{(k)}, \overline{\mathbf{z}}^{(k)}] \subseteq [\underline{\mathbf{v}}^{(k)}, \overline{\mathbf{v}}^{(k)}]$ by assumption in Proposition E.3. Proving Equation (13) also proves Proposition E.3 when $k = 0$. We proceed by finite induction over $k$ from $K$ to 0, following the backwards walk performed by Algorithm 3.

*Induction start.* We consider $k = K$. In this case, $\underline{\mathbf{A}}^{(K)} = \overline{\mathbf{A}}^{(K)} = \mathbf{I}$ and $\underline{\mathbf{d}}^{(K)} = \overline{\mathbf{d}}^{(K)} = \mathbf{0}$, where $\mathbf{I} \in \mathbb{R}^{n_K \times n_K}$ is the identity matrix and $\mathbf{0} \in \mathbb{R}^{n_K}$ is the all-zero vector. Therefore, we have

$$\underline{\mathbf{A}}^{(K)}\mathbf{z}^{(K)} + \underline{\mathbf{d}}^{(K)} = \mathbf{I}\mathbf{z}^{(K)} + \mathbf{0} \ge \underline{\mathbf{v}}^{(K)}$$
$$\overline{\mathbf{A}}^{(K)}\mathbf{z}^{(K)} + \overline{\mathbf{d}}^{(K)} = \mathbf{I}\mathbf{z}^{(K)} + \mathbf{0} \le \overline{\mathbf{v}}^{(K)},$$

for $\mathbf{z}^{(K)} \in [\underline{\mathbf{v}}^{(K)}, \overline{\mathbf{v}}^{(K)}]$. This proves Equation (13) for $k = K$.

*Induction step.* Let $k \in [K]$ and assume Equation (13) holds for $k$. We show that it also holds for $k - 1$. We differentiate two cases, based on whether $f^{(k)}$ is a affine function or ReLU.

- *Affine function.* If $f^{(k)}(\mathbf{z}) = \mathbf{W}\mathbf{z} + \mathbf{b}$, CROWNRULE computes the linear bounds on $f^{(k:)}$ as in Equation (10), so that we obtain
$$\underline{\mathbf{A}}^{(k-1)}\mathbf{z} + \underline{\mathbf{d}}^{(k-1)} = \underline{\mathbf{A}}^{(k)}\mathbf{W}\mathbf{z} + \underline{\mathbf{A}}^{(k)}\mathbf{b} + \underline{\mathbf{d}}^{(k)} = \underline{\mathbf{A}}^{(k)}(\mathbf{W}\mathbf{z} + \mathbf{b}) + \underline{\mathbf{d}}^{(k)} \ge \underline{\mathbf{v}}^{(K)},$$
where $\mathbf{z} \in [\underline{\mathbf{v}}^{(k-1)}, \overline{\mathbf{v}}^{(k-1)}]$. The final inequality is due to the induction assumption that Equation (13) holds for $k$, which we can apply since $\mathbf{W}\mathbf{z} + \mathbf{b} \in [\underline{\mathbf{v}}^{(k)}, \overline{\mathbf{v}}^{(k)}]$, since CIE computes $[\underline{\mathbf{v}}^{(k)}, \overline{\mathbf{v}}^{(k)}]$ from $[\underline{\mathbf{v}}^{(k-1)}, \overline{\mathbf{v}}^{(k-1)}]$ using Equation (9). Similarly, we obtain
$$\overline{\mathbf{A}}^{(k-1)}\mathbf{z} + \overline{\mathbf{d}}^{(k-1)} = \overline{\mathbf{A}}^{(k)}\mathbf{W}\mathbf{z} + \overline{\mathbf{A}}^{(k)}\mathbf{b} + \overline{\mathbf{d}}^{(k)} = \overline{\mathbf{A}}^{(k)}(\mathbf{W}\mathbf{z} + \mathbf{b}) + \overline{\mathbf{d}}^{(k)} \le \overline{\mathbf{v}}^{(K)},$$
again by using the induction assumption.

- *ReLU.* Let $\mathbf{z} \in [\underline{\mathbf{v}}^{(k-1)}, \overline{\mathbf{v}}^{(k-1)}]$ and $i \in [n_k]$. CROWN computes the linear bounds on $f^{(k:)}$ according to Equation (11), which yields

$$\underline{\mathbf{A}}_{:,i}^{(k-1)}\mathbf{z}_i + \underline{\mathbf{d}}_i^{(k-1)} = \left(\left[\underline{\mathbf{A}}_{:,i}^{(k)}\right]^+\underline{\boldsymbol{\alpha}}_i + \left[\underline{\mathbf{A}}_{:,i}^{(k)}\right]^-\overline{\boldsymbol{\alpha}}_i\right)\mathbf{z}_i + \left[\underline{\mathbf{A}}_{:,i}^{(k)}\right]^-\overline{\boldsymbol{\beta}}_i + \underline{\mathbf{d}}_i^{(k)}$$

$$\overline{\mathbf{A}}_{:,i}^{(k-1)}\mathbf{z}_i + \overline{\mathbf{d}}_i^{(k-1)} = \left(\left[\overline{\mathbf{A}}_{:,i}^{(k)}\right]^+\overline{\boldsymbol{\alpha}}_i + \left[\overline{\mathbf{A}}_{:,i}^{(k)}\right]^-\underline{\boldsymbol{\alpha}}_i\right)\mathbf{z}_i + \left[\overline{\mathbf{A}}_{:,i}^{(k)}\right]^+\overline{\boldsymbol{\beta}}_i + \overline{\mathbf{d}}_i^{(k)},$$

where $\mathbf{A}_{:,i}$ denotes the $i$-th column of matrix $\mathbf{A}$. We now differentiate three cases based on the signs of $\underline{\mathbf{z}}_i^{(k-1)}$ and $\overline{\mathbf{z}}_i^{(k-1)}$.

(i) Consider $\overline{\mathbf{z}}_i^{(k-1)} \le 0$. We have $\underline{\boldsymbol{\alpha}}_i = \overline{\boldsymbol{\alpha}}_i = \overline{\boldsymbol{\beta}}_i = 0$. Therefore,

$$\underline{\mathbf{A}}_{:,i}^{(k-1)}\mathbf{z}_i + \underline{\mathbf{d}}_i^{(k-1)} = \underline{\mathbf{d}}_i^{(k)} \ge \underline{\mathbf{v}}^{(K)}$$

$$\overline{\mathbf{A}}_{:,i}^{(k-1)}\mathbf{z}_i + \overline{\mathbf{d}}_i^{(k-1)} = \overline{\mathbf{d}}_i^{(k)} \le \overline{\mathbf{v}}^{(K)},$$

where both inequalities are due to the induction assumption with $\mathbf{z}_i^{(k)} = 0$ in Equation (13). We can apply the induction assumption since $0 = \left[\overline{\mathbf{z}}_i^{(k-1)}\right]^+ \in [\underline{\mathbf{z}}^{(k)}, \overline{\mathbf{z}}^{(k)}]$ and, therefore, $0 \in [\underline{\mathbf{v}}^{(k)}, \overline{\mathbf{v}}^{(k)}]$ since $[\underline{\mathbf{z}}^{(k)}, \overline{\mathbf{z}}^{(k)}] \subseteq [\underline{\mathbf{v}}^{(k)}, \overline{\mathbf{v}}^{(k)}]$.

(ii) Consider $\overline{\mathbf{z}}_i^{(k-1)} > 0$ and $\underline{\mathbf{z}}_i^{(k-1)} \ge 0$. First, we find that $[\underline{\mathbf{v}}_i^{(k)}, \overline{\mathbf{v}}_i^{(k)}] = [\underline{\mathbf{v}}_i^{(k-1)}, \overline{\mathbf{v}}_i^{(k-1)}]$ according to Equation (12) since $\overline{\mathbf{v}}_i^{(k-1)} \ge \overline{\mathbf{z}}_i^{(k-1)} > 0$. Regarding CROWN, we have $\underline{\boldsymbol{\alpha}}_i = \overline{\boldsymbol{\alpha}}_i = 1$ and $\overline{\boldsymbol{\beta}}_i = 0$. Therefore,

$$\underline{\mathbf{A}}_{:,i}^{(k-1)}\mathbf{z}_i + \underline{\mathbf{d}}_i^{(k-1)} = \left(\left[\underline{\mathbf{A}}_{:,i}^{(k)}\right]^+ + \left[\underline{\mathbf{A}}_{:,i}^{(k)}\right]^-\right)\mathbf{z}_i + \underline{\mathbf{d}}_i^{(k)} = \underline{\mathbf{A}}_{:,i}^{(k)}\mathbf{z}_i + \underline{\mathbf{d}}_i^{(k)} \ge \underline{\mathbf{v}}^{(K)}$$

$$\overline{\mathbf{A}}_{:,i}^{(k-1)}\mathbf{z}_i + \overline{\mathbf{d}}_i^{(k-1)} = \left(\left[\overline{\mathbf{A}}_{:,i}^{(k)}\right]^+ + \left[\overline{\mathbf{A}}_{:,i}^{(k)}\right]^-\right)\mathbf{z}_i + \overline{\mathbf{d}}_i^{(k)} = \overline{\mathbf{A}}_{:,i}^{(k)}\mathbf{z}_i + \overline{\mathbf{d}}_i^{(k)} \le \overline{\mathbf{v}}^{(K)},$$

where the inequalities are due to the induction assumption that we can apply since $\mathbf{z}_i \in [\underline{\mathbf{v}}_i^{(k-1)}, \overline{\mathbf{v}}_i^{(k-1)}] = [\underline{\mathbf{v}}_i^{(k)}, \overline{\mathbf{v}}_i^{(k)}]$.

(iii) Finally, consider $\underline{\mathbf{z}}_i^{(k-1)} < 0 < \overline{\mathbf{z}}_i^{(k-1)}$. With the same argument as in the previous case, we have $[\underline{\mathbf{v}}_i^{(k)}, \overline{\mathbf{v}}_i^{(k)}] = [\underline{\mathbf{v}}_i^{(k-1)}, \overline{\mathbf{v}}_i^{(k-1)}]$. We have

$$\overline{\boldsymbol{\alpha}}_i = \frac{\overline{\mathbf{z}}_i^{(k-1)}}{\overline{\mathbf{z}}_i^{(k-1)} - \underline{\mathbf{z}}_i^{(k-1)}}, \quad \overline{\boldsymbol{\beta}}_i = -\underline{\mathbf{z}}_i^{(k-1)}\overline{\boldsymbol{\alpha}}_i = \frac{-\underline{\mathbf{z}}_i^{(k-1)}\overline{\mathbf{z}}_i^{(k-1)}}{\overline{\mathbf{z}}_i^{(k-1)} - \underline{\mathbf{z}}_i^{(k-1)}}$$

and $\underline{\boldsymbol{\alpha}}_i \in [0, 1]$. Now,

$$\underline{\mathbf{A}}_{:,i}^{(k-1)}\mathbf{z}_i + \underline{\mathbf{d}}_i^{(k-1)}$$

$$= \left(\left[\underline{\mathbf{A}}_{:,i}^{(k)}\right]^+\underline{\boldsymbol{\alpha}}_i + \left[\underline{\mathbf{A}}_{:,i}^{(k)}\right]^-\overline{\boldsymbol{\alpha}}_i\right)\mathbf{z}_i + \left[\underline{\mathbf{A}}_{:,i}^{(k)}\right]^-\overline{\boldsymbol{\beta}}_i + \underline{\mathbf{d}}_i^{(k)} \tag{14a}$$

$$\ge \left[\underline{\mathbf{A}}_{:,i}^{(k)}\right]^+\underline{\boldsymbol{\alpha}}_i\underline{\mathbf{v}}_i^{(k-1)} + \left[\underline{\mathbf{A}}_{:,i}^{(k)}\right]^-\overline{\boldsymbol{\alpha}}_i\overline{\mathbf{v}}_i^{(k-1)} + \left[\underline{\mathbf{A}}_{:,i}^{(k)}\right]^-\overline{\boldsymbol{\beta}}_i + \underline{\mathbf{d}}_i^{(k)} \tag{14b}$$

$$\ge \left[\underline{\mathbf{A}}_{:,i}^{(k)}\right]^+\underline{\mathbf{v}}_i^{(k-1)} + \left[\underline{\mathbf{A}}_{:,i}^{(k)}\right]^-\frac{\overline{\mathbf{z}}_i^{(k-1)}\overline{\mathbf{v}}_i^{(k-1)} - \underline{\mathbf{z}}_i^{(k-1)}\overline{\mathbf{z}}_i^{(k-1)}}{\overline{\mathbf{z}}_i^{(k-1)} - \underline{\mathbf{z}}_i^{(k-1)}} + \underline{\mathbf{d}}_i^{(k)} \tag{14c}$$

$$\ge \left[\underline{\mathbf{A}}_{:,i}^{(k)}\right]^+\underline{\mathbf{v}}_i^{(k-1)} + \left[\underline{\mathbf{A}}_{:,i}^{(k)}\right]^-\frac{\overline{\mathbf{z}}_i^{(k-1)}\overline{\mathbf{v}}_i^{(k-1)} - \underline{\mathbf{z}}_i^{(k-1)}\overline{\mathbf{v}}_i^{(k-1)}}{\overline{\mathbf{z}}_i^{(k-1)} - \underline{\mathbf{z}}_i^{(k-1)}} + \underline{\mathbf{d}}_i^{(k)} \tag{14d}$$

$$\ge \left[\underline{\mathbf{A}}_{:,i}^{(k)}\right]^+\underline{\mathbf{v}}_i^{(k-1)} + \left[\underline{\mathbf{A}}_{:,i}^{(k)}\right]^-\overline{\mathbf{v}}_i^{(k-1)} + \underline{\mathbf{d}}_i^{(k)}$$

$$\ge \underline{\mathbf{A}}_{:,i}^{(k)}\left(\mathbf{1}_{\underline{\mathbf{A}}_{:,i}^{(k)} \ge 0}\underline{\mathbf{v}}_i^{(k-1)} + \mathbf{1}_{\underline{\mathbf{A}}_{:,i}^{(k)} < 0}\overline{\mathbf{v}}_i^{(k-1)}\right) + \underline{\mathbf{d}}_i^{(k)}$$

$$\ge \underline{\mathbf{v}}^{(K)}, \tag{14e}$$

where $\mathbf{1}_{\underline{\mathbf{A}}^{(k)}_{:,i} \geq 0} \in \{0,1\}^{n_k}$ with

$$\left(\mathbf{1}_{\underline{\mathbf{A}}^{(k)}_{:,i} \geq 0}\right)_j = \begin{cases} 1 & \underline{\mathbf{A}}^{(k)}_{j,i} \geq 0 \\ 0 & \text{otherwise} \end{cases}$$

for $j \in [n_k]$ and $\mathbf{1}_{\underline{\mathbf{A}}^{(k)}_{:,i} < 0}$ is defined equivalently. Above, the inequality in Equation (14b) is by choosing $\mathbf{z}_i \in [\underline{\mathbf{v}}^{(k-1)}, \overline{\mathbf{v}}^{(k-1)}]$ as the minimiser of Equation (14a). The inequality in Equation (14c) is by choosing $\underline{\boldsymbol{\alpha}}_i = 1$ which minimises Equation (14b) since $\underline{\mathbf{v}}^{(k-1)}_i \leq \underline{\mathbf{z}}^{(k-1)}_i < 0$. To justify the inequality in Equation (14d) we note that $-\underline{\mathbf{z}}^{(k-1)}_i [\underline{\mathbf{A}}^{(k)}_{:,i}]^- \leq 0$ since $\underline{\mathbf{z}}^{(k-1)}_i < 0$ and, further $\overline{\mathbf{v}}^{(k-1)}_i \geq \overline{\mathbf{z}}^{(k-1)}_i$. Finally, Equation (14e) follows from the induction assumption that we can apply since

$$\left(\mathbf{1}_{\underline{\mathbf{A}}^{(k)}_{:,i} \geq 0} \underline{\mathbf{v}}^{(k-1)}_i + \mathbf{1}_{\underline{\mathbf{A}}^{(k)}_{:,i} < 0} \overline{\mathbf{v}}^{(k-1)}_i\right) \in [\underline{\mathbf{v}}^{(k-1)}, \overline{\mathbf{v}}^{(k-1)}].$$

For the upper bound, we apply similar steps to obtain

$$\overline{\mathbf{A}}^{(k-1)}_{:,i} \mathbf{z}_i + \overline{\mathbf{d}}^{(k-1)}_i = \left(\left[\overline{\mathbf{A}}^{(k)}_{:,i}\right]^+ \overline{\boldsymbol{\alpha}}_i + \left[\overline{\mathbf{A}}^{(k)}_{:,i}\right]^- \underline{\boldsymbol{\alpha}}_i\right) \mathbf{z}_i + \left[\overline{\mathbf{A}}^{(k)}_{:,i}\right]^+ \overline{\boldsymbol{\beta}}_i + \overline{\mathbf{d}}^{(k)}_i$$

$$\leq \left[\overline{\mathbf{A}}^{(k)}_{:,i}\right]^+ \overline{\boldsymbol{\alpha}}_i \overline{\mathbf{v}}^{(k-1)}_i + \left[\overline{\mathbf{A}}^{(k)}_{:,i}\right]^- \underline{\boldsymbol{\alpha}}_i \mathbf{v}^{(k-1)}_i + \left[\overline{\mathbf{A}}^{(k)}_{:,i}\right]^+ \overline{\boldsymbol{\beta}}_i + \overline{\mathbf{d}}^{(k)}_i$$

$$\leq \left[\overline{\mathbf{A}}^{(k)}_{:,i}\right]^+ \frac{\overline{\mathbf{z}}^{(k-1)}_i \overline{\mathbf{v}}^{(k-1)}_i - \underline{\mathbf{z}}^{(k-1)}_i \overline{\mathbf{z}}^{(k-1)}_i}{\overline{\mathbf{z}}^{(k-1)}_i - \underline{\mathbf{z}}^{(k-1)}_i} + \left[\overline{\mathbf{A}}^{(k)}_{:,i}\right]^- \underline{\mathbf{v}}^{(k-1)}_i + \overline{\mathbf{d}}^{(k)}_i$$

$$\leq \left[\overline{\mathbf{A}}^{(k)}_{:,i}\right]^+ \frac{\overline{\mathbf{z}}^{(k-1)}_i \overline{\mathbf{v}}^{(k-1)}_i - \underline{\mathbf{z}}^{(k-1)}_i \overline{\mathbf{v}}^{(k-1)}_i}{\overline{\mathbf{z}}^{(k-1)}_i - \underline{\mathbf{z}}^{(k-1)}_i} + \left[\overline{\mathbf{A}}^{(k)}_{:,i}\right]^- \underline{\mathbf{v}}^{(k-1)}_i + \overline{\mathbf{d}}^{(k)}_i$$

$$\leq \left[\overline{\mathbf{A}}^{(k)}_{:,i}\right]^+ \overline{\mathbf{v}}^{(k-1)}_i + \left[\overline{\mathbf{A}}^{(k)}_{:,i}\right]^- \underline{\mathbf{v}}^{(k-1)}_i + \overline{\mathbf{d}}^{(k)}_i$$

$$\leq \overline{\mathbf{A}}^{(k)}_{:,i} \left(\mathbf{1}_{\overline{\mathbf{A}}^{(k)}_{:,i} \geq 0} \overline{\mathbf{v}}^{(k-1)}_i + \mathbf{1}_{\overline{\mathbf{A}}^{(k)}_{:,i} < 0} \underline{\mathbf{v}}^{(k-1)}_i\right) + \overline{\mathbf{d}}^{(k)}_i$$

$$\leq \overline{\mathbf{v}}^{(K)}.$$

By this induction, we have shown Equation (13). Proposition E.3 now follows for $k = 0$, where $\underline{\mathbf{A}}^{(0)} = \underline{\mathbf{A}}$, $\overline{\mathbf{A}}^{(0)} = \overline{\mathbf{A}}$, $\underline{\mathbf{d}}^{(0)} = \underline{\mathbf{d}}$, $\overline{\mathbf{d}}^{(0)} = \overline{\mathbf{d}}$ and $[\underline{\mathbf{v}}^{(0)}, \overline{\mathbf{v}}^{(0)}] = [\underline{\mathbf{x}}, \overline{\mathbf{x}}]$. $\qquad\square$

Proposition E.3 requires layer bounds $[\underline{\mathbf{z}}^{(1)}, \overline{\mathbf{z}}^{(1)}], \ldots, [\underline{\mathbf{z}}^{(K-1)}, \overline{\mathbf{z}}^{(K-1)}]$ which may not be looser than the CIE bounds. As we have shown in Proposition E.2, this applies for INTERVALARITHMETIC. A consequence of Proposition E.3 is that this also applies to Algorithm 4.

**Corollary E.4.** *Let* net, $[\underline{\mathbf{x}}, \overline{\mathbf{x}}]$ *and* $[\underline{\mathbf{v}}^{(K)}, \overline{\mathbf{v}}^{(K)}]$ *be as in Proposition E.3. It holds that*

$$\underline{\mathbf{v}}^{(K)} \leq \underline{\mathbf{A}}\mathbf{x} + \underline{\mathbf{d}} \quad and \quad \overline{\mathbf{A}}\mathbf{x} + \overline{\mathbf{d}} \leq \overline{\mathbf{v}}^{(K)}, \qquad \forall \mathbf{x} \in [\underline{\mathbf{x}}, \overline{\mathbf{x}}],$$

*where* $\underline{\mathbf{A}}\mathbf{x} + \underline{\mathbf{d}}$ *and* $\overline{\mathbf{A}}\mathbf{x} + \overline{\mathbf{d}}$ *are the bounds on* net$(\mathbf{x})$ *for* $\mathbf{x} \in [\underline{\mathbf{x}}, \overline{\mathbf{x}}]$ *computed by Algorithm 4.*

*Proof.* Corollary E.4 follows from Proposition E.3 by an inductive argument over $k \in [K]$. Let net $= f^{(K)} \circ \cdots \circ f^{(1)}$, $[\underline{\mathbf{x}}, \overline{\mathbf{x}}]$ and $[\underline{\mathbf{v}}^{(1)}, \overline{\mathbf{v}}^{(1)}], \ldots, [\underline{\mathbf{v}}^{(K)}, \overline{\mathbf{v}}^{(K)}]$ be as in Proposition E.3.

*Induction start.* Applying Algorithm 3 to $f^{(:1)} = f^{(1)}$ does not require layer bounds, so that Proposition E.3 holds for the (concretised) CROWN bounds on $f^{(:1)}$.

*Induction step.* Let $k \in \{2, \ldots, K\}$. Assume the concretised CROWN bounds $[\underline{\mathbf{z}}^{(1)}, \overline{\mathbf{z}}^{(1)}], \ldots, [\underline{\mathbf{z}}^{(k-1)}, \overline{\mathbf{z}}^{(k-1)}]$ on $f^{(:1)}, \ldots, f^{(:k-1)}$ satisfy Proposition E.3, that is, $[\underline{\mathbf{z}}^{(k)}, \overline{\mathbf{z}}^{(k)}] \subseteq [\underline{\mathbf{v}}^{(k)}, \overline{\mathbf{v}}^{(k)}]$ for every $k \in [k-1]$. Since we only require layer bounds on $f^{(:1)}, \ldots, f^{(:k-1)}$ to apply CROWN to $f^{(:k)}$, Proposition E.3 applies to the (concretised) CROWN bounds we obtain for $f^{(:k)}$. $\qquad\square$

We are now able to prove that CROWN satisfies Definition C.1 as a corollary of Proposition D.5 and Proposition E.3.

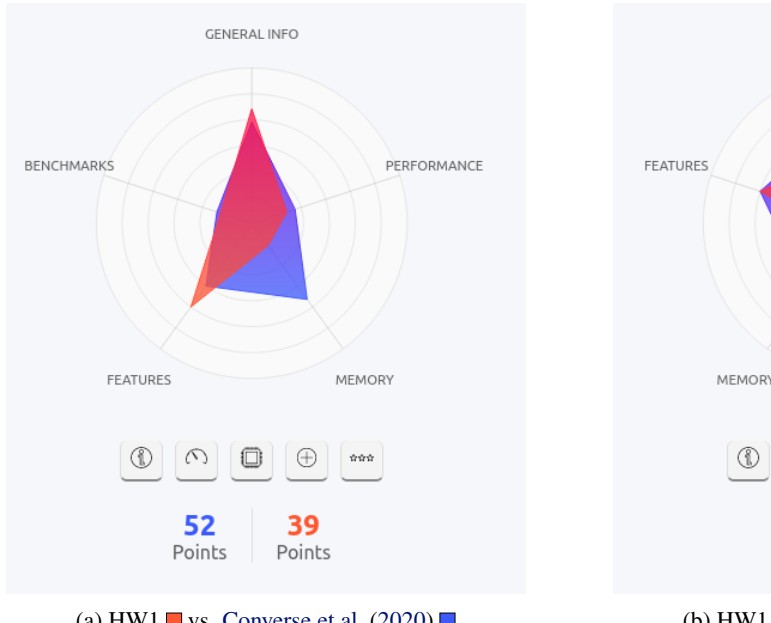
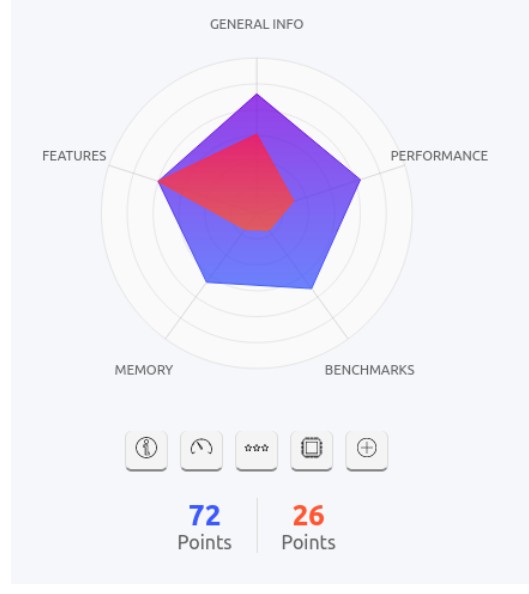

(a) HW1 ■ vs. Converse et al. (2020) ■      (b) HW1 ■ vs. Marzari et al. (2023b) ■

*Figure 4.* Hardware comparison from `versus.com`. The figures are taken from `https://versus.com/en/amd-epyc-7401p-vs-intel-core-i7-6700` and `https://versus.com/en/intel-core-i5-13600kf-vs-intel-core-i7-4820k`, respectively. Accessed on the 17th of May 2024.

**Corollary E.5.** CROWN *satisfies Definition C.1.*

*Proof.* Since CIE is a Lipschitz interval extension according to Proposition E.1, Proposition D.5 applies. Therefore, CIE satisfies Definition C.1. Now, due to Proposition E.3, CROWN also satisfies Definition C.1. □

## F. Experiments

This section contains additional details on the experiments from Section 6 and additional experimental results. A reproducibility package containing all datasets, networks, and raw data from our experiments is available at `https://doi.org/10.5281/zenodo.15521583`. Our code is also available at `https://github.com/sen-uni-kn/probspecs`.

### F.1. Hardware

We run all our experiments on a workstation running Ubuntu 22.04 with an Intel i7–4820K CPU, 32 GB of memory and no GPU (HW1). This CPU model is ten years old (introduced in late 2013) and has four cores and eight threads.

In comparison, Converse et al. (2020) use an AMD EPYC 7401P CPU for their ACAS Xu robustness experiments, limiting their tool to use 46 threads and at most 4GB of memory. AMD EPYC 7401P was introduced in mid-2017, targeting servers. Marzari et al. (2023b) use a Ubuntu 22.04 workstation with an Intel i5–13600KF CPU and an Nvidia GeForce RTX 4070 Ti GPU. This CPU was introduced in end-2022 and has 14 cores with 20 threads.

Figure 4 contains the CPU comparison from `versus.com`. As the figure shows, our HW1 is less performant when considering both the theoretical performance according to the hardware specification and the actual performance on a series of computational benchmarks. Therefore, HW1 is inferior in computation power to the hardware used by Converse et al. (2020) and Marzari et al. (2023b).

### F.2. FairSquare Benchmark

We provide additional details on the FairSquare benchmark and present additional results.

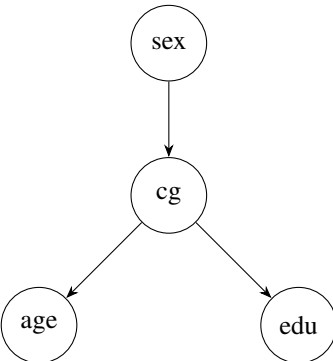

*Figure 5.* FairSquare Bayes Net 1. The network structure of the Bayesian Network from the FairSquare benchmark. In this figure, 'cg' denotes the capital gain variable and 'edu' denotes the number of years of education.

### F.2.1. EXTENDED DESCRIPTION OF THE BENCHMARK

The input space in the FairSquare benchmark consists of three unbounded continuous variables for the age, the years of education ('edu'), and the yearly gain in capital of a person ('capital gain'), respectively, as well as an additional discrete protected variable indicating a person's (assumed binary) sex. The neural networks use 'age' and 'edu' or 'age', 'edu' and 'capital gain', depending on the network architecture, to predict whether a person has a high salary (higher than \$50 000). The three input distributions used by the FairSquare benchmark are

1. the combination of three independent normal distributions for the continuous variables and a Bernoulli distribution for the person's sex,

2. a Bayesian Network with the structure in Figure 5 introducing correlations between the variables that are similarly distributed as for the independent input distribution ('Bayes Net 1'), and

3. the same Bayesian Network augmented with a constraint that the years of education may not exceed a person's age ('Bayes Net 2').

The integrity constraint is implemented by Albarghouthi et al. (2017) as a post-processing of the samples from the Bayesian Network. In particular, a person's years of education are set to the person's age if the sampled years of education are larger than the sampled age. We introduce a pre-processing layer before the neural network we want to verify to implement this integrity constraint in our setting. This pre-processing layer computes $\mathrm{edu}' = \min(\mathrm{edu}, \mathrm{age})$ and leaves the remaining inputs unchanged.

Besides experiments on the demographic parity fairness notion, the extended FairSquare benchmarks also include experiments on the parity of qualified persons fairness notion as defined in Appendix A.1. For both fairness notions, the FairSquare benchmark uses a fairness threshold of $\gamma = 0.85$.

### F.2.2. EXTENDED RESULTS

Table 4 contains the comparison of PV and FAIRSQUARE on the extended set of FairSquare benchmarks from Albarghouthi et al. (2017). The runtime from this table is visualised in Figure 2.

### F.3. ACAS Xu Safety

We provide a comparison of PROBBOUNDS with $\varepsilon$-PROVE (Marzari et al., 2024) on all 36 ACAS Xu networks to which property $\phi_2$ of Katz et al. (2017b) applies. These results are contained in Table 5, revealing that the networks in Table 3 are outliers. However, PROBBOUNDS still computes tighter sound bounds faster than $\varepsilon$-PROVE computes probably sound bounds for 12 of the 36 networks.

Table 6 contains additional results for running PROBBOUNDS with a finer grid of time budgets on the networks from Table 3 and two additional challenging instances from Katz et al. (2017b).

*Table 4.* FairSquare benchmark results. The first two columns indicate the neural network net that is verified and the input probability distribution $\mathbb{P}_{\mathbf{x}}$, respectively. We verify two fairness notions: demographic parity and parity of qualified persons. For each fairness notion, the first two columns contain the runtime of PV and FAIRSQUARE in seconds. Here, 'TO' indices timeout (900s). The last column contains the result of the verification with PV.

| Benchmark Instance | | Demographic Parity | | | Parity of Qualified Persons | | |
| | | Runtime (s) | | | Runtime (s) | | |
| net | $\mathbb{P}_{\mathbf{x}}$ | PV | FairSquare | Fair? | PV | FairSquare | Fair? |
| --- | --- | --- | --- | --- | --- | --- | --- |
| $NN_{2,1}$ | independent | 2 | 2 | ✓ | 2 | 3 | ✓ |
| $NN_{2,1}$ | Bayes Net 1 | 4 | 76 | ✓ | 8 | 38 | ✓ |
| $NN_{2,1}$ | Bayes Net 2 | 5 | 51 | ✓ | 12 | 26 | ✓ |
| $NN_{2,2}$ | independent | 2 | 4 | ✓ | 2 | 7 | ✓ |
| $NN_{2,2}$ | Bayes Net 1 | 4 | 33 | ✓ | 11 | 60 | ✓ |
| $NN_{2,2}$ | Bayes Net 2 | 6 | 59 | ✓ | 17 | 61 | ✓ |
| $NN_{3,2}$ | independent | 2 | 451 | ✓ | 1 | 657 | ✓ |
| $NN_{3,2}$ | Bayes Net 1 | 4 | TO | ✓ | 40 | TO | ✓ |
| $NN_{3,2}$ | Bayes Net 2 | 5 | TO | ✓ | 57 | TO | ✓ |

## F.4. ACAS Xu Robustness

We first provide a more detailed description of the ACAS Xu robustness benchmark. As Converse et al. (2020), we consider 25 reference inputs — five for each class — and allow these reference inputs to be perturbed in the first two dimensions by at most 5% of the diameter of the input space in the respective dimension. To compute bounds on the output distribution, we bound the probability of each of the five ACAS Xu classes for each of the 25 inputs. Since the ACAS Xu training data is not publicly available, we sample the reference inputs randomly.

Table 7 contains the bounds computed by PROBBOUNDS for each reference input and each output class: COC (Clear-of-Conflict), WL (steer Weak Left), WR (steer Weak Right), SL (steer Strong Left), and SR (steer Strong Right). The table reveals that the ACAS Xu network $N_{1,1}$ could tend to classify inputs as Clear-of-Conflict (COC), regardless of the class assigned to the reference input. This insight does not agree with the insight drawn by Converse et al. (2020). However, both results are based on a tiny sample of the input space of only 25 points. Obtaining valid results requires considering a significantly larger sample of the input space or quantifying global robustness (Katz et al., 2017a).

## F.5. MiniACSIncome

We provide additional details on the MiniACSIncome benchmark, including how we construct the dataset and train the networks.

### F.5.1. BENCHMARK

To create probabilistic verification problems of increasing difficulty, we consider an increasing number of input variables from ACSIncome. The smallest instance, MiniACSIncome-1, only contains the binary 'SEX' variable. In contrast, the largest instance, MiniACSIncome-8, contains 'SEX' and seven more variables from ACSIncome, including age, education, and working hours per week. We train a neural network with a single layer of ten neurons for each MiniACSIncome-$i$, $i \in [8]$. For MiniACSIncome-4, we additionally train deeper and wider networks to investigate the scalability of PV with respect to the network size. We fit a Bayesian Network to the MiniACSIncome-8 dataset to obtain an input distribution. The process is described in detail in Appendix F.5.3. We use this distribution for all MiniACSIncome-$i$ instances by taking the variables not contained in MiniACSIncome-$i$ as latent variables. The verification problem is then to verify the demographic parity of a neural network with respect to 'SEX' under this input distribution.

*Table 5.* Comparison of PROBBOUNDS and $\varepsilon$-PROVE for ACAS Xu safety. A PROBBOUNDS column is marked with an arrow ($\leftarrow$) if PROBBOUNDS computes a tighter upper bound than $\varepsilon$-PROVE within a certain time budget. If this is not the case for any time budget, the $\varepsilon$-PROVE entry is marked with an arrow ($\leftarrow$). In total, PROBBOUNDS computes tighter upper bounds within 60s in 16 cases. In 12 cases, it even computes a tighter bound faster than $\varepsilon$-PROVE.

| | | PROBBOUNDS | | | $\varepsilon$-PROVE | |
| | | 10s | 30s | 60s | 99.9% confid. | |
| $\phi$ | net | $\ell, u$ | $\ell, u$ | $\ell, u$ | $u$ | Rt |
|---|---|---|---|---|---|---|
| $\varphi_2$ | $N_{2,1}$ | $0.03\%, 3.19\%$ | $0.09\%, 2.50\%\leftarrow$ | $0.16\%, 2.09\%$ | $2.58\%$ | 64s |
| | $N_{2,2}$ | $0.00\%, 3.86\%$ | $0.08\%, 2.93\%$ | $0.24\%, 2.64\%$ | $2.49\%\leftarrow$ | 44s |
| | $N_{2,3}$ | $0.51\%, 3.46\%\leftarrow$ | $0.85\%, 2.98\%$ | $1.00\%, 2.68\%$ | $3.58\%$ | 55s |
| | $N_{2,4}$ | $0.00\%, 2.72\%\leftarrow$ | $0.03\%, 2.13\%$ | $0.04\%, 2.06\%$ | $2.78\%$ | 66s |
| | $N_{2,5}$ | $0.03\%, 4.23\%$ | $0.12\%, 3.69\%$ | $0.34\%, 3.14\%$ | $3.06\%\leftarrow$ | 47s |
| | $N_{2,6}$ | $0.01\%, 3.67\%$ | $0.16\%, 2.89\%$ | $0.30\%, 2.50\%$ | $2.32\%\leftarrow$ | 45s |
| | $N_{2,7}$ | $0.28\%, 5.56\%$ | $0.84\%, 4.87\%$ | $1.26\%, 4.32\%$ | $4.04\%\leftarrow$ | 47s |
| | $N_{2,8}$ | $0.96\%, 3.75\%$ | $1.23\%, 3.00\%\leftarrow$ | $1.37\%, 2.79\%$ | $3.28\%$ | 53s |
| | $N_{2,9}$ | $0.01\%, 2.85\%$ | $0.08\%, 1.65\%$ | $0.10\%, 1.39\%$ | $0.58\%\leftarrow$ | 11s |
| | $N_{3,1}$ | $0.23\%, 4.25\%$ | $0.46\%, 3.41\%$ | $0.88\%, 2.74\%\leftarrow$ | $2.84\%$ | 40s |
| | $N_{3,2}$ | $0.00\%, 3.57\%$ | $0.00\%, 1.49\%$ | $0.00\%, 1.15\%$ | $0.00\%\leftarrow$ | 1s |
| | $N_{3,3}$ | $0.00\%, 2.94\%$ | $0.00\%, 1.91\%$ | $0.00\%, 0.87\%$ | $0.00\%\leftarrow$ | 1s |
| | $N_{3,4}$ | $0.00\%, 3.27\%$ | $0.09\%, 2.07\%$ | $0.13\%, 1.53\%$ | $1.36\%\leftarrow$ | 31s |
| | $N_{3,5}$ | $0.31\%, 2.90\%$ | $0.53\%, 2.29\%\leftarrow$ | $0.63\%, 2.02\%$ | $2.67\%$ | 42s |
| | $N_{3,6}$ | $0.00\%, 6.23\%$ | $0.00\%, 5.42\%$ | $0.01\%, 5.19\%$ | $2.69\%\leftarrow$ | 32s |
| | $N_{3,7}$ | $0.00\%, 5.76\%$ | $0.00\%, 4.59\%$ | $0.00\%, 3.36\%$ | $0.91\%\leftarrow$ | 30s |
| | $N_{3,8}$ | $0.15\%, 5.45\%$ | $0.21\%, 4.43\%$ | $0.32\%, 3.48\%$ | $2.79\%\leftarrow$ | 61s |
| | $N_{3,9}$ | $0.28\%, 4.83\%$ | $1.06\%, 4.00\%$ | $1.39\%, 3.72\%$ | $3.52\%\leftarrow$ | 32s |
| | $N_{4,1}$ | $0.00\%, 3.02\%$ | $0.01\%, 1.96\%$ | $0.04\%, 1.57\%$ | $1.16\%\leftarrow$ | 26s |
| | $N_{4,2}$ | $0.00\%, 2.62\%$ | $0.00\%, 1.69\%$ | $0.00\%, 1.36\%$ | $0.00\%\leftarrow$ | 1s |
| | $N_{4,3}$ | $0.17\%, 2.92\%$ | $0.34\%, 2.65\%$ | $0.61\%, 2.27\%\leftarrow$ | $2.43\%$ | 41s |
| | $N_{4,4}$ | $0.05\%, 2.47\%$ | $0.17\%, 2.22\%$ | $0.27\%, 1.98\%$ | $1.94\%\leftarrow$ | 40s |
| | $N_{4,5}$ | $0.00\%, 4.52\%$ | $0.14\%, 3.50\%$ | $0.46\%, 3.04\%$ | $2.85\%\leftarrow$ | 41s |
| | $N_{4,6}$ | $0.32\%, 4.54\%$ | $0.89\%, 3.54\%$ | $1.16\%, 3.27\%$ | $3.22\%\leftarrow$ | 39s |
| | $N_{4,7}$ | $0.09\%, 4.17\%$ | $0.31\%, 3.32\%$ | $0.55\%, 2.92\%$ | $2.58\%\leftarrow$ | 30s |
| | $N_{4,8}$ | $0.15\%, 3.91\%$ | $0.56\%, 3.20\%\leftarrow$ | $0.85\%, 2.84\%$ | $3.51\%$ | 52s |
| | $N_{4,9}$ | $0.00\%, 3.36\%$ | $0.00\%, 1.96\%$ | $0.00\%, 1.55\%$ | $0.39\%\leftarrow$ | 11s |
| | $N_{5,1}$ | $0.02\%, 2.60\%$ | $0.30\%, 2.13\%\leftarrow$ | $0.42\%, 1.99\%$ | $2.14\%$ | 35s |
| | $N_{5,2}$ | $0.17\%, 2.66\%\leftarrow$ | $0.47\%, 2.03\%$ | $0.52\%, 1.92\%$ | $3.22\%$ | 67s |
| | $N_{5,3}$ | $0.00\%, 1.72\%$ | $0.00\%, 0.73\%$ | $0.00\%, 0.56\%$ | $0.00\%\leftarrow$ | 1s |
| | $N_{5,4}$ | $0.01\%, 2.94\%$ | $0.14\%, 2.34\%$ | $0.22\%, 2.08\%\leftarrow$ | $2.31\%$ | 54s |
| | $N_{5,5}$ | $0.74\%, 3.29\%$ | $1.32\%, 2.68\%\leftarrow$ | $1.37\%, 2.62\%$ | $2.77\%$ | 33s |
| | $N_{5,6}$ | $0.18\%, 4.46\%\leftarrow$ | $0.78\%, 3.19\%$ | $0.98\%, 3.02\%$ | $4.82\%$ | 74s |
| | $N_{5,7}$ | $1.19\%, 4.63\%$ | $1.74\%, 4.02\%\leftarrow$ | $1.99\%, 3.75\%$ | $4.27\%$ | 45s |
| | $N_{5,8}$ | $0.89\%, 4.16\%$ | $1.31\%, 3.39\%$ | $1.55\%, 3.10\%\leftarrow$ | $3.38\%$ | 42s |
| | $N_{5,9}$ | $0.62\%, 3.75\%$ | $1.24\%, 3.06\%\leftarrow$ | $1.41\%, 2.89\%$ | $3.06\%$ | 36s |
| | # $\leftarrow$ | 4 | 8 | 5 | 20 | |

*Table 6.* Extended PROBBOUNDS results for ACAS Xu safety.

| | | Timeout | | | | | | | |
|---|---|---|---|---|---|---|---|---|---|
| | | **10s** | | **30s** | | **1m** | | | |
| $\phi$ | net | $\ell, u$ | $u - \ell$ | $\ell, u$ | $u - \ell$ | $\ell, u$ | $u - \ell$ | | |
| $\phi_2$ | $N_{4,3}$ | $0.17\%, 2.92\%$ | $2.74\%$ | $0.34\%, 2.62\%$ | $2.30\%$ | $0.61\%, 2.27\%$ | $1.66\%$ | | |
| | $N_{4,9}$ | $0.00\%, 3.36\%$ | $3.36\%$ | $0.00\%, 1.96\%$ | $1.96\%$ | $0.00\%, 1.55\%$ | $1.55\%$ | | |
| | $N_{5,8}$ | $0.89\%, 4.16\%$ | $3.26\%$ | $1.31\%, 3.39\%$ | $2.08\%$ | $1.55\%, 3.10\%$ | $1.55\%$ | | |
| $\phi_7$ | $N_{1,9}$ | $0.00\%, 98.71\%$ | $98.71\%$ | $0.00\%, 94.18\%$ | $94.18\%$ | $0.00\%, 87.62\%$ | $87.62\%$ | | |
| $\phi_8$ | $N_{2,9}$ | $0.00\%, 76.39\%$ | $76.39\%$ | $0.00\%, 65.83\%$ | $65.83\%$ | $0.00\%, 58.11\%$ | $58.11\%$ | | |

| | | **10m** | | **1h** | | |
|---|---|---|---|---|---|---|
| $\phi$ | net | $\ell, u$ | $u - \ell$ | $\ell, u$ | $u - \ell$ | |
| $\phi_2$ | $N_{4,3}$ | $0.95\%, 1.93\%$ | $0.98\%$ | $1.12\%, 1.75\%$ | $0.63\%$ | |
| | $N_{4,9}$ | $0.03\%, 0.52\%$ | $0.48\%$ | $0.08\%, 0.29\%$ | $0.21\%$ | |
| | $N_{5,8}$ | $1.90\%, 2.66\%$ | $0.76\%$ | $1.97\%, 2.57\%$ | $0.60\%$ | |
| $\phi_7$ | $N_{1,9}$ | $0.00\%, 51.93\%$ | $51.93\%$ | $0.00\%, 30.71\%$ | $30.71\%$ | |
| $\phi_8$ | $N_{2,9}$ | $0.00\%, 34.60\%$ | $34.60\%$ | $0.00\%, 15.33\%$ | $15.33\%$ | |

### F.5.2. DATASET

The MiniACSIncome benchmarks are built by sampling about $100\,000$ entries from the ACS PUMPS 1-Year horizon data for all states of the USA for the year 2018 using the `folktables` Python package (Ding et al., 2021). In line with ACSIncome (Ding et al., 2021), we only sample individuals older than 16 years, with a yearly income of at least \$100, reported working hours per week of at least 1, and a 'PWGTP' (more details in Ding et al. (2021)) of at least 1. In total, our dataset contains $102\,621$ samples.

**Variable Order.** For obtaining the benchmark MiniACSIncome-$i$, $i \in [8]$, we select $i$ input variables in the following order: 'SEX', 'COW', 'SCHL', 'WKHP', 'MAR', 'RAC1P', 'RELP', 'AGEP'. We choose 'SEX' as the first variable so that we can verify the fairness with respect to 'SEX' on every benchmark instance. The order of the remaining variables is chosen based on each variable's expected predictive value and the number of discrete values. In particular, we select 'COW' (class of work), 'SCHL' (level of education), and 'WKHP' (work hours per week) first, as we consider these variables to be more predictive than 'MAR' (marital status), 'RAC1P' (races of a person), 'RELP' (relationship), and 'AGEP' (age of a person). The variables are ordered by their number of discrete values within these groups of expected predictive value. For example, 'COW' has nine categories, while 'WKHP' has 99 possible integer values.

Table 8 contains the number of input dimensions and the total number of discrete values in each MiniACSIncome-$i$ input space. The input space contains more dimensions than input variables due to the one-hot encoding of all categorical variables.

### F.5.3. INPUT DISTRIBUTION

The Bayesian Network input distribution of MiniACSIncome has the network structure depicted in Figure 6. For using 'AGEP' as a parent node, we summarised the age groups 17–34, 35–59, and 60–95. This means that the conditional probability table of 'MAR' does not have 78 entries for 'AGEP', but three, corresponding to the ranges 17–34, 35–59, and 60–95.

To fit the Bayesian Network, we walk the network from sources to sinks and fit each conditional distribution to match the empirical distribution of the data subset that matches the current condition. For example, for fitting the conditional distribution of 'SCHL' given 'SEX=1', 'RAC1P=1', we select the samples in the dataset having 'SEX=1' and 'RAC1P=1' and fit the conditional distribution of 'SCHL' to match the empirical distribution of these samples. We use categorical distributions for all categorical variables and 'WKHP'. For 'AGEP', we fit a mixture model of four truncated normal

*Table 7.* ACAS Xu robustness results. We report the lower and upper bounds ($\ell$, $u$) PROBBOUNDS computes for the probability that a perturbed input is classified as the target label and the runtime in seconds (Rt) of PROBBOUNDS for computing these bounds. For each combination of unperturbed label, input, and target label, we run PROBBOUNDS until the difference between the lower and the upper bound is at most 0.1%.

| Reference Input | | COC | | WL | | WR | | SL | | SR | |
| | | \multicolumn | | | | Target Label | | | | | |
| Label | Input # | $\ell$, $u$ | Rt | $\ell$, $u$ | Rt | $\ell$, $u$ | Rt | $\ell$, $u$ | Rt | $\ell$, $u$ | Rt |
|---|---|---|---|---|---|---|---|---|---|---|---|
| COC | 1 | 100%, 100% | 0s | 0.0%, 0.0% | 0s | 0.0%, 0.0% | 0s | 0.0%, 0.0% | 0s | 0.0%, 0.0% | 0s |
| COC | 2 | 99.9%, 100% | 1s | 0.0%, 0.0% | 1s | 0.0%, 0.1% | 1s | 0.0%, 0.0% | 1s | 0.0%, 0.0% | 1s |
| COC | 3 | 91.7%, 91.8% | 3s | 1.3%, 1.4% | 6s | 1.9%, 2.0% | 4s | 2.1%, 2.2% | 12s | 2.8%, 2.8% | 4s |
| COC | 4 | 100%, 100% | 0s | 0.0%, 0.0% | 0s | 0.0%, 0.0% | 0s | 0.0%, 0.0% | 0s | 0.0%, 0.0% | 0s |
| COC | 5 | 100%, 100% | 0s | 0.0%, 0.0% | 0s | 0.0%, 0.0% | 0s | 0.0%, 0.0% | 0s | 0.0%, 0.0% | 0s |
| WL | 1 | 89.9%, 90.0% | 2s | 1.6%, 1.7% | 38s | 4.7%, 4.8% | 2s | 3.6%, 3.7% | 39s | 0.0%, 0.0% | 1s |
| WL | 2 | 92.9%, 93.0% | 3s | 2.3%, 2.4% | 2s | 3.0%, 3.1% | 3s | 1.4%, 1.5% | 3s | 0.2%, 0.3% | 1s |
| WL | 3 | 90.4%, 90.5% | 3s | 1.1%, 1.2% | 8s | 3.6%, 3.7% | 4s | 3.8%, 3.9% | 11s | 0.8%, 0.9% | 2s |
| WL | 4 | 83.6%, 83.7% | 18s | 5.4%, 5.5% | 52s | 2.6%, 2.7% | 52s | 4.6%, 4.7% | 60s | 3.5%, 3.6% | 65s |
| WL | 5 | 96.8%, 96.9% | 2s | 2.9%, 3.0% | 1s | 0.0%, 0.1% | 1s | 0.0%, 0.0% | 1s | 0.1%, 0.2% | 1s |
| WR | 1 | 62.2%, 62.3% | 18s | 0.0%, 0.0% | 1s | 3.4%, 3.5% | 19s | 15.8%, 15.9% | 9s | 18.5%, 18.6% | 12s |
| WR | 2 | 96.6%, 96.7% | 1s | 1.5%, 1.6% | 1s | 1.7%, 1.8% | 2s | 0.0%, 0.1% | 1s | 0.0%, 0.1% | 1s |
| WR | 3 | 94.8%, 94.9% | 3s | 0.0%, 0.1% | 28s | 1.9%, 2.0% | 213s | 0.0%, 0.1% | 5s | 3.2%, 3.3% | 123s |
| WR | 4 | 94.3%, 94.4% | 2s | 1.8%, 1.9% | 2s | 3.4%, 3.5% | 2s | 0.3%, 0.4% | 2s | 0.0%, 0.1% | 1s |
| WR | 5 | 91.1%, 91.1% | 4s | 0.8%, 0.9% | 7s | 4.2%, 4.3% | 9s | 3.0%, 3.1% | 12s | 0.7%, 0.8% | 4s |
| SL | 1 | 81.1%, 81.2% | 22s | 2.3%, 2.4% | 77s | 4.8%, 4.9% | 155s | 4.8%, 4.9% | 64s | 6.7%, 6.8% | 138s |
| SL | 2 | 93.8%, 93.9% | 3s | 1.1%, 1.2% | 26s | 3.7%, 3.8% | 44s | 0.7%, 0.8% | 11s | 0.5%, 0.6% | 16s |
| SL | 3 | 83.3%, 83.4% | 10s | 2.8%, 2.9% | 15s | 6.5%, 6.6% | 41s | 2.1%, 2.2% | 12s | 5.0%, 5.1% | 28s |
| SL | 4 | 81.8%, 81.9% | 18s | 3.3%, 3.4% | 94s | 5.2%, 5.3% | 135s | 5.6%, 5.7% | 85s | 4.0%, 4.1% | 58s |
| SL | 5 | 84.9%, 85.0% | 13s | 1.7%, 1.8% | 25s | 5.4%, 5.5% | 40s | 3.0%, 3.1% | 26s | 4.8%, 4.9% | 39s |
| SR | 1 | 91.0%, 91.0% | 5s | 0.0%, 0.1% | 3s | 5.2%, 5.3% | 65s | 0.0%, 0.1% | 2s | 3.6%, 3.7% | 34s |
| SR | 2 | 87.0%, 87.1% | 7s | 0.9%, 1.0% | 5s | 5.7%, 5.8% | 71s | 1.2%, 1.3% | 6s | 5.0%, 5.1% | 88s |
| SR | 3 | 88.4%, 88.5% | 3s | 2.5%, 2.6% | 9s | 4.7%, 4.8% | 9s | 2.6%, 2.7% | 15s | 1.5%, 1.6% | 9s |
| SR | 4 | 93.0%, 93.1% | 6s | 0.7%, 0.8% | 62s | 3.7%, 3.8% | 119s | 1.0%, 1.1% | 30s | 1.4%, 1.5% | 41s |
| SR | 5 | 79.3%, 79.4% | 6s | 0.2%, 0.3% | 2s | 9.1%, 9.2% | 10s | 3.7%, 3.8% | 5s | 7.4%, 7.5% | 6s |

Table 8. MiniACSIncome details and PV verification results.

| | #Input Variables | | | | | | | |
| --- | --- | --- | --- | --- | --- | --- | --- | --- |
| | 1 | 2 | 3 | 4 | 5 | 6 | 7 | 8 |
| #Input Dimensions | 2 | 10 | 34 | 35 | 40 | 49 | 67 | 68 |
| #Discrete Values | 2 | 16 | 382 | 38K | 190K | 1.7M | 31M | 2B |
| PV Runtime | 16s | 44s | 127s | 231s | 374s | 699s | 1383s | TO |
| 10-Neuron Network Fair? | × | × | × | × | × | × | × | ? |

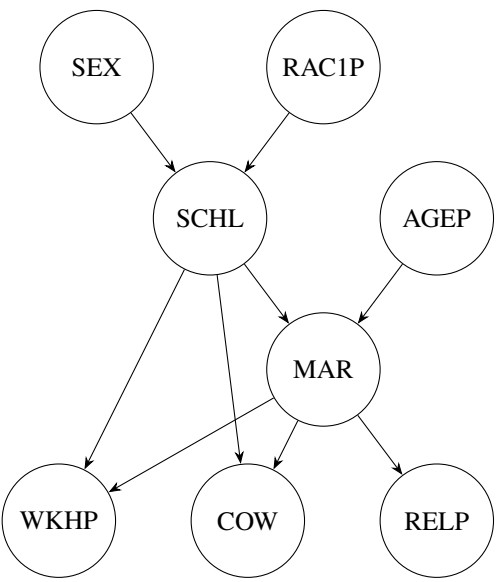

Figure 6. MiniACSIncome Bayesian network structure.

distributions that we discretise to integer values.

Figure 7 depict the marginal distributions of 10 000 samples from the fitted Bayesian Network and the empirical marginal distributions of the MiniACSIncome-8 dataset. Figure 8 contains the correlation matrix of the same sample compared to the correlation matrix of MiniACSIncome-8. The Bayesian Network approximates the empirical marginal distribution and the correlation structure of MiniACSIncome-8 reasonably well.

Note that for real-world fairness verification, the input distribution should not be fitted to the same dataset on which the neural network to verify is trained on. Instead, the input distribution needs to be carefully constructed by domain experts. For fairness audits, the input distribution could also be designed adversarially by a fairness auditing entity.

### F.5.4. TRAINING

All MiniACSIncome neural networks are trained on a 56%/14%/30% split of the MiniACSIncome-$i$ dataset into training, validation, and testing data. This split is identical for all MiniACSIncome-$i$ datasets. All networks are trained using: the Adam optimiser (Kingma & Ba, 2015), cross entropy as loss function, no $L_2$ regularisation (weight decay), $\beta_1 = 0.9$, $\beta_2 = 0.999$, $\varepsilon = 10^{-8}$ as suggested by Kingma & Ba (2015), and a learning rate decay by 0.1 after 2000 and 4000 iterations. The learning rate and the number of training epochs are contained in Table 9. We perform five random restarts for each network and select the network with the lowest cross-entropy on the validation data.

Table 10 contains the accuracy, precision, and recall for the overall test set, the persons with female sex in the test set, and the persons with male sex in the test set. Additionally, the table contains whether a network satisfies the demographic parity fairness notion according to PV.

We used OPTUNA (Akiba et al., 2019) for an initial exploration of the hyperparameter space but did not apply automatic hyperparameter optimisation to obtain the final training hyperparameters.

### F.5.5. EXTENDED RESULTS

We provide concrete results on the effect of the network size on the runtime of PV on the MiniACSIncome benchmark. Figure 9 displays the runtime of PV for MiniACSIncome-4 networks of various sizes. The studied networks include wide single-layer networks of up to 10 000 neurons and deep networks of up to 10 layers of 10 neurons. As the figure shows, PV is largely unaffected by the size of the MiniACSIncome-4 networks. This is unexpected since the network size indirectly determines the performance of PV through the complexity of the decision boundary. However, larger networks need not necessarily have a more complex decision boundary, and large networks do not provide a performance benefit for MiniACSIncome-4, as apparent from Table 10. Thoroughly exploring the impacts of network size requires more intricate datasets for which larger networks actually provide a benefit.

## G. Heuristics for PROBBOUNDS

In this section, we experimentally compare the LONGESTEDGE, BABSB, and BABSB-LONGESTEDGE-k heuristics, as well as INTERVALARITHMETIC and CROWN to justify our decision to use BABSB and CROWN in Section 6.

### G.1. Experiments

We experimentally compare the LONGESTEDGE, BABSB and BABSB-LONGESTEDGE-k heuristics to justify our selection in Section 6. Concretely, we study BABSB-LONGESTEDGE-10. Additionally, we compare INTERVALARITHMETIC to CROWN when used as COMPUTEBOUNDS procedure of PROBBOUNDS. To perform the comparison, we run the different variants of PV on the FairSquare benchmark, as described in Section 6.1.

Figure 10 compares the LONGESTEDGE, BABSB and BABSB-LONGESTEDGE-10 heuristics. It contains the runtime of PV when using SELECTPROB, CROWN and either LONGESTEDGE, BABSB, or BABSB-LONGESTEDGE-10 as SPLIT heuristic. While using LONGESTEDGE is faster for four easy-to-solve benchmark instances, using it only allows solving eight instances from the FairSquare benchmark, while using BABSB allows solving all 18 instances. Using BABSB-LONGESTEDGE-10 also allows us to solve all benchmark instances while requiring slightly more time per benchmark instance than using BABSB.

Figure 11 contains the runtime of PV when using SELECTPROB, BABSB and either INTERVALARITHMETIC or CROWN

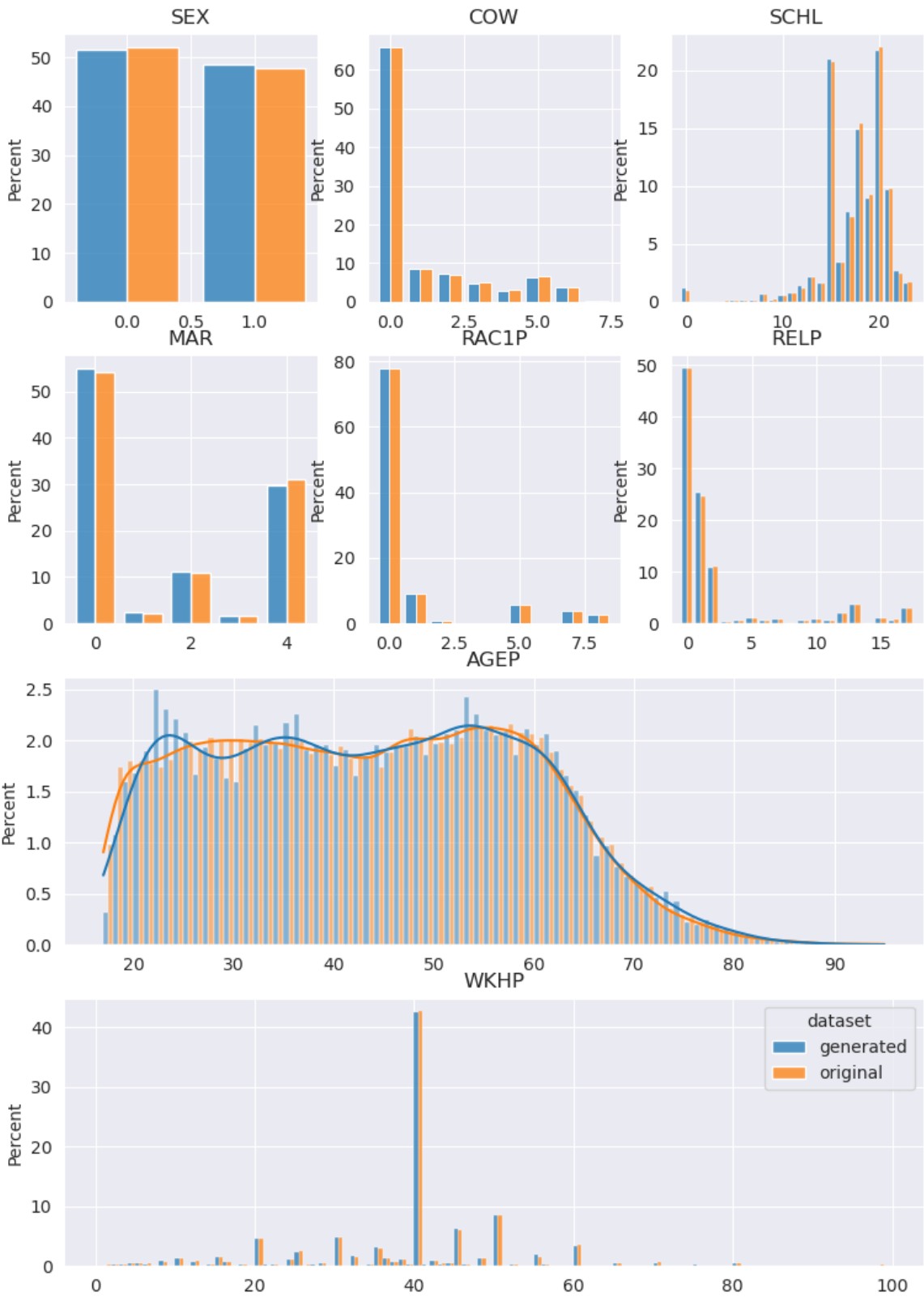

*Figure 7.* MiniACSIncome Bayesian network — marginal distributions. The 'generated' data is sampled from the Bayesian Network, while the 'original' data is the MiniACSIncome-8 dataset.

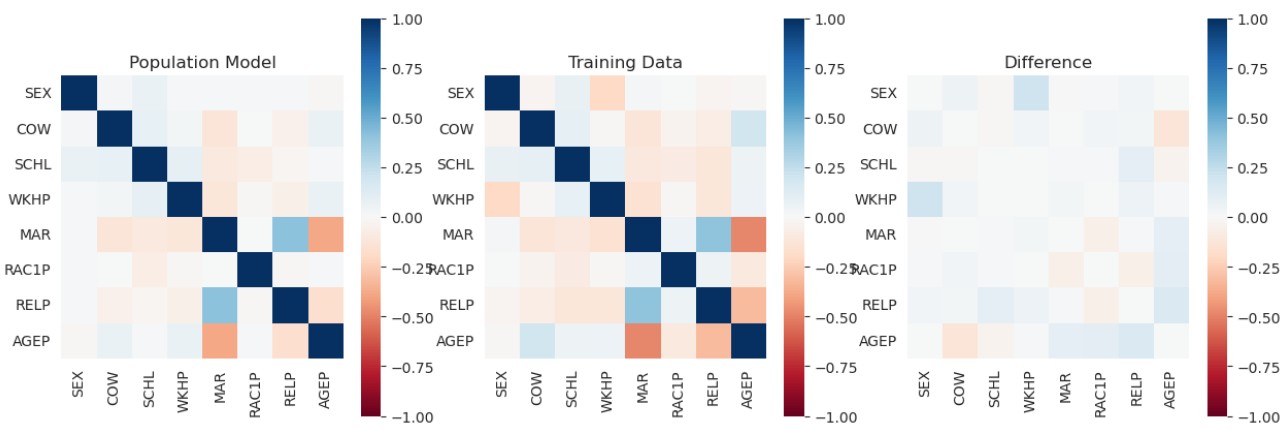

*Figure 8.* MiniACSIncome Bayesian network — correlation matrix. 'Population Model' denotes the fitted Bayesian Network, while 'Training Data' stands for the full MiniACSIncome-8 dataset.

*Table 9.* MiniACSIncome training hyperparameters.

| Dataset | Architecture | Learning Rate | # Epochs |
|---|---|---|---|
| MiniACSIncome-1 | $1\times10$ | 0.0001 | 1 |
| MiniACSIncome-2 | $1\times10$ | 0.001 | 2 |
| MiniACSIncome-3 | $1\times10$ | 0.001 | 3 |
| MiniACSIncome-4 | $1\times10$ | 0.001 | 4 |
| MiniACSIncome-5 | $1\times10$ | 0.001 | 5 |
| MiniACSIncome-6 | $1\times10$ | 0.001 | 3 |
| MiniACSIncome-7 | $1\times10$ | 0.001 | 3 |
| MiniACSIncome-8 | $1\times10$ | 0.001 | 3 |
| MiniACSIncome-4 | $1\times1000$ | 0.001 | 2 |
| MiniACSIncome-4 | $1\times2000$ | 0.001 | 2 |
| MiniACSIncome-4 | $1\times3000$ | 0.001 | 2 |
| MiniACSIncome-4 | $1\times4000$ | 0.001 | 2 |
| MiniACSIncome-4 | $1\times5000$ | 0.001 | 2 |
| MiniACSIncome-4 | $1\times6000$ | 0.001 | 2 |
| MiniACSIncome-4 | $1\times7000$ | 0.001 | 2 |
| MiniACSIncome-4 | $1\times8000$ | 0.001 | 2 |
| MiniACSIncome-4 | $1\times9000$ | 0.001 | 2 |
| MiniACSIncome-4 | $1\times10\,000$ | 0.001 | 2 |
| MiniACSIncome-4 | $2\times10$ | 0.001 | 2 |
| MiniACSIncome-4 | $3\times10$ | 0.001 | 2 |
| MiniACSIncome-4 | $4\times10$ | 0.001 | 2 |
| MiniACSIncome-4 | $5\times10$ | 0.001 | 2 |
| MiniACSIncome-4 | $6\times10$ | 0.001 | 2 |
| MiniACSIncome-4 | $7\times10$ | 0.001 | 2 |
| MiniACSIncome-4 | $8\times10$ | 0.001 | 2 |
| MiniACSIncome-4 | $9\times10$ | 0.001 | 2 |
| MiniACSIncome-4 | $10\times10$ | 0.001 | 2 |

*Table 10.* MiniACSIncome neural networks. The abbreviation 'mACSI-$i$' stands for MiniACSIncome-$i$. We report the accuracy (A), precision (P), and recall (R) of each trained network (Net) for the whole test dataset (Overall), the persons with 'SEX=2' in the test set (Female), and the persons with 'SEX=1' in the test set (Male). Additionally, we report whether the network satisfies the demographic parity fairness notion according to PV (Fair?).

| Dataset | Net | Overall | | | Female | | | Male | | | Fair? |
|---------|-----|---|---|---|---|---|---|---|---|---|-------|
| | | **A** | **P** | **R** | **A** | **P** | **R** | **A** | **P** | **R** | |
| mACSI-1 | 1×10 | 57% | 44% | 63% | 72% | – | 0% | 44% | 44% | 100% | ✗ |
| mACSI-2 | 1×10 | 65% | 55% | 14% | 71% | – | 0% | 58% | 55% | 23% | ✗ |
| mACSI-3 | 1×10 | 73% | 67% | 48% | 76% | 65% | 35% | 69% | 68% | 55% | ✗ |
| mACSI-4 | 1×10 | 75% | 68% | 60% | 78% | 66% | 48% | 72% | 69% | 66% | ✗ |
| mACSI-5 | 1×10 | 76% | 69% | 63% | 79% | 65% | 53% | 74% | 71% | 69% | ✗ |
| mACSI-6 | 1×10 | 77% | 69% | 63% | 79% | 65% | 55% | 74% | 72% | 68% | ✗ |
| mACSI-7 | 1×10 | 77% | 71% | 64% | 79% | 67% | 50% | 76% | 72% | 72% | ✗ |
| mACSI-8 | 1×10 | 78% | 70% | 68% | 80% | 67% | 56% | 76% | 71% | 75% | ? |
| mACSI-4 | 1×1000 | 75% | 71% | 54% | 79% | 70% | 44% | 72% | 71% | 61% | ✗ |
| mACSI-4 | 1×2000 | 75% | 65% | 67% | 78% | 61% | 60% | 72% | 67% | 71% | ✓ |
| mACSI-4 | 1×3000 | 74% | 63% | 71% | 77% | 58% | 67% | 72% | 66% | 73% | ✓ |
| mACSI-4 | 1×4000 | 75% | 71% | 55% | 79% | 69% | 44% | 72% | 71% | 61% | ✗ |
| mACSI-4 | 1×5000 | 75% | 69% | 58% | 78% | 71% | 39% | 73% | 68% | 69% | ✗ |
| mACSI-4 | 1×6000 | 75% | 69% | 59% | 79% | 69% | 44% | 72% | 68% | 67% | ✗ |
| mACSI-4 | 1×7000 | 75% | 66% | 64% | 78% | 62% | 55% | 72% | 68% | 68% | ✗ |
| mACSI-4 | 1×8000 | 75% | 66% | 64% | 78% | 63% | 55% | 72% | 68% | 69% | ✗ |
| mACSI-4 | 1×9000 | 75% | 70% | 53% | 78% | 66% | 47% | 71% | 72% | 56% | ✗ |
| mACSI-4 | 1×10 000 | 75% | 67% | 63% | 79% | 69% | 45% | 72% | 66% | 73% | ✗ |
| mACSI-4 | 2×10 | 75% | 67% | 59% | 78% | 65% | 48% | 72% | 69% | 65% | ✗ |
| mACSI-4 | 3×10 | 75% | 68% | 58% | 79% | 67% | 47% | 72% | 69% | 65% | ✗ |
| mACSI-4 | 4×10 | 75% | 68% | 58% | 78% | 66% | 46% | 72% | 69% | 66% | ✗ |
| mACSI-4 | 5×10 | 75% | 68% | 59% | 78% | 67% | 47% | 72% | 69% | 66% | ✗ |
| mACSI-4 | 6×10 | 75% | 67% | 61% | 78% | 67% | 46% | 72% | 67% | 70% | ✗ |
| mACSI-4 | 7×10 | 75% | 65% | 66% | 78% | 63% | 52% | 72% | 65% | 74% | ✗ |
| mACSI-4 | 7×10 | 75% | 67% | 59% | 78% | 65% | 46% | 72% | 68% | 67% | ✗ |
| mACSI-4 | 9×10 | 75% | 67% | 59% | 78% | 65% | 47% | 72% | 68% | 67% | ✗ |
| mACSI-4 | 19×10 | 75% | 67% | 61% | 78% | 64% | 49% | 72% | 68% | 68% | ✗ |

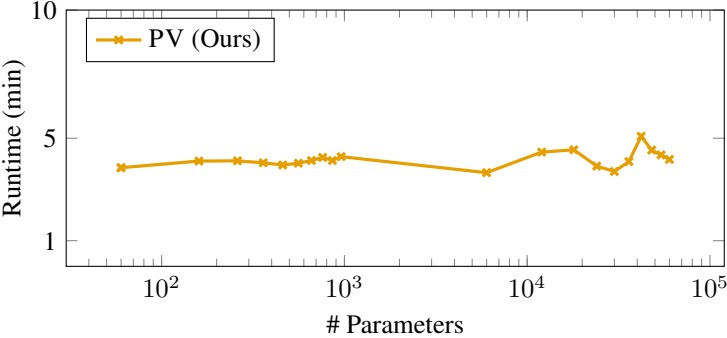

*Figure 9.* MiniACSIncome network size results. The plot depicts the runtime of PV for MiniACSIncome-4 networks of varying sizes.

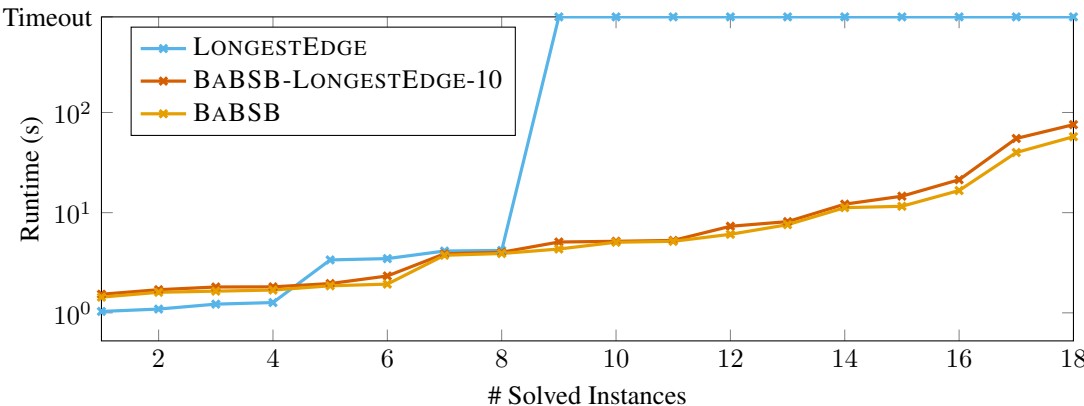

*Figure 10.* SPLIT heuristic comparison on the FairSquare benchmark. The timeout for the FairSquare benchmark is 15min.

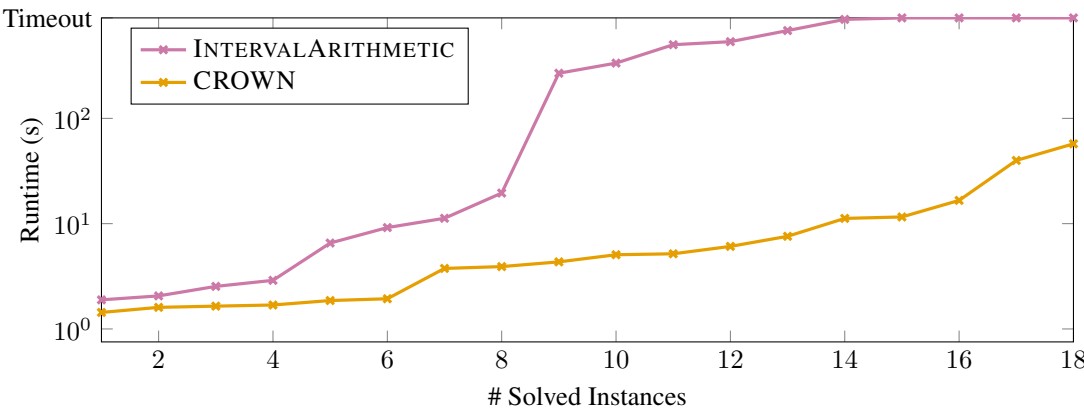

*Figure 11.* COMPUTEBOUNDS procedure comparison on the FairSquare benchmark. The timeout for the FairSquare benchmark is 15min.

as COMPUTEBOUNDS procedure in PROBBOUNDS. As the figure reveals, using CROWN allows PV to terminate faster on all benchmark instances from the FairSquare benchmark. Furthermore, using INTERVALARITHMETIC only allows PV to solve 14 benchmark instances, while CROWN enables PV to solve all 18 benchmark instances.

