# OpenReview forum: "Solving Probabilistic Verification Problems of Neural Networks using Branch and Bound"
_ICML.cc/2025/Conference — ICML 2025 poster_

### Official Review · Reviewer_SiXd · 2025-03-07

**Overall Recommendation:** 4

**Summary:**

This paper is concerned with the quantitative (probabilistic) verification of neural network. It proposes PV, a branch-and-bound algorithm that is correct and (under reasonable assumptions) complete. Additionally, the paper proposes a fairness benchmark for the probabilistic verification of NNs that is more challenging than previous work.

## update after rebuttal

The authors clarified some aspects and adequately addressed my concerns in the rebuttal, I raised my score to accept.

**Claims And Evidence:**

*"Each probability distribution Px(i) needs to allow for computing the probability of a hyperrectangle in closed form. This requirement is satisfied by a large class of probability distributions..."*

The class seems pretty small to me*.

*"...including discrete distributions with a closed-form probability mass function and univariate continuous distributions with a closed-form cumulative density function, as well as Mixture Models and probabilistic graphical models (Bishop, 2007), such as Bayesian Networks, of such distributions."*

*=I think that I might be misinterpreting what "closed-form" means here, as I'm assuming it is a notion related to the complexity of the task. In  "The computational complexity of probabilistic inference using Bayesian belief networks", Cooper (AIJ, 1990) shows that SAT can be reduced to (a decision version of) inference on a binary BN. To the best of my understanding, each conditional can be computed in closed form given the parents, but the overall task is clearly NP-hard. Limiting inference to hyper-rectangles doesn't really fixes that, since the hyperrectangle could fully contain the support of the BN. Is there something I'm missing?

I checked Section A.4 only to be even more confused by this paragraph on expressivity:

*"We first show how we can apply PV to multivariate normal distributions even though they do not admit a closed-form solution
for the probability of a hyperrectangle. Consider a multivariate normal distribution Pz with mean μ and covariance Σ =
AAT . Let Px be a standard multivariate normal distribution. Now, z = Ax + μ is distributed according to Pz . Therefore,
by prepending the linear transformation Ax + μ to net, we can apply PV to multivariate normal distributions, since the
probability of a hyperrectangle under a standard multivariate normal distribution has a closed-form solution."*

Isn't the conclusion contraddicting the premise?

**Essential References Not Discussed:**

[1] "Provable preimage under-approximation for neural networks" by Zhang et al. (TACAS24), is concerned with quantitative verification of NNs. Their scope is less general, focusing on uniformly distributed inputs and ReLU nets. The proposed approach, however, has many similarities with PV:
1) They also consider input (as well as ReLU) splitting in their work and propagate linear relaxations of the volumes that need to be quantified. To the best of my understanding, they also leverage AutoLiRPA.
2) They incrementally compute tighter approximations of the verification problem.

**Experimental Designs Or Analyses:**

Despite introducing a novel benchmark, the empirical evaluation is not extremely convincing for a paper concerned with scalability.

My main criticism is that it consider low-dimensional settings only, with at most 8 input variables.
In their work [1], Zhang et al. argue that input splitting only works for low dimensional settings and propose ReLU splitting as a better alternative when the input dimensionality grows. Since PV is based on input splitting, one is left wondering what happens when the dimensionality is > 8.

Albeit restricted to uniform distributions, the experimental evaluation in [1] is more thorough, involving the verification of NN-based controllers (low input dimensionality) as well as MNIST classification robustness to patch attacks (up to 7x7 input variables).

Another problem is that, despite the focus on verification with meaningful input priors, how the approach scales to more complex priors is not analyzed. My guess is that this is somehow dictated by the approach being restricted to very simple distributions.

While I appreciate the novel setting introduced by this paper (MiniACSIncome), it seems to be tailored to PV's working assumptions. I wish it provided a more "neutral" perspective on the challenges of probabilistic verification of NNs. To the best of my understanding, it cannot be used for generating instances with input dimensionality > 8 and it cannot be used to test how a probabilistic verification algorithm scales to more expressive/complex input distributions (the BN seems to be fixed).

**Methods And Evaluation Criteria:**

The overall idea behind PV makes sense to me, but I didn't check the details of the proofs.

I am not very convinced by the emprirical evaluation (see below).

**Other Comments Or Suggestions:**

NA

**Other Strengths And Weaknesses:**

The paper tackles an important problem in ML verification and contributes with a novel algorithm as well as establishing new benchmarks.
I appreciated the related work section, which pointed me to relevant work that I missed.

I found the problem definition in Section 3.1 (Eq. 3) quite involved. What are these "satisfaction functions" useful for? How does the definition relates to the example in Eq. 1 or to other problem definitions in similar works?

Better characterizing the family of input distributions that is supported would be beneficial, the scope is not very clear at the moment.

The empirical evaluation is mostly concerned with showing scalability to larger NNs. I would assume that larger NNs are needed when the number of inputs grows and their distribution becomes very complex and multimodal. The paper doesn't make a case for adopting complex NNs in relatively simple / low-dimensional input spaces.

**Questions For Authors:**

1) The case for employing deep neural network in the settings considered in the paper is not clear. How crucial is scaling to networks with hundreds or thousands of parameters when limited to low-dimensional input spaces? Or, alternatively, how does your approach scale to higher dimensional input spaces?

2) How does PV compare with [1] in uniform settings? Or is there anything preventing a direct comparison?

3) Can you elaborate on the complexity of computing the probability of hyperrectangles in BNs? What do you mean by closed-form inference exactly? Can you provide some examples?

4) Can you elaborate on the expressiveness of the supported family of distributions? Is it able to encode complex, multimodal distributions? Can you show that empirically?

**Relation To Broader Scientific Literature:**

The related work section is well written and thorough, with one exception (see section below).

**Theoretical Claims:**

I did not carefully check the proofs.

---

> ### Author Rebuttal · Authors · 2025-03-29
>
> We thank reviewer SiXd for their review.
>
> > Q3. Closed-form inference.
>
> By "closed-form", we wanted to express that **a terminating algorithm exists** for computing a probability exactly, not necessarily that this algorithm has polynomial runtime.
> We will state this more clearly in our paper.
> For example, we use a Bayesian Network (BN) in our MiniACSIncome experiment, for which exact inference is NP-complete, as you also pointed out.
> For practical applicability, the assumption is that computing the probability of a hyperrectangle should be significantly faster than solving the overall verification problem.
> This is the case for our BN (inference is NP-complete) and the MiniACSIncome probabilistic verification problem (#P-hard).
>
> > Q4. Expressiveness of the supported family of distributions.
>
> **Yes**, we are able to encode complex, multimodal distributions!
> For example, consider Figures 7 and 8 in our Appendix, which visualise the BN of our MiniACSIncome benchmark.
> Figure 7 shows that this distribution is multimodal.
> For example, the marginal distributions of "SCHL" and "AGEP" marginal distribution have several peaks.
> Regarding complexity, these figures also show that our BN is a reasonably good fit of real-world US census data (ACSIncome).
> Therefore, we believe that our MiniACSIncome experiment demonstrates that our approach **is applicable to complex, multimodal distributions.**
>
> > Multivariate normal distributions paragraph.
>
> We will clarify this paragraph in the updated version of the paper.
> The key point here is that while *general* multivariate normal distributions do not admit a closed-form solution, *standard* multivariate normal distributions (or, more generally, multivariate normal distributions with diagonal covariance matrices) do admit a closed-form solution.
> Or idea is to emulate a general multivariate normal distribution through a standard multivariate normal distribution and an additional linear layer in the network we want to verify.
> By doing this, we can apply PV to a (general) multivariate normal distribution, although we can not use this multivariate normal distribution as an input distribution for PV directly.
>
> > Q2. How does PV compare with [1]?
>
> We would like to point out that **[1] does not provide *sound* coverage ratios for MNIST**.
> The coverage ratios they report are sampling approximations.
> While [1] also provides sound results in the "Quantitative Verification" section, these results are for 2d and 3d input spaces.
> Due to the unsoundness of the MNIST results, a comparison with our approach on this case study would be unfair, but we performed a comparison with [1] on the VCAS quantitative verification case study.
> Our PV algorithm is faster than the algorithm from [1] by two orders of magnitude for this case study.
>
> | Algorithm                | Runtime |
> |--------------------------|---------|
> | PreimageApproxForNNs [1] |  16.42s |
> | PV (Ours)                |   0.13s |
>
> Both results were obtained on the hardware also used in our paper (HW1).
> The code for running both tools is available at https://drive.google.com/drive/folders/1KBbBwzxLvCXeufo24A-cZhuvwhGTB2MD?usp=sharing.
>
> > ReLU Splitting
>
> The problem with ReLU splitting for probabilistic verification is that it creates branches that are non-convex in the input space.
> The approach of [1] to underapproximate these branches using linear relaxations is interesting.
> However, these underapproximations are still polytopes and computing the volume of a high-dimensional polytope is very costly.
> The approach in [1] to use Monte Carlo estimates of such volumes is enticing, but it entails unsoundness.
>
> > Main criticism: only low-dimensional settings.
>
> Earlier works that consider non-uniform probability distributions (FairSquare and SpaceScanner) only scale to 2d input spaces.
> Although uniform input distributions significantly simplify verification, [1] and Marzari et al. (ProVe_SLR) only demonstrate sound #DNN-verification for up to 3 and 5 input dimensions, respectively.
> In light of this, **our ability to handle 7 input variables** while not assuming uniform input distributions **is a major step forward**.
> We would also like to point out that the actual input dimension of MiniACSIncome-7 is 67 (Table 7 in our Appendix).
>
> > Q1. Large networks for low-dim settings.
>
> ACAS Xu (Julian et al., 2018) is a good example of a low-dimensional application requiring a medium-sized neural network (> 1000 parameters).
> Other examples are the neural network controllers in [1].
>
> > Satisfaction functions (Eq. 3)
>
> We agree that Equation (3) is somewhat intricate.
> This is because of the generality of our algorithm.
> The satisfaction functions encapsulate, for example, the fraction in Equation (1).
> The full satisfaction function for Equation (1) is given in Example A.1.
> Our $f_{Sat}$ function roughly corresponds to the formulae $\varphi_{\mathrm{post}}$ of Albarghouthi et al. (2017) and $\Gamma$ of Morrettin et al. (2024).

---

> > ### Comment · Reviewer_SiXd · 2025-04-03
> >
> > Thank you for the clarifications.
> >
> > I think that [1] should be cited and discussed in the paper.
> >
> > *" In light of this, our ability to handle 7 input variables while not assuming uniform input distributions is a major step forward. "*
> >
> > It is, but that also depends on how structurally complex the input distribution is. Nonetheless, this is solid work and I am happy to raise my score.

---

> > > ### Author Response · Authors · 2025-04-03
> > >
> > > Thank you for acknowledging our contribution. We will cite [1] in our paper and add the comparison we performed for the rebuttal. We agree on the point regarding the complexity of the input distribution and will extend our discussion to this end as outlined in the rebuttal. Thank you for your valuable feedback on this issue.

---

### Official Review · Reviewer_8dyc · 2025-03-10

**Overall Recommendation:** 2

**Summary:**

This paper proposed a branch & bound and interval propagation based probabilistic verification method for neural networks. This paper provides a sound verification methodology and speed up comparing to previous neural network verification techniques in mainstream neural network verification benchmarks.

**Claims And Evidence:**

The main claim of this paper is the soundness of its verification technique. As the branch and bound technique and interval propagation is widely used in nerual network verification, the soundness proof looks convincing to me.

**Essential References Not Discussed:**

No

**Experimental Designs Or Analyses:**

Yes, I have checked the experiment setting.

1. The benchmark selections align with previous works, and are well-known in this area.

2. The comparision of baseline and the new methods are confusing: I do not understand the data listed in the table, especially these percentages. This part need to be refined and more ellaboration.

**Methods And Evaluation Criteria:**

Ths benchmark datasets are currently widely-used in neural network verification tasks, and shared by a group of work in this area. The benchmark is meaningful in this area. The proposed method follows the previous working diagram to perform the verification, which satisfies the current verification application.

**Other Comments Or Suggestions:**

N/A

**Other Strengths And Weaknesses:**

Strengths:

- This paper proposed an efficient probabilistic verification technique for neural networks, which can exploit the parallism of GPUs.
- This method can be proved to be sound and complete(under some reasonable assumptions).

Weakness:

- The proposed idea is quite straightforward based on interval propagation and branch-and-bound.
- Lack of comparisons of non-probabilistic verification tools.

**Questions For Authors:**

1. The updating strategy for upper and lower bounds based on probabilistics need more clearification. Why we can directly use the probabilistic to refine the interval?

2.The branch-and-bound and interval propagation methods are widely used in neural network verification tools, and their properties have been explored a lot. This paper seems like merging them together, lacking of new insights.

3. As previous listed, the paper experiment needs more ellaboration. I do not understand the experiment setting of table 2. What is the meaning of the percentages in neural network verifications? Why not use the same setting for the baselines?

**Relation To Broader Scientific Literature:**

This paper is related to a group of work on neural network verification based on abstract interpretation and abstract refinement. The interval refinement can be viewed as a kind of abstract refinement.

**Theoretical Claims:**

The following parts are checked and I think they are ok:

- Soundness of verirication.
- Completeness proof under certain condition from 5.3.

---

> ### Author Rebuttal · Authors · 2025-03-29
>
> We thank reviewer 8dyc for their review. We address the questions posed in the review below.
>
> > 1. The updating strategy for upper and lower bounds based on probabilistics need more clearification. Why we can directly use the probabilistic to refine the interval?
>
> We assume that this question refers to Lines 7 and 8 of Algorithm 2. Please clarify in case the question is referring to a different part of our paper. We can add the probability of $\mathcal{X}\_{\mathrm{Sat}}^{(t)}$ to the lower bound, since we know that $g\_{\mathrm{Sat}}(\mathbf{x}, \mathrm{net}(\mathbf{x}) \geq 0$ for $x \in \mathcal{X}\_{\mathrm{Sat}}^{(t)}$. This is, in turn, due to the soundness of ComputeBounds (concretely, CROWN or interval arithmetic in our paper). We can compute $\mathbb{P}[\mathcal{X}\_{\mathrm{Sat}}^{(t)}]$ directly, since $\mathcal{X}\_{\mathrm{Sat}}^{(t)}$ is a union of disjoint hyperrectangles. Because of this, we can leverage the third axiom of probabilities (i.e. the probability of a countable union of disjoint sets is the sum of the probabilities of these sets). The probability of each hyperrectangle can be computed exactly due to our assumptions on the probability distributions that are described in Section 4.2.
>
> > 2. The branch-and-bound and interval propagation methods are widely used in neural network verification tools, and their properties have been explored a lot. This paper seems like merging them together, lacking of new insights.
>
> Branch and bound and bound propagation are indeed widely used for deterministic neural network verification. However, this is not yet the case for verifying *probabilistic* specifications for neural networks.
> **We use these tools and apply them to a new problem** (solving *probabilistic* neural network verification problems).
> In this context, our paper provides the following new insights:
>  1. We apply branch and bound (BaB) and bound propagation (BP) for solving probabilistic neural network verification problems, a class of problems to which these techniques have not yet been applied to.
>  2. We conduct a thorough theoretical analysis of our BaB and BP-based approach. To the best of our knowledge, there are no previous results on the theoretical properties of BaB and BP for *probabilistic* verification problems.
>  3. Our experiments demonstrate that our proposed approach (based on BaB and BP) provides a substantial runtime advantage for solving *probabilistic* verification problems. The magnitude of this runtime advantage is a surprising novel insight.
>
> > 3. As previous listed, the paper experiment needs more ellaboration. I do not understand the experiment setting of table 2. What is the meaning of the percentages in neural network verifications? Why not use the same setting for the baselines?
>
> The percentages in Table 2 are probabilities given as percentages. We will clarify this in the table caption.
> The "VR" percentages in this table are the exact probabilities, that is, the verification result of quantitative verification.
> The probabilities for ProbabilityBounds are pairs of lower and upper bounds on these values.
> The percentages for $\varepsilon$-ProVe are 99.9% confidence upper bounds on these values.
> **Overall, these percentages are *results*, not *settings* of the verifiers.**
>
> If the remark regarding using the same settings for the baselines refers to the timeout for ProbabilityBounds (10s, 1m, 1h): these settings are not applicable for ProVe\_SLR and $\varepsilon$-ProVe, since these algorithms do not provide intermediate sound results.
> This is one of the advantages of ProbabilityBounds that we seek to demonstrate in Table 2.
>
> > [Weakness:] Lack of comparisons of non-probabilistic verification tools.
>
> Since **non-probabilistic verifiers can not be applied to probabilistic verification problems**, it is not possible to compare our approach to them experimentally.
> For example, it is impossible to apply $\alpha$-$\beta$-CROWN to the FairSquare benchmark, since $\alpha$-$\beta$-CROWN can not handle probabilities.
>
> To position our work more clearly in the context of other verification approaches, we provide the following table:
>
> | Method     | Deterministic Statements | Quantitative Statements (#DNN-Verification) | Group Fairness | General Probabilistic Statements |
> |---|---|---|---|---|
> | $\alpha,\beta$-CROWN, NNV | yes | no | no | no |
> | ProVe_SLR (Marzari et al., 2023b) | no | yes | no | no |
> | SpaceScanner (Converse et al., 2020) | no | yes | yes | yes |
> | FairSquare (Albarghouti et al., 2017) | no | no | yes | no |
> | PV (Ours)       | no | yes | yes | yes |
>
> The rebuttal for reviewer BFqm includes an extended version of this table that we will add to our related work section.
> We hope that this clarifies the relation between our approach and non-probabilistic verifiers.
> If there remain further questions, we would be happy to address them in the second round of the author-reviewer discussion.

---

### Official Review · Reviewer_GWj1 · 2025-03-12

**Overall Recommendation:** 4

**Summary:**

The paper proposes a generic approach for the probabilistic verification of neural networks which can leverage pre-existing qualitative verification tools. To this end, the paper proposes to combine input space splitting with bound computation to compute bounds on (un)safe behavior. Purely safe/unsafe regions can then be probabilistically quantified. The paper also puts forward an argument for the completeness of the approach.

## update after rebuttal
The authors managed to lift my concern about the theoretical assumptions and promised to clarify this misunderstanding in the paper's final draft. I trust this will happen and am thus in favour of accepting the paper.

Concerning the comments from other reviewers, I believe it's important to emphasize that the verification guarantees derived in the paper are meant to hold *globally*, i.e. on large parts of the input space. It is common knowledge that NN verification for lage input spaces can (so far) only be scaled to small input dimensions and/or small NNs. Consequently, I am happy with the presented experiments. This can also be seen in other work that aims to derive global guarantees (e.g. [CAV24]).

[CAV24] https://link.springer.com/chapter/10.1007/978-3-031-65630-9_17

**Claims And Evidence:**

The proposed approach comes with correctness and completeness claims (see Theoretical Claims) and is evaluated on multiple benchmark sets that compare the approach to competing techniques (see Methods and Evaluation Criteria).
I find the empirical evaluation convincing, my concerns about the theoretical claims are discussed in below.

**Essential References Not Discussed:**

While it should probably be deemed concurrent literature, a recent ICSE paper by Kim et al. [1] proposes the same approach of bounding probabilities [1]. In particular, just like PV (Algorithm 1) the paper also bounds probability by computing hyperrectangles which are fully safe/unsafe (see Algorithm 1 and Section IV in [1]). However, this paper is specific to fairness and hence less broad in its scope.
Moreover, the paper was first published on arXiv on 5th September 2024...

[1] https://arxiv.org/abs/2409.03220

**Experimental Designs Or Analyses:**

From reading of Chapter 6 the experimental design and the derived conclusions seem sound to me.

**Methods And Evaluation Criteria:**

The benchmarks seem well chosen to me. The work does not reproduce all results from competing papers on the same machine, but instead reuses the numbers of other authors while using a weaker machine for the own experiments. While this is not ideal, it seems to me the evaluation is nonetheless sound and clearly demonstrates the superority of the approach at hand.

**Other Comments Or Suggestions:**

In Example A.1 you reformulated demographic parity. It seems to me, that the problem would turn out to be computationally easier if the division was resolved by multiplying the divisor on both sides of the inequality (admissible due to non-negativity) and then subtracting $\gamma$*divisor. Is there any particular reason you did not do this?

**Other Strengths And Weaknesses:**

A major strength of the paper is its generality:
The approach supports both discrete and continuous input spaces, a broad range of probability distributions and builds upon off-the-shelf tools for qualitative verification. In particular, this allows the approach to benefit from future improvements in verification technology.

**Questions For Authors:**

**(Q1)** Am I missing something about assumption 5.3. or is it indeed more restrictive than discussed in the paper? If so, might a focus on completeness for open properties be a more natural formulation?
**(Q2)** You mention support for unbounded hyberrectangles. I am under the impression that this is not something typically supported by the qualitative verification tools used by the approach. How do you handle this in practice? Did you have any benchmarks for this setting?

**Relation To Broader Scientific Literature:**

There is relatively little literature on probabilistic verification of neural networks which, in my view, makes this contribution all the more welcome. The few papers that do exist are cited and are used for comparison in Section 6. The presented approach clearly outperforms alternative techniques.

**Theoretical Claims:**

The paper makes two theoretical claims: Soundness (Corollary 5.2) and Completeness (Theorem 5.5).
While I agree with the former, I am worried about the assumption necessary for completeness (Assumption 5.3).
My hope is that we can resolve this issue through the rebuttal in which case I'm happy to raise my score:

The verification problem as specified in (3) requires that $f(p_1,\dots,p_v)$ must be non-negative where $p_i$ is the probability that some $g^{(i)}(\dots)$ is non-negative. Assumption 5.3 now requires that $f$ is never zero and the probability of $g$ being zero is zero. It is then noted that this assumption is only "mildly restrictive" as we can strengthen (3) to be a constraint of the form $\geq \varepsilon$.
It seems to me, that this is beside the point: Of course, I can add $\varepsilon$ to $f$, but in most cases (especially the ones discussed in the paper) $f$ and $g$ are continuous and then there will be other inputs for which $f$ becomes zero (of for which $g$ becomes zero and which do not have zero probability).
More intuitively: It seems to me that it is quite a natural phenomenon that the boundary between the safe and unsafe regions may not be axis-aligned. In this case, axis-aligned splitting as done in this paper *cannot* derive exact probability bounds but only approximate them.
It seems to me a more natural assumption for Theorem 5.5 might be simply that it is complete for open properties (i.e. $>$ in both equations in (3))? Alternatively, it seems to me at the very least this restriction warrants further discussion.

---

> ### Author Rebuttal · Authors · 2025-03-29
>
> We thank reviewer GWj1 for their review, in particular their careful evaluation of our theoretical assumptions.
>
> > Theoretical Claims / Assumption 5.3
>
> If the safe/unsafe region is not axis-aligned, indeed, Algorithm 2 can not compute the *exact* probability of safety.
> However, Algorithm 2 still computes a converging sequence of bounds on the unknown exact probability $p_i$.
> The question that motivates Assumption 5.3 is when converging bounds are sufficient for solving a probabilistic verification problem.
>
> For illustration, assume we want to show $y \geq 0$ and have converging sequences of bounds ${(\ell\_t)}\_{t \in \mathbb{N}}$ and ${(u\_t)}\_{t \in \mathbb{N}}$ with $\ell_t \leq y \leq u_t$ for each $t \in \mathbb{N}$ and $\lim_{t \to \infty} \ell_t = \lim_{t \to \infty} u_t = y$.
> If the actual (exact) $y$ is zero, sequences of bounds that only converge in the limit do not suffice to prove $y \geq 0$, since there may be no $T \in \mathbb{N}$ with $\ell_T = 0$.
> However, if the actual value of $y$ is different from zero, knowing a finite number of iterates of ${(\ell_t)}\_{t}$ and ${(u_t)}\_{t}$ always suffices to prove or disprove $y \geq 0$.
> Concretely, there will be a $T \in \mathbb{N}$, such that either $\ell_T > 0$ or $u_T < 0$, that proves, respectively, disproves $y \geq 0$.
>
> Unfortunately, the situation does not change fundamentally if we study $y > 0$ instead of $y \geq 0$.
> Before, if $y = 0$, our specification actually held, but we were not able to ever prove this.
> Now, our specification $y > 0$ is actually violated if $y = 0$, but we will never be able to disprove it.
> Concretely, if ${(u_t)}_{t}$ only converges in the limit, there may be no $T \in \mathbb{N}$, such that $u_T = 0$.
>
> Our assumption that $f\_{Sat}(p_1, \ldots, p_v) \neq 0$ corresponds to assuming $y \neq 0$ in the discussion above.
> Note that $f\_{Sat}(p_1, \ldots, p_v)$ is a constant in Equation (3).
> Our comment on studying $f\_{Sat}(...) \geq \varepsilon$ instead of $f\_{Sat}(...) \geq 0$ means solving a different verification problem with $f'\_{Sat}(...) = f\_{Sat}(...) - \varepsilon$.
> Practically, if you only want to show Equation (1) and PV seems not to be terminating for $\gamma = 0.8$, you can run PV again with $\gamma' = 0.81$ and find that your neural network either violates or satisfies a slightly stronger fairness specification.
>
> The motivation for requiring $p = \mathbb{P}[g\_{Sat}(...) = 0] = 0$ is similar.
> However, the reason why it is not restrictive to assume $p = 0$ is different.
> Unlike $f\_{Sat}(...)$, $g\_{Sat}(...)$ is not a constant in Equation (3).
> Here, our argumentation why the assumption that $p = 0$ is not restrictive is concerned with the level sets $G\_r = \\{\mathbf{x} \in \mathbb{R}^n \mid g\_{Sat}(\mathbf{x}, \mathrm{net}(\mathbf{x})) = r\\}$ of $g$ that have positive probability ($\mathbb{P}[G\_r] > 0$).
> If $\mathbb{P}[G\_r] > 0$ for some $r$, $G\_r$ also needs to have positive volume in $\mathbb{R}^n$ (we assume a continuous probability distribution here; for discrete distributions we actually do not require the assumption p = 0 due to splitting differently).
> A set $G\_r$ having positive volume means that the plot of $g\_{Sat}(...)$ has a "flat" part where $g\_{Sat}(...) = r$.
> Unless $g\_{Sat}$ has a continuous range of "flat parts", we can find a small $\varepsilon > 0$, such that $\mathbb{P}[g\_{Sat}(...) = \varepsilon] = 0$.
> While there may be pathological functions that have continuous ranges of "flat parts", neural networks with finitely many neurons with standard activation functions can not have such continuous ranges.
>
> We hope this discussion clarifies our assumption and why we consider it only mildly restrictive.
> We will use your feedback and the above discussion to improve the description of Assumption 5.3 in our paper.
>
> > Unbounded Input Spaces (Q2)
>
> Indeed, qualitative verifiers rarely support unbounded input spaces.
> In practice, we use $\underline{\mathbf{y}} = -\infty$ and $\underline{\mathbf{y}} = \infty$ for unbounded branches in line 5 of Algorithm 2.
> With this, unbounded branches are never pruned, and LongestEdge always first selects dimensions that are unbounded in a branch.
> We modify BaBSB to also first select dimensions that are unbounded in a branch.
> Unfortunately, this was not documented in our submission, but we will add this to the last paragraph in Section 4.1.
>
> The FairSquare benchmark (Section 6.1) features an unbounded input space (discussed in Appendix F.2.1).
>
> > Related Work: Individual Fairness (Kim et al.)
>
> Besides what was already noted by the reviewer, Kim et al. study individual fairness, which is a non-probabilistic fairness specification.
> Therefore, the work of Kim et al. is more closely related to the papers of Marzari et al. (2023a, 2023b, 2024), than to our approach.
> Nonetheless, this paper is important related work and we will incorporate it into our discussion of #DNN-verification in Section 2.

---

> > ### Comment · Reviewer_GWj1 · 2025-04-02
> >
> > I thank the authors for their detailed response.
> > I believe what confused me about the (non-)restrictiveness of Assumption 5.3 was that it sounded to me as though for any NN and specification we can rephrase the specification *equivalently* such that Assumption 5.3 is satisfied and the completeness guarantee holds.
> > This is not the case and also not what was meant by the authors. Instead, the idea is that there is a "semantically close" specification which evades the completeness issue.
> > This is a statement I can get behind.
> > I encourage the authors to update the draft to make this clearer and will, as promised, raise my score.

---

> > > ### Author Response · Authors · 2025-04-02
> > >
> > > We are happy that we could clarify our assumptions in the rebuttal and will use the reviewer's feedback to improve the discussion of our assumptions in the paper. We want to thank the reviewer for their helpful comments and questions.

---

### Official Review · Reviewer_BFqm · 2025-03-14

**Overall Recommendation:** 2

**Summary:**

This paper proposes a probabilistic neural network verification method through lower and upper bounds of probabilities of outputs. By branch and bound and bound propagation, it can run faster than other verifiers.

**Claims And Evidence:**

The definitions of soundness and completeness in this paper seem to be quite different from the commonly used ones in the neural network verification community, where soundness usually means once the verifier gives true, the output specification will truly hold. In this sense, probabilistic verification seems to be a false proposition. I suggest the authors should clarify it clearly at the very beginning of the paper to avoid potential confusion.

**Essential References Not Discussed:**

A key branch of probabilistic certification/verification methods in the literature is missing, i.e. randomized smoothing based neural network verification methods [1,2,3].

[1] Cohen et al. Certified Adversarial Robustness via Randomized Smoothing, 2019

[2] Yang et al. Randomized Smoothing of All Shapes and Sizes, 2020

[3] Li et al. Sok: Certified robustness for deep neural networks, 2020

**Experimental Designs Or Analyses:**

As a neural network verification method in machine learning, experiments are expected to involve  adversarial machine learning tasks, e.g. mnist with adversarial perturbation. More can be found in VNN-COMP benchmarks in recent years. Besides, some important baselines are missing, e.g. several best non-probabilistic verifiers $\alpha$-$\beta$-CROWN, NNV, etc.

**Methods And Evaluation Criteria:**

- In Eq (3), I wonder whether each support of $P_{x^{(i)}}$ is assumed to be not overlapped. If so, the joint distribution is assumed to be invariant to the permutation of each distribution, which should be explicitly discussed.

- In Alg 1, it is not clear if $PB_i$ is also related to $t$ on Line 5. It should be stated clearly on Line 2 as well.

- In Alg 2, Line 7 and Line 8 seem to be conflicted with the statement of the right column on Line 220, where $l^t$ is the probability of a union of some sets. However, it should follow the union bound with inequality. At the same time, in Alg 2, the iteration of lower and upper bounds is assumed to be with the equality condition of the union bound, which may not hold and cause correctness issues.

**Other Comments Or Suggestions:**

See above

**Other Strengths And Weaknesses:**

See above

**Questions For Authors:**

See above

**Relation To Broader Scientific Literature:**

Related to neural network verification.

**Theoretical Claims:**

The proof of Theorem 5.1 seems to hold with the assumption of uniform distribution and disjoint support of each single-value distribution. It holds in such special cases, however, more formal and general propositions should be derived based on formal arguments, e.g. through conformal prediction or hypothesis testing.

---

> ### Author Rebuttal · Authors · 2025-03-29
>
> We thank reviewer BFqm for their review. We believe there has been a misunderstanding of our definition of soundness that underlies most of the review.
>
> > The definitions of soundness [...] in this paper seem to be quite different from the commonly used ones in the neural network verification community [...]. In this sense, probabilistic verification seems to be a false proposition.
>
> **We actually use the standard definition of soundness**, as expressed by Definition 3.2 in our paper.
> The review appears to assume that we provide what we call a "probably sound" algorithm, for which a verifier output of true means that the output specification *likely* holds, typically with a certain confidence level (PAC-style guarantee).
> We instead provide a ***definite* guarantee for a *probabilistic* statement**.
>
> We agree that the term "probabilistic verification" can be misleading.
> Therefore, we propose to change the title of our paper to "Solving Probabilistic Verification Problems of Neural Networks using Branch and Bound" and similarly modify our abstract and introduction to avoid this source of confusion.
> If there are further suggestions where our presentation can be improved in this regard, we would be happy to incorporate them.
>
> To summarise, our paper provides a **sound algorithm** for solving **probabilistic verification problems** (PVPs), that is, proving statements involving probabilities over neural networks, such as fairness statements of the type of Equation (1).
> Other methods, like hypothesis testing, provide PAC-style guarantees for PVPs that only hold with a certain probability.
> **Unlike these methods, our guarantees hold with certainty.**
>
> > In Eq(3), I wonder whether each support of $P_{x^{(i)}}$ is assumed to be not overlapped.
> > The proof of Theorem 5.1 seems to hold with the assumption of uniform distribution and disjoint support of each single-value distribution.
>
> We do not require these assumptions.
> Our assumptions are summarized in Section 4.2
> If you have concerns regarding particular steps of the proofs (Appendix C), we would be happy to discuss them in the second round of the author-reviewer discussions.
>
> > In Alg 1, it is not clear if $PB_i$ is also related to $t$ on Line 5
>
> In Algorithm 1, $PB_i$ is not dependent on $t$.
> However, $PB_i$ is an instantiation of Algorithm 2 rather than a fixed value.
> Note that Algorithm 2 also has a loop counter $t$.
> While some $PB_i$ may advance faster than others in our implementation, the counters $t$ in Algorithms 1 and 2 are synchronized in our paper to simplify the exposition.
>
> > In Alg 2, Line 7 and Line 8 [...] conflicted with [...] Line 220.
>
> Our approach is **not based on the union bound** but rather on the third axiom of probabilities (the probability of a countable union of disjoint sets is the sum of the probabilities of these sets).
> The third axiom of probabilities holds with equality.
>
> > Adversarial machine learning.
>
> **We consider adversarial machine learning tasks in our ACAS Xu robustness experiment.**
> Unlike probably sound verifiers, sound verifiers for PVPs (such as PV) do not currently scale to image datasets, but our work provides major advances in scaling to larger input spaces.
> We discuss this in more depth in the rebuttal to reviewer SiXd.
>
> > [...] some important baselines are missing, e.g.[...] $\alpha$-$\beta$-CROWN, NNV, etc.
>
> Since **non-probabilistic verifiers can not be applied to PVPs**, it is not possible to compare our approach to these verifiers experimentally.
> For example, it is impossible to apply $\alpha$-$\beta$-CROWN to MiniACSIncome, since $\alpha$-$\beta$-CROWN can not handle probabilities.
>
> To position our work more clearly in the context of other verification approaches, we will add the following table to our related work section:
>
> | Method     | Deterministic Statements | Quantitative Statements (#DNN-Verification) | Group Fairness | General PVPs | Type of Verifier Guarantee |
> |---|---|---|---|---|---|
> | $\alpha,\beta$-CROWN, NNV | yes | no | no | no | sound |
> | Randomized Smoothing  | yes | no | no | no | probably sound (i.e., PAC-style) |
> | Hypothesis Testing | yes | yes | yes | yes | probably sound (i.e., PAC-style) |
> | ProVe_SLR (Marzari et al., 2023b) | no | yes | no | no | sound |
> | SpaceScanner (Converse et al., 2020) | no | yes | yes | yes | sound (quantitative statements), unsound (otherwise)
> | FairSquare (Albarghouti et al., 2017) | no | no | yes | no | sound |
> | PV (Ours)       | no | yes | yes | yes | sound |
>
> We hope that this clarifies the relation between our method (PV) and other verifiers solving deterministic (non-probabilistic) verification tasks, as well as other methods mentioned in the review.
>
> >  key [...] literature is missing.
>
> Thank you for mentioning this.
> We will add the mentioned references to our discussion of probably sound approaches in Section 2 (third paragraph).

---

> > ### Comment · Reviewer_BFqm · 2025-04-02
> >
> > Thanks for the clarification of sound probabilistic verification and I will raise my score to 2. It seems the definite guarantee for a probabilistic statement can only be adopted in non-deterministic cases. However, a CROWN-like non-probabilistic verifier can be applied to the special case of the probabilistic verification problem with the probability of 1, which is supposed to be compared as a baseline regarding tightness and scalability. A comparison with a CROWN-like verifier with branch-and-bound is still expected under some fair settings.

---

> > > ### Author Response · Authors · 2025-04-03
> > >
> > > *We edited this comment. Edits are typeset in italics.*
> > >
> > > Thank you for considering our rebuttal and raising your score.
> > > Following your suggestion to perform an experimental comparison with a CROWN-based verifier, we experimentally compared our approach to $\alpha,\beta$-CROWN on the ACAS Xu safety benchmark described in Section 6.2 of our paper.
> > > *Our experiment shows that the tools are complementary and perform well at their respective tasks.*
> > > For non-probabilistic verification on ACAS Xu, $\alpha,\beta$-CROWN is the current state-of-the-art tool, as determined in VNN-COMP 2024 (Brix et al., 2024).
> > >
> > > *For this experiment, non-probabilistic verification corresponds to proving $\forall x: \neg(net(x) \text{ is unsafe})$, which is almost equivalent to $\mathbb{P}[net(x) \text{ is unsafe}] = 0$, except for the handling of probability-zero events.*
> > > *When ignoring probability-zero events, it is possible to disregard the probability distribution entirely to solve this verification problem.*
> > > *This simplifies the verification problem: non-probabilistic verification is NP-complete, while general probabilistic verification is #P-hard (see Section 3.1 in our paper).*
> > > *Due to the issue of probability-zero events and the different complexity classes, **a comparison of our approach with $\alpha,\beta$-CROWN is not entirely faithful and not entirely fair**.*
> > > *We performed the comparison regardless to demonstrate the differences between non-probabilistic verification and probabilistic verification.*
> > >
> > > The table below contains a selection of results from our experiment.
> > > Due to the character limit, we can not present the full results here.
> > > The entire table is available at https://drive.google.com/file/d/1OO8k-S-HCOLsGGolW_dxAPUm1V9gkDDA/view?usp=sharing.
> > > The results for our approach are taken from Table 4 in the Appendix of our paper.
> > >
> > > | net | Non-Probabilitic Verification Result ($\alpha,\beta$-CROWN) | Quantitative Verification Result ($\alpha,\beta$-CROWN) | Runtime ($\alpha,\beta$-CROWN) | Non-Probabilitic Verification Result (ProbabilityBounds, Ours, 60s timeout) | Quantitative Verification Result (ProbabilityBounds, Ours, 60s timeout) |
> > > |---|---|---|---|---|---|
> > > | $N_{3,1}$ | $\color{red}\times$       | $0\\%, 100\\%$ | $<1s$  | $\color{red}\times$ | $  0.88\\%,   2.74\\%$
> > > | $N_{3,2}$ | $\color{red}\times$       | $0\\%, 100\\%$ | $<1s$  | $?$                 | $  0.00\\%,   1.15\\%$
> > > | $N_{3,3}$ | $\color{green}\checkmark$ | $0\\%,0\\%$    | $451s$ | $?$                 | $  0.00\\%,   0.84\\%$
> > > | $N_{3,4}$ | $\color{red}\times$       | $0\\%, 100\\%$ | $<1s$  | $\color{red}\times$ | $  0.13\\%,   1.53\\%$
> > > | $N_{3,5}$ | $\color{red}\times$       | $0\\%, 100\\%$ | $<1s$  | $\color{red}\times$ | $  0.63\\%,   2.01\\%$
> > > | $N_{3,6}$ | $\color{red}\times$       | $0\\%, 100\\%$ | $<1s$  | $\color{red}\times$ | $  0.02\\%,   5.18\\%$
> > > | $N_{3,7}$ | $\color{red}\times$       | $0\\%, 100\\%$ | $<1s$  | $?$                 | $  0.00\\%,   3.32\\%$
> > > | $N_{3,8}$ | $\color{red}\times$       | $0\\%, 100\\%$ | $<1s$  | $\color{red}\times$ | $  0.32\\%,   3.46\\%$
> > > | $N_{3,9}$ | $\color{red}\times$       | $0\\%, 100\\%$ | $<1s$  | $\color{red}\times$ | $  1.39\\%,   3.72\\%$
> > >
> > > In this table, the "Non-Probabilistic Verification Result" columns indicate whether the respective verifier determined that the network is safe ($\color{green}\checkmark$), unsafe ($\color{red}\times$), or whether it did not provide a conclusive result ($?$).
> > > The "Quantitative Verification Result" columns contain a lower and an upper bound on the probability of an unsafe network output under a uniform input distribution, as computed by the verifiers.
> > >
> > > **This experiment demonstates that $\alpha,\beta$-CROWN and our approach both shine in their own domains**.
> > > While $\alpha,\beta$-CROWN is fast at solving the non-probabilistic verification problem, its results only entail trivial bounds on the probability of unsafety, since $\alpha,\beta$-CROWN is not conceived for performing quantitative verification.
> > > On the other hand, our approach is less effective at performing non-probabilistic verification (for which it was not conceived) but can provide insightful bounds on the probability of unsafety.
> > > **As such, the two approaches complement each other.**
> > > This is typically the idea behind quantitative verification: finding out the probability of unsafety, *after* a non-probabilistic verifier proved that there is unsafety.
> > >
> > > To summarize our results:
> > >  - For tightness, our approach outperforms $\alpha,\beta$-CROWN, since $\alpha,\beta$-CROWN can only provide trivial bounds on probabilities.
> > >  - For scalability, $\alpha,\beta$-CROWN outperforms our approach on non-probabilistic verification problems, since it is specialized to these problems.
> > >
> > > We hope this comparison demonstrates the usefulness and strengths of our approach compared to CROWN-based verifiers.
> > > We kindly ask you to reevaluate our paper in light of these qualities.

---

### Decision · Program_Chairs · 2025-05-01

**Decision:**

Accept (poster)

**Comment:**

This paper proposes a new algorithm for neural network probabilistic verification based on branch-and-bound and bound propagation. The  algorithm can compute and refine rigorous lower and upper bounds of probabilities satisfying some properties when model's input is following a probabilistic distribution. The positive sides are the novel application of branch-and-bound and bound propagation to the probabilistic verification problem, comprehensive evaluation showing effectiveness, and significant clarification efforts during rebuttal that address most of the reviewers' comments. The negative sides are limited applicability when input dimensionality is high and a relatively straightforward methodology. After thorough discussion among reviewers and AC, we recommend acceptance. Congratulations! Authors should incorporate the reviewer's feedback and discussion into the camera-ready version.